# Beyond Gaussian Initializations:
# Signal Preserving Weight Initialization for Odd-Sigmoid Activations

## Abstract

Activation functions critically influence trainability and expressivity, and recent work has therefore explored a broad range of nonlinearities. However, widely used Gaussian i.i.d. initializations are designed to preserve activation variance under wide or infinite width assumptions. In deep and relatively narrow networks with sigmoidal nonlinearities, these schemes often drive preactivations into saturation, and collapse gradients. To address this, we introduce an odd-sigmoid activations and propose an activation aware initialization tailored to any function in this class. Our method remains robust over a wide band of variance scales, preserving both forward signal variance and backpropagated gradient norms even in very deep and narrow networks. Empirically, across standard image benchmarks we find that the proposed initialization is substantially less sensitive to depth, width, and activation scale than Gaussian initializations. In physics informed neural networks (PINNs), scaled odd-sigmoid activations combined with our initialization achieve lower losses than Gaussian based setups, suggesting that diagonal-plus-noise weights provide a practical alternative when Gaussian initialization breaks down.

## 1 Introduction

Deep learning has advanced substantially across diverse domains (LeCun et al., 2015; He et al., 2016; Hwang et al., 2022). A central ingredient in these successes is the choice of activation function, which controls both the expressive power of a network and the way signals and gradients propagate through depth (Poole et al., 2016). Recent work has proposed a wide variety of nonlinearities beyond classical sigmoid and ReLU (e.g., GELU, Swish, Mish, SELU) to improve trainability, stability, and accuracy (Hendrycks & Gimpel, 2016; Ramachandran et al., 2017; Misra, 2019; Klambauer et al., 2017; Murray et al., 2022; Bingham & Miikkulainen, 2023; Zhang et al., 2024). However, activation functions and weight initialization are tightly coupled (He et al., 2015; Lee et al., 2024). The scale and shape of the nonlinearity determine how information propagates with depth. As a result, an initialization that is not tuned to the chosen activation can drive activations into saturation and cause gradients to vanish or explode, even when the activation itself is reasonable. These issues are especially pronounced for sigmoidal activations, which are widely used in sequence models and physics informed neural network (PINN) (Raissi et al., 2019).

Standard Gaussian i.i.d. initializations (Xavier (Glorot & Bengio, 2010), He (He et al., 2015), EOC (Hayou et al., 2019) are derived under wide or infinite width assumptions and choose a single variance parameter to preserve signal propagation. In practice, we find that this design breaks down in deep and relatively narrow networks and under activation rescaling: small deviations in the variance can trigger saturation, or extreme learning rate sensitivity. Motivated by these

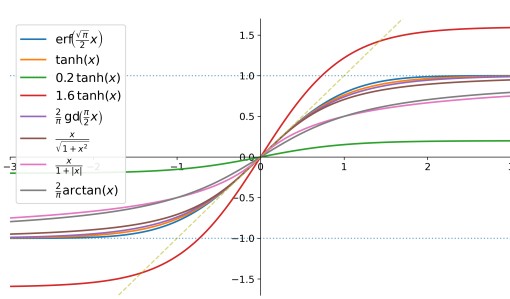

Figure 1: odd–sigmoid activations

observations, we propose an activation aware initialization for fully feedforward neural networks (FFNNs) with odd–sigmoid activations in the class $\mathcal{F}$ (Figure 1). For any $f \in \mathcal{F}$ with $\omega := 1/f'(0)$, we study the scalar dynamics of $x \mapsto f(ax)$ and choose the layerwise gains $a$ from a distribution with mean $\omega$ so that trajectories neither collapse to zero nor saturate. This gain design is then implemented via a simple initial weight matrix whose effective gain statistics match the desired behavior. Unlike Gaussian i.i.d. schemes, whose trainability is highly sensitive to the exact variance, the proposed initialization keeps both forward activations and backpropagated gradients in a well conditioned range across depth (Raghu et al., 2017). Mean field analysis further shows that the corresponding gradient amplification factor $\chi_\ell$ remains close to 1 over a significantly wider band of noise scales (Schoenholz et al., 2016).

We evaluate the proposed initialization in extensive experiments with odd–sigmoid activations, covering both the regime $\omega \approx 1$ and the more challenging regime $\omega \gg 1$ or $\omega \ll 1$. On standard image classification benchmarks, Gaussian initializations often fail or degrade sharply in deep, narrow, or rescaled-activation networks, whereas our scheme trains stably and is more data-efficient than these Gaussian baselines. We also study scaled and composite activations (e.g., $\alpha f(x)$, $f(\alpha x)$, and positive linear combinations) and observe that our initialization remains stable even for extreme scales (up to $\alpha \approx 10^9$). In PINN settings, appropriately scaled odd–sigmoid activations combined with our initialization consistently achieve lower losses than EOC-based baselines and do so without batch normalization (BN) (Ioffe, 2015), indicating that diagonal–plus–noise weights provide a practical alternative when Gaussian schemes become brittle. Our main contributions are as follows:

- We define the odd–sigmoid function class and characterize its properties (Section 4.1).
- We view a deep feedforward network as a parallel ensemble of scalar dynamical systems, and use the resulting gain statistics to design a proposed initialization (Section 4.2).
- We show that the proposed initialization is more variance robust than standard Gaussian i.i.d. initializations and better preserves forward and backward signals (Section 4.3).
- We empirically validate the proposed initialization across a wide range of activations on standard image classification benchmarks and PINNs (Section 5.1 and 5.2).

## 2 RELATED WORK

Classic initialization schemes were originally designed to stabilize layerwise variance, and later work showed that good performance also hinges on well-tuned statistics and momentum, with poor initialization causing saturation, variance collapse, and learning-rate fragility (Sutskever et al., 2013; Narkhede et al., 2022). For sigmoidal networks, follow-up studies adjust the mean or support of the weights or use more spread-out mappings to keep units responsive and accelerate convergence (Yilmaz & Poli, 2022; Qiao et al., 2016; Sodhi et al., 2014). Beyond first-order variance matching, curvature- and correlation-based criteria (Hessian-norm control, depth×width bounds, and propagation analyses in constrained architectures such as input-convex networks and transformers) and architectural tweaks such as ReZero and dynamical-isometry–preserving setups all point toward initializations that are explicitly calibrated to the activation and the network's depth and shape (Skorski et al., 2021; Iyer et al., 2023; Hoedt & Klambauer, 2023; Noci et al., 2022; Bachlechner et al., 2021; Lee et al., 2024; Price et al., 2024; Blumenfeld et al., 2020). Our work makes this activation–initializer coupling explicit by formalizing an odd–sigmoid class and, for any $f$ in this class, deriving a closed-form noise scale that keeps preactivations in a high-gain regime up to a target depth, yielding dispersed forward signals and stable gradients without normalization (Noci et al., 2022; Skorski et al., 2021).

On the activation side, early bounded nonlinearities (Sigmoid, Tanh) suffered from vanishing gradients, whereas ReLU variants improved efficiency but can produce dying neurons (Adeli et al., 2025). Surveys emphasize that no single activation is universally optimal and highlight desirable traits such as continuity, boundedness, monotonicity, symmetry, and smoothness (Alcaide et al., 2025; Adeli et al., 2025), with bounded, odd-symmetric forms helping to center signals and stabilize optimization—directly motivating our initializer. In parallel, adaptive and learned activation families (e.g., parametric activations for DL and PINNs, error-function–corrected ReLU variants, per-neuron self-activating networks, and tailored activations for zeroing neural networks) demonstrate that carefully structured nonlinearities can improve convergence and robustness (Jagtap et al., 2020; Wang et al.,

2023; Ullah et al., 2025; Tutuncuoglu, 2025; Liu, 2025), further supporting the need for initialization strategies aligned with activation geometry.

## 3 PRELIMINARIES

In this section, we introduce the notation used throughout the paper, review a simplified elementwise analysis of signal propagation (Lee et al., 2025), and recall the classical edge of chaos mean field theory (Schoenholz et al., 2016) and its associated Gaussian i.i.d. EOC initialization ((Hayou et al., 2019)).

**Notation.** We consider a feedforward neural network with $L$ layers. The dataset comprises $K$ pairs $\{(\boldsymbol{x}_i, \boldsymbol{y}_i)\}_{i=1}^K$, where $\boldsymbol{x}_i \in \mathbb{R}^{N_x}$ is the input and $\boldsymbol{y}_i \in \mathbb{R}^{N_y}$ is the corresponding target. For $\ell = 1, \ldots, L$, the layerwise update is

$$\boldsymbol{h}^\ell = \mathbf{W}^\ell \boldsymbol{x}^{\ell-1} + \mathbf{b}^\ell, \qquad \boldsymbol{x}^\ell = f(\boldsymbol{h}^\ell) \in \mathbb{R}^{N_\ell},$$

where $\mathbf{W}^\ell \in \mathbb{R}^{N_\ell \times N_{\ell-1}}$ is the weight matrix, $\mathbf{b}^\ell \in \mathbb{R}^{N_\ell}$ is the bias vector, and $f(\cdot)$ denotes an activation function. We write $\mathbf{W}^\ell = [w_{ij}^\ell]$.

**Signal Propagation Analysis.** Following the simplified framework of Lee et al. (2025), we analyze signal propagation in feedforward networks with an elementwise activation $f$. For convenience, all layers, including input and output, are assumed to have dimension $n$ (i.e., $N_\ell = n$ for all $\ell$). A key observation of this approach is that the usual matrix–vector multiplication $\mathbf{W}^\ell \boldsymbol{x}^{\ell-1}$ can be reformulated elementwise, such that each coordinate is expressed in terms of an effective self-scaling factor. Formally, for $\ell = 0, \ldots, L-1$ and $i = 1, \ldots, n$,

$$x_i^{\ell+1} = f(a_i^{\ell+1} x_i^\ell), \qquad a_i^{\ell+1} = w_{ii}^{\ell+1} + \sum_{j \neq i} \frac{w_{ij}^{\ell+1} x_j^\ell}{x_i^\ell}, \tag{1}$$

assuming that $\mathbf{b}^\ell = \mathbf{0}$. This elementwise decomposition highlights the role of $a_i^{\ell+1}$ as an effective gain that combines both diagonal and off-diagonal terms and provides a tractable basis for studying signal propagation in deep networks.

**Edge of chaos.** We briefly recall the "edge of chaos" viewpoint based on mean field theory for fully connected networks with standard i.i.d. Gaussian initialization, where the weights are sampled as $w_{ij}^\ell \sim \mathcal{N}(0, \sigma_w^2/N_\ell)$ (Poole et al., 2016; Schoenholz et al., 2016). In this setting, the preactivations $h_i^\ell$ are modeled as i.i.d. Gaussians with layerwise variance $q^\ell$, and both forward signals and backward gradients admit simple recursions in the infinite width limit. In particular, the mean field analysis of backpropagation shows that the squared gradient norm obeys

$$\chi_{\ell+1} \approx \sigma_w^2 \, \mathbb{E}[f'(\mathbf{h}^{\ell+1})^2],$$

where $\mathbf{h}^{\ell+1}$ denotes a typical preactivation at layer $\ell + 1$. The scalar factor $\chi_{\ell+1}$ is the average gradient amplification of layer $\ell+1$: in the ordered phase ($\chi_{\ell+1} < 1$) gradients vanish exponentially with depth, while in the chaotic phase ($\chi_{\ell+1} > 1$) they explode. The critical curve defined by $\chi_{\ell+1} \approx 1$ is called the edge of chaos and separates these two regimes. At this edge, information and gradients can propagate through many layers (Schoenholz et al., 2016). Building on this perspective, (Hayou et al., 2019) refined the EOC analysis and showed that choosing $(\sigma_w^2, \sigma_b^2)$ on the EOC curve leads to better information propagation and faster training in deep networks. Throughout this paper, we refer to Gaussian i.i.d. initializations with $(\sigma_w^2, \sigma_b^2)$ on the EOC curve as EOC initialization.

## 4 METHODOLOGY

We first define the odd–sigmoid function class (Section 4.1). Building on this, we propose a weight initialization for feedforward networks with odd–sigmoid activations (Section 4.2). In Section 4.3, we highlight how the proposed initialization differs from standard Gaussian i.i.d. initializations, and show these differences through theoretical analysis and numerical experiments. The proofs of the theoretical results are provided in Appendix A, and additional numerical results are presented in Appendix B.

## 4.1 Odd-Sigmoid Function Class

In practical neural networks with sigmoidal activations, both forward signals and backpropagated gradients tend to saturate or vanish as the depth increases. Our goal in this work is to move beyond a single sigmoid and consider a broad class of odd–sigmoid activation functions, and to design a weight initialization that prevents such vanishing for any activation in this class. To this end, Section 4.1 introduces the odd–sigmoid function class and establishes its basic properties, which will be used in our subsequent analysis. Formal definitions of the activation functions used in this paper are provided in Appendix B.1.

**Definition 4.1.** *A function $f : \mathbb{R} \to \mathbb{R}$ is an **odd-sigmoid function** if it satisfies the following:*

    *(i)* ***Regularity:*** *$f \in C^1(\mathbb{R})$.*

    *(ii)* ***Odd symmetry:*** *$f(-x) = -f(x)$ for all $x \in \mathbb{R}$.*

    *(iii)* ***Boundedness:*** *$\sup_{x \in \mathbb{R}} |f(x)| < \infty$.*

    *(iv)* ***Strict monotonicity:*** *$f'(x) > 0$ for all $x \in \mathbb{R}$.*

    *(v)* ***Slope decay:*** *$f'$ is strictly decreasing on $[0, \infty)$.*

Denote the class of all odd-sigmoid functions as $\mathcal{F}$. In the following, we establish several basic properties of functions in $\mathcal{F}$.

**Pitchfork Bifurcation.** Recall that $x^* \in \mathbb{R}$ is a fixed point of $f : \mathbb{R} \to \mathbb{R}$ if $f(x^*) = x^*$. For $f \in \mathcal{F}$ define $\omega_f := 1/f'(0) > 0$ and, unless stated otherwise, write $\omega$ simply. Consider $\phi_a(x) := f(ax)$ for $a > 0$. We say that $\phi_a$ has a pitchfork bifurcation at $a = \omega$ if it has exactly one fixed point $\{0\}$ for $0 < a < \omega$ and exactly three fixed points $\{0, \pm\xi_a\}$ for $a > \omega$, with the nonzero points occurring as a symmetric pair. We show that this holds for every $f \in \mathcal{F}$.

**Proposition 4.2.** *Suppose $f \in \mathcal{F}$ with $\omega := 1/f'(0)$, and for a fixed $a > 0$ define $\phi_a(x) := f(ax)$. Then*

    *(i) If $0 < a \leq \omega$, then $\phi_a(x)$ has a unique fixed point $x^* = 0$.*

    *(ii) If $a > \omega$, then $\phi_a(x)$ has three distinct fixed points: $x^* = -\xi_a,\ 0,\ \xi_a$ such that $\xi_a > 0$.*

The following theorem establishes convergence properties of $x_{n+1} = f(ax_n)$ for all $x_0 > 0$.

**Theorem 4.3.** *Suppose $f \in \mathcal{F}$ with $\omega := 1/f'(0)$, and for a fixed $a > 0$ define*

$$x_0 > 0, \qquad x_{n+1} = \phi_a(x_n), \qquad n = 0, 1, 2, \ldots.$$

*Then the sequence $\{x_n\}$ converges for every $x_0 > 0$. Furthermore,*

    *(1) if $0 < a \leq \omega$, then $x_n \to 0$ as $n \to \infty$.*

    *(2) if $a > \omega$, then $x_n \to \xi_a$ as $n \to \infty$.*

According to Theorem 4.3, when $a > \omega$, for any initial value $x_0 > 0$ (resp. $x_0 < 0$), the sequence defined by $x_{n+1} = f(ax_n)$ converges to $\xi_a$ (resp. $-\xi_a$) as $n \to \infty$. From a signal propagation viewpoint in FFNN (Equation 1), this implies that activations do not vanish as depth increases. See Figures 8 and 9 for the convergence of the iterates $x_{n+1} = \phi_a(x_n)$ as a function of $a$ and the initial value $x_0$. Proposition A.2 and Corollary A.3 analyze the iteration with a coefficient sequence $\{a_m\}$ that may vary across steps. They show that excessively large or small $a_m$ drives the dynamics into saturation, highlighting the importance of appropriately scaling $\{a_m\}$ when designing the initialization (Figure 2).

**Corollary 4.4.** *Let $f_1, f_2 \in \mathcal{F}$ and let $c_1, c_2 \geq 0$ with $(c_1, c_2) \neq (0, 0)$. If $g = c_1 f_1 + c_2 f_2$, then $g \in \mathcal{F}$. Furthermore, it holds that*

$$\frac{1}{\omega_g} = \frac{c_1}{\omega_{f_1}} + \frac{c_2}{\omega_{f_2}}.$$

Since $\mathcal{F}$ is closed under addition, more generally, any finite sum $\sum_{j=1}^{M} c_j f_j \in \mathcal{F}$ for all $c_j \geq 0$ with $(c_1, \ldots, c_M) \neq \mathbf{0}$ and $f_j \in \mathcal{F}$.

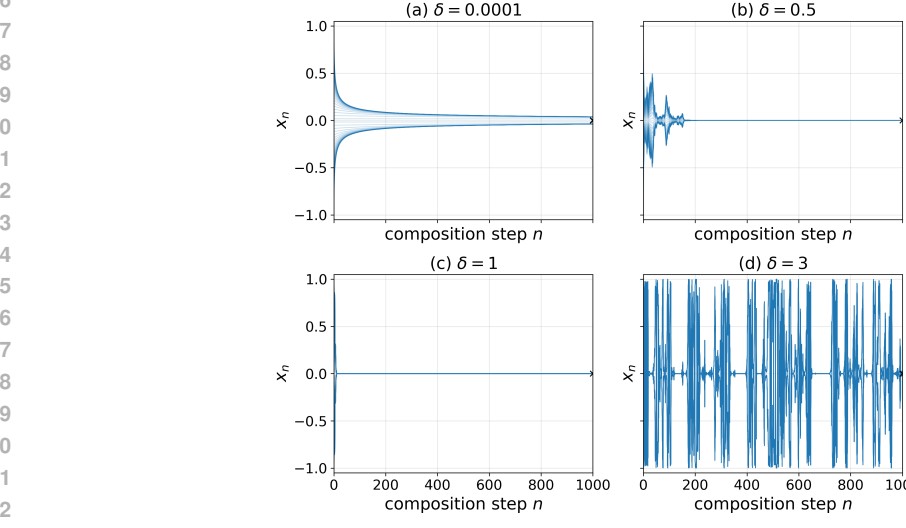

Figure 2: Iterated dynamics of the $\tanh$ activation under randomly varying gains. For each panel, we fix a sequence $(a_1, \ldots, a_{1000})$ sampled i.i.d. from $\mathcal{U}[1-\delta,\, 1+\delta]$ and iterate $x_{n+1} = \tanh(a_{n+1} x_n)$ from 60 initial points $x_0 \in [-1, 1]$. Panels (a)–(d) correspond to $\delta = 0.0001,\, 0.5,\, 1,\, 3$.

### 4.2 PROPOSED WEIGHT INITIALIZATION METHOD

We build on the theoretical results in Section 4.1 to design an initialization scheme that keeps forward signals well dispersed and avoids saturation even in very deep networks. We then compare the proposed initialization with standard Gaussian i.i.d. schemes (e.g., Xavier, He, EOC) from both forward and backward pass perspectives, studying signal and gradient propagation through theoretical analysis and experiments.

**Proposed Weight Initialization**   Consider $f \in \mathcal{F}$ with $\omega := 1/f'(0) > 0$ and a target depth $L$. We initialize each layer as $\mathbf{W}^\ell = \mathbf{D}^\ell + \mathbf{Z}^\ell \in \mathbb{R}^{N_\ell \times N_{\ell-1}}$, where $(\mathbf{D}^\ell)_{ij} = \omega$ if $i \equiv j \pmod{N_{\ell-1}}$ and 0 otherwise. The noise matrix $\mathbf{Z}^\ell$ is sampled with noise scale $\sigma_z$ as

$$(\mathbf{Z}^\ell)_{ij} \sim \mathcal{N}\Big(0,\ \sigma_z^2/N_{\ell-1}\Big).$$

The value of $\sigma_z$ is determined via a calibrated scalar surrogate model. Let $p_{\text{real}} = 0.4$ (see Section B.7) be the desired negative rate at depth $L$ in the full network, and let $p_{\text{sur}}(L)$ denote the surrogate target used in the scalar model. As shown in Appendix B.4, for shallow depths the optimal surrogate target remains close to $p_{\text{real}}$, whereas for larger depths it decays approximately exponentially. In particular, a least-squares fit for $L \geq 10$ yields

$$p_{\text{sur}}(L) \approx 2.05\, e^{-0.133L}.$$

In practice we therefore set

$$p(L) \;:=\; \begin{cases} p_{\text{real}}, & L \leq L_{th}, \\ p_{\text{sur}}(L), & L > L_{th}, \end{cases}$$

with $L_{th} = 10$, and use this $p(L)$ in the closed-form calibration

$$\sigma^*(p(L), L, \omega) \;=\; -\frac{\omega}{\Phi^{-1}\!\left(\dfrac{1 - (1 - 2p(L))^{1/L}}{2}\right)},$$

where $\Phi$ denotes the standard normal cumulative distribution function. We then set $\sigma_z :=\sigma^*(p(L), L, \omega)$.

For the learning rate, we use a band proportional to $\omega$; for Adam (Kingma & Ba, 2014),

$$\eta \;\in\; [\, 10^{-5}\omega,\ 10^{-3}\omega \,].$$

We discuss the learning-rate range $\eta$ and the surrogate calibration in Section 5.2 and Appendix B.4.

**Derivation of the initialization.** Using the elementwise formulation from Section 4.1, the usual matrix–vector product $\mathbf{W}^\ell \mathbf{x}^{\ell-1}$ can be rewritten as

$$x_i^{\ell+1} = f\left(a_i^{\ell+1}\, x_i^\ell\right), \qquad a_i^{\ell+1} = w_{ii}^{\ell+1} + \sum_{j \neq i} \frac{w_{ij}^{\ell+1} x_j^\ell}{x_i^\ell}, \tag{2}$$

so that $a_i^{\ell+1}$ plays the role of an effective gain for neuron $i$ in layer $\ell+1$. Under our proposed initialization, Lemma 4.5 shows that $a_i^{\ell+1}$ is approximately Gaussian with mean $\omega$ and a data-dependent variance.

**Lemma 4.5.** *Using the elementwise formulation in equation 1 and employing the proposed weight initialization, fix an arbitrary layer $\ell$ and index $i$ such that $x_i^\ell \neq 0$. Then, conditionally on $x^\ell$,*

$$a_i^{\ell+1} \;\sim\; \mathcal{N}\!\left(\omega,\; \frac{\sigma_z^2}{N_\ell}\Big(1 + \sum_{j \neq i}\Big(\frac{x_j^\ell}{x_i^\ell}\Big)^2\Big)\right). \tag{3}$$

*Moreover, if $|x_j^\ell| \leq M$ for all $j$ and $|x_i^\ell| \geq \varepsilon > 0$, then*

$$\frac{\sigma_z^2}{N_\ell} \;\leq\; \mathrm{Var}\!\left(a_i^{\ell+1} \mid x^\ell\right) \;\leq\; \sigma_z^2 \frac{M^2}{\varepsilon^2}.$$

The distribution of the effective gain $a_i^\ell$ is crucial for understanding signal propagation. We first analyze its mean, denoted by $\mu_a$. This mean is determined by the diagonal entries of the proposed initialization matrix $\mathbf{D}$. In particular, we are interested in the supercritical regime $\mu_a > \omega$, which is analyzed in Theorem 4.6 below.

**Theorem 4.6.** *Let $f \in \mathcal{F}$ be an odd–sigmoid activation with $\omega := 1/f'(0)$, and fix any $\varepsilon > 0$. Consider the feedforward network and proposed initialization, except that the diagonal element is set to $a_0 := \omega + \varepsilon$, and let $a_i^{\ell+1}$ be the effective gain defined in equation 1. Fix a tolerance $\gamma \in (0,1)$ and a finite depth $L \in \mathbb{N}$. Then there exist a threshold depth $L_0 \leq L$ and a noise threshold $\sigma_0 > 0$ such that, for all $0 < \sigma_z \leq \sigma_0$,*

$$\mathbb{P}\Big((1-\gamma)\,\sigma_z^2 \;\leq\; \mathrm{Var}(a_i^{\ell+1} \mid x^\ell) \;\leq\; (1+\gamma)\,\sigma_z^2 \quad \textit{for all } L_0 \leq \ell < L,\ 1 \leq i \leq N_\ell\Big) \;\geq\; 1-\gamma. \tag{4}$$

The data dependency of the variance term can be characterized as follows. For each layer $\ell$ and neuron $i$ with $x_i^\ell(\sigma_z) \neq 0$, define $R_\ell(\sigma_z; i) := \frac{\|\mathbf{x}^\ell(\sigma_z)\|_2^2}{N_\ell\,(x_i^\ell(\sigma_z))^2}$. Here, $x_i^\ell(\sigma_z)$ denotes the activation at layer $\ell$ under noise scale $\sigma_z$. Theorem 4.6 implies that when the diagonal mean $\mu_a > \omega$, there exist a depth threshold $L_0$ and a noise threshold $\sigma_0 > 0$ such that $R_\ell(\sigma_z; i) \approx 1$ for all $\ell \geq L_0$ and $\sigma_z \leq \sigma_0$. This means that all neurons in deep layers converge to nearly identical values, indicating saturation of activations and loss of data dependence (Figure 8). Similarly, for $\mu_a < \omega$, Theorem 4.6 yields the same qualitative behavior. Deeper networks require smaller initialization variance to remain trainable (Poole et al., 2016; Schoenholz et al., 2016). Motivated by this, we set the diagonal mean to the critical value $a_0 = \omega$ when initializing deep networks.

Proposition A.2 and Corollary A.3 suggest that, for most gains $a_i^\ell$ across layers, if the variance of the gains is either too small or too large, the activations tend to saturate. We define the negative rate at depth $L$ as $\pi_L(\sigma) := \mathbb{P}\big(x_L < 0 \,\big|\, x_0 > 0\big)$, i.e., the probability that a neuron that starts with a positive activation ends up with a negative sign at layer $L$. Under the surrogate model introduced below, it suffices to track a single neuron with $x_i^1 > 0$, since for any $f \in \mathcal{F}$ the activation is sign-preserving and odd. Consequently, the probability that a positive entry becomes negative at layer $L$ is equal to the probability that a negative entry becomes positive, and does not depend on the particular choice of $x_i^1$.

To handle this dependence on the current activations, we approximate the gain distribution by a scalar surrogate model, $a_i^{\ell+1} \sim \mathcal{N}(\omega, \sigma_z^2)$, and estimate the negative rate from this Gaussian approximation. Our initialization has a nonzero mean gain $\omega$, which makes sign changes along a given coordinate well defined across layers. We interpret frequent sign flips during forward propagation

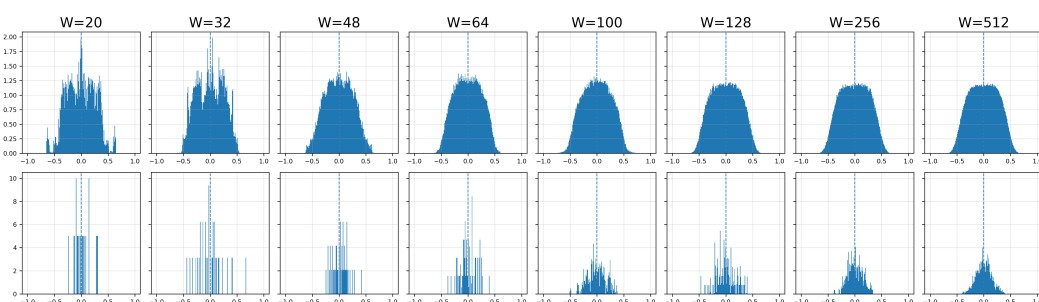

Figure 3: Last layer activation histograms for tanh networks with depth $L = 1000$ and varying width under the proposed initialization **(top row)** and the EOC initialization **(bottom row)**. Each column corresponds to a different hidden width.

as a form of information loss and therefore use the surrogate calibration to control the negative rate at the final layer. In practice, we set the noise level so that the empirical negative rate at depth $L$ is close to $p_{\text{real}} = 0.4$, preserving most sign information while still retaining sufficient randomness for feature learning. The next theorem shows how, given a target negative rate at depth $L$, we can compute the corresponding noise scale $\sigma^*$ in this surrogate model.

**Theorem 4.7.** *Fix a target $p \in [0, \frac{1}{2})$, a depth $\ell \in \mathbb{N}$, and $\omega > 0$. There exists a unique $\sigma^\star = \sigma^\star(p, \ell, \omega) > 0$ such that $\pi_\ell(\sigma^\star) = p$, and it is given by*

$$\sigma^\star(p, \ell, \omega) = -\frac{\omega}{\Phi^{-1}\left(\dfrac{1 - (1 - 2p)^{1/\ell}}{2}\right)}. \tag{5}$$

Figure 10 shows how the negative rate varies with $L$ and $p$. As shown in Figure 11, for the proposed initialization $\chi_\ell$ at the $\sigma^*(p, L, \omega)$ stays within a few percent of 1 for both $p = 0.01$ and $p = 0.49$ over depths up to $2 \times 10^5$, suggesting that the calibration preserves trainable gradient scales. The relationship between the FFNN-level negative rate and our scalar surrogate, as well as the corresponding validation experiments, is detailed in Appendix B.4.

### 4.3    COMPARATIVE ANALYSIS OF GAUSSIAN AND PROPOSED INITIALIZATIONS

Gaussian i.i.d. schemes such as Xavier, He, and EOC all draw weights independently from $\mathcal{N}(0, \sigma_w^2/N_\ell)$, differing only in the choice of $\sigma_w$. In this section, we contrast our method with EOC, examining their forward and backward signal propagation both theoretically and empirically.

**Forward Pass.**    As shown in Figures 19, 20, and 21, our proposed initialization keeps the activations well dispersed even at depth 10,000, whereas the EOC initialization drives them toward saturation near zero. Moreover, Figures 3, 22, and 23 show that, for EOC, signal propagation degrades as the width decreases, reflecting that this initialization is derived under an infinite width (or sufficiently wide) assumption. In contrast, our initialization is obtained by directly controlling the effective gain $a_i^\ell$ and does not rely on any large width approximation, which leads to stable forward propagation even in deep and relatively narrow networks. The theoretical motivation for forward signal propagation with odd-sigmoid activations is discussed in Sections 4.1 and 4.2.

**Backward Pass.**    Although our initialization is derived without mean field assumptions, for the backward pass we adopt the standard mean field framework as an analytical tool to study gradient propagation. Let $\mathbf{g}^\ell = \partial \mathcal{L}/\partial \mathbf{x}^\ell$ denote the gradient at depth $\ell$ for a scalar loss $\mathcal{L}$. Under the assumptions, the backpropagated gradients satisfy the form $\frac{1}{N_\ell}\mathbb{E}\|\mathbf{g}^\ell\|_2^2 \approx \chi_{\ell+1}\frac{1}{N_{\ell+1}}\mathbb{E}\|\mathbf{g}^{\ell+1}\|_2^2$, where $\chi_{\ell+1}$ is the average gradient amplification factor of layer $\ell + 1$. Values $\chi_{\ell+1} \ll 1$ correspond to vanishing gradients, whereas $\chi_{\ell+1} \gg 1$ lead to exploding gradients; thus, keeping $\chi_{\ell+1}$ close to 1 across layers is essential for stable training in very deep networks. For our proposed initialization we show in Appendix A.3 that $\chi_{\ell+1}$ can be written as $\chi_{\ell+1} \approx (\omega^2 + \sigma_z^2)\mathbb{E}\big[f'(\mathbf{h}^{\ell+1})^2\big]$. For

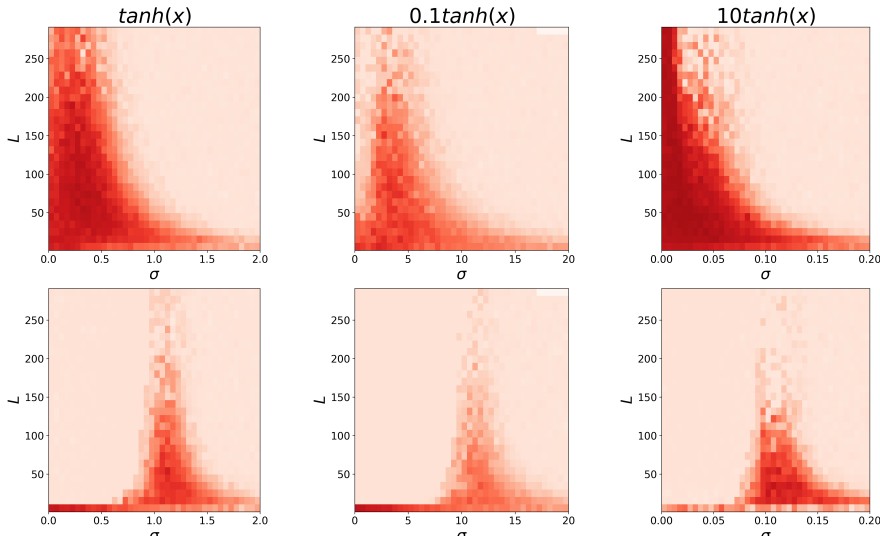

Figure 5: Heatmaps of validation accuracy on MNIST as a function of depth $L$ and noise scale $\sigma$ for three activations $\tanh(x)$, $0.1\tanh(x)$, and $10\tanh(x)$. Each model is a FFNN with width $128$ trained with Adam. Each cell shows the mean validation accuracy over 3 runs. **(Top row)** Proposed. **(Bottom row)** EOC. Darker colors indicate higher validation accuracy.

comparison, under a standard i.i.d. Gaussian initialization, the corresponding factor takes the form $\chi^*_{\ell+1} \approx \sigma_w^2 \, \mathbb{E}[f'(\mathbf{h}^{\ell+1})^2]$.

For our proposed initialization, as $\sigma_z$ grows, $\mathbb{E}[f'(\mathbf{h}^{\ell+1})^2]$ decreases, so the product stays close to 1 over a relatively wide range of $\sigma_z$. In contrast, for standard Gaussian i.i.d. initialization $\chi^*_{\ell+1}(\sigma_w) \approx \sigma_w^2 \, \mathbb{E}[f'(\mathbf{h}^{\ell+1})^2]$ has no analogous $\omega^2$ term and is much more sensitive to perturbations of $\sigma_w$ around its critical value. Therefore, $\chi_{\ell+1}$ remains close to 1 over a wide range of noise scales $\sigma_z$, leading to much more robust training (Figure 24). Consistent with this analysis, Figure 5 indicates that our initialization supports accurate training over a much broader region in $(L, \sigma)$ and across activation scales, whereas Gaussian initializations remain effective only in a narrow band of $\sigma$ and at relatively shallow depths. Figure 4 shows the gradient norm for the quadratic loss $L = \frac{1}{2}\|\mathbf{y}\|_2^2$ as a function of depth. Under our proposed initialization, the gradient norm is effectively preserved over very

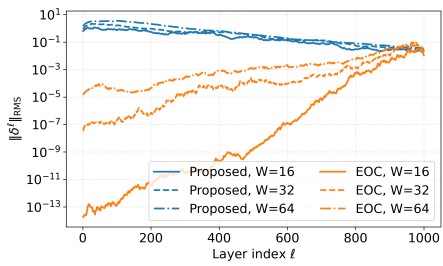

Figure 4: Layerwise gradient norms at initialization for a $\tanh$ FFNN of depth $L = 1000$. We set $\boldsymbol{\delta}^\ell := \partial L / \partial \mathbf{h}^\ell$.

deep networks and remains stable across different widths. In contrast, EOC and Gaussian initializations exhibit rapid gradient decay at moderate widths, consistent with their reliance on infinite width assumptions, whereas our scheme maintains well scaled forward activations and backward gradients even in finite width regimes.

## 5 EXPERIMENTS

In this section, we evaluate the proposed initialization method. Section 5.1 studies widely used activations with $\omega \approx 1$, comparing our method against Gaussian initializations (Xavier, He, EOC) in terms of data efficiency and the dependence on network width and depth. Section 5.2 then turns to activations with $\omega \not\approx 1$, where we analyze performance, the ability to train effectively without batch normalization, learning-rate sensitivity, and the trainability of networks across a wide range

Table 1: Mean of the best validation accuracies within 50 epochs over 10 independent runs for a 50 layer FFNN (512 units) on MNIST and Fashion MNIST, trained on 100 or 500 sample subsets.

| Dataset | Method | $\tanh(x)$ | | $\text{erf}(x)$ | | $\arctan(x)$ | | $\text{gd}(x)$ | | $\text{softsign}_2(x)$ | | $\text{softsign}_1(x) + \text{softsign}_2(x)$ |
|---|---|---|---|---|---|---|---|---|---|---|---|---|
| | | 100 | 500 | 100 | 500 | 100 | 500 | 100 | 500 | 100 | 500 | 100 | 500 |
| MNIST | Xavier | 64.00 | 79.85 | 66.15 | 84.63 | 64.58 | 81.30 | 60.87 | 84.18 | 60.52 | 81.38 | 32.63 | 53.97 |
| | He | 41.65 | 72.55 | 21.05 | 48.37 | 51.40 | 77.68 | 42.50 | 72.75 | 48.35 | 76.18 | 11.35 | 10.02 |
| | EOC | 59.83 | 82.92 | 64.58 | 82.97 | 64.52 | 83.73 | 62.30 | 84.05 | 66.57 | 82.48 | 59.60 | 71.57 |
| | Proposed | **68.23** | **86.75** | **66.53** | **87.13** | **67.63** | **86.82** | **66.38** | **86.23** | **69.51** | **86.43** | **65.65** | **84.02** |
| FMNIST | Xavier | 67.65 | 74.53 | 68.92 | 76.13 | 66.75 | 73.78 | 67.10 | 72.65 | 66.85 | 74.47 | 49.38 | 69.88 |
| | He | 61.13 | 74.60 | 44.60 | 65.25 | 62.92 | 76.75 | 62.88 | 76.55 | 64.97 | 75.22 | 11.50 | 11.70 |
| | EOC | 67.22 | 76.87 | 66.85 | 73.83 | 66.93 | 76.45 | 67.00 | 77.63 | 67.08 | 75.67 | 65.48 | 73.07 |
| | Proposed | **70.67** | **77.43** | **70.63** | **78.17** | **71.55** | **77.92** | **68.62** | **77.80** | **68.17** | **78.33** | **67.93** | **76.43** |

of activation scales, including experiments on physicsinformed neural networks (PINNs). The experimental setting is described in Appendix C.

### 5.1 Experiments with $\omega \approx 1$ Activations

**Dataset Efficiency.** Table 1 presents the best validation accuracy within 50 epochs for a 50 layer FFNN (512 units), trained on small subsets of size 100 or 500. We compare four initializations across odd–sigmoid activations $f \in \mathcal{F}$. The proposed initialization attains the top accuracy in all settings, across both datasets and both sample regimes, with the improvements most pronounced at 100 samples. These results indicate that our method is data efficient, achieving higher validation accuracy with limited data.

**Network Size Independence.** We assess how independent the proposed initialization is of depth and width in networks. Figures 25, 26, 27, and 28 show validation accuracy versus depth for FFNNs (width 64) on MNIST, Fashion MNIST, CIFAR-10, and CIFAR-100, using odd sigmoid activations and four initializations. Each panel fixes one activation and shows the best validation accuracy over 10 epochs for depths $L \in \{20, 50, 100, 150, 200\}$. Across all datasets, activations, and depths, the proposed initialization consistently achieves the highest validation accuracy among all Gaussian initializations. Figures 6 and 32 further examine the effect of width in deep networks and show that standard Gaussian initializations struggle when the network is either too narrow or too wide, whereas the proposed scheme maintains strong performance over a broad range of widths.

### 5.2 Experiments with Activations Far from $\omega \approx 1$

In this section we investigate activations $f, g \in \mathcal{F}$ whose $\omega = 1/f'(0)$ is not close to 1, including rescaled variants $\alpha f(x)$, input scaled variants $f(\alpha x)$, and positive linear combinations $\alpha f(x) + \beta g(x)$ with $\alpha, \beta > 0$.

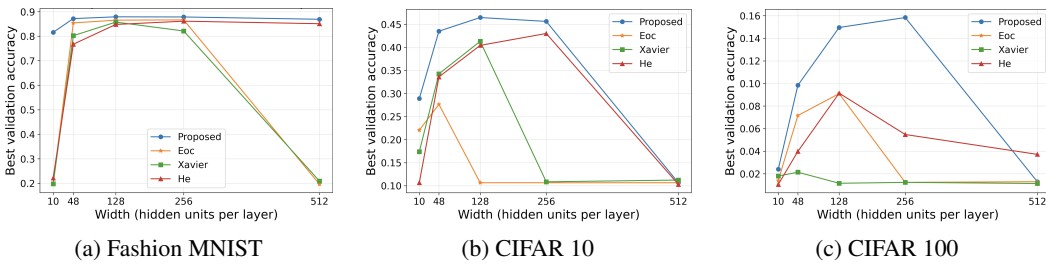

| (a) Fashion MNIST | (b) CIFAR 10 | (c) CIFAR 100 |

Figure 6: Best validation accuracy versus width $\{10, 48, 128, 150, 200, 512\}$ for a 100 layer $\tanh$ FFNN. Each curve shows the Proposed, EOC, Xavier, and He initialization schemes, and each point corresponds to the best validation accuracy over 20 training epochs.

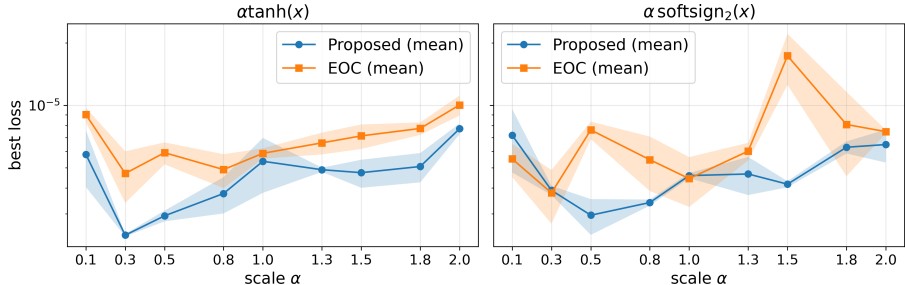

Figure 7: Best PINN loss versus activation scale for the Black–Scholes PINN (depth 50, width 64), comparing the proposed and EOC initializations. **(Left)** $a \tanh(x)$ with $a \in \{0.1, 0.3, 0.5, 0.8, 1.0, 1.3, 1.5, 1.8, 2.0\}$. **(Right)** $b \operatorname{softsign}_2(x)$ with the same set of scales.

**Batch Normalization Free Training.** For

$$f(x) = \tanh(ax) + \operatorname{erf}(bx) + \frac{x}{1 + |cx|} + \operatorname{gd}(dx), \tag{6}$$

by Corollary 4.4 this $f$ is an odd–sigmoid activation. Empirical results for various $(a, b, c, d)$ appear in Figure 29. Across all settings, the proposed initialization achieves the highest validation accuracy even without batch normalization, outperforming Gaussian i.i.d. initializations both with and without batch normalization. We further investigate the effect of applying batch normalization, including training 800 layer networks, in Figures 30 and 31.

**Learnable Learning Rate.** To study optimization stability, we plot learning rate versus validation accuracy curves on MNIST and Fashion MNIST for 20 layer, width 512 FFNNs with activations $f(x) = \frac{2}{\pi} \arctan(ax)$ and $f(x) = \tanh(\alpha x)$, using scales $\alpha \in \{10^2, 10^1, 1, 10^{-1}, 10^{-2}, 10^{-3}\}$ (Figures 33, 34, 35, and 36). The proposed method consistently yields a learnable, typically wider LR window across a broad range of $\alpha$, demonstrating that training remains feasible across diverse $\omega$ scales. If the learning rate is chosen too large or too small relative to the $\omega$-scaled band, the first parameter update either destroys the calibrated noise spread or becomes negligible, so that training effectively fails from the second pass onward. Motivated by this, we use the practical learning rate range

$$\eta \in [\, 10^{-5}\,\omega,\; 10^{-3}\,\omega\,].$$

**Scale Preserving Odd–Sigmoid Activations.** For $f \in \mathcal{F}$, we investigate whether the proposed and EOC initializations preserve the activation range of the scaled activation $\alpha f$ (Figure 37). Unlike EOC, the proposed initialization maintains an activation range proportional to $\alpha$ even as the network depth increases. Networks with $\alpha \tanh$, $\alpha \arctan$, and $\alpha \operatorname{softsign}_2$ activations are trained on MNIST and Fashion MNIST for $\alpha$ ranging from $10^{-2}$ to $10^9$, and the proposed initialization enables stable training across all $\alpha$ scales for every activation (Table 2–7). Since an $\alpha$ scaled activation directly controls the scale of the output $y_i$, we also evaluate our initialization on regression problems using physics informed neural networks (PINNs). As shown in Figures 7 and 38, these results indicate that scaled odd–sigmoid activations are well suited for regression tasks, where controlling the output range is important. We defer detailed PINN setups and PDE formulations to Appendix C.4

## 6 CONCLUSION

We introduced an activation aware initialization scheme for FFNNs with odd–sigmoid activations that does not rely on infinite or very wide width assumptions. Unlike standard Gaussian i.i.d. initializations, the proposed method is more robust to variance choice and maintains stable signal and gradient propagation even in very deep and relatively narrow networks. In physics informed neural networks, we further observed that appropriately scaled odd–sigmoid activations, combined with our initialization, can achieve lower PINN losses than standard choices. These results suggest that, with a suitable initialization, scaled odd–sigmoid activations can be used as practical design knobs to improve performance beyond what is possible with conventional Gaussian initializations.

ETHICS STATEMENT

This research adheres to the ICLR Code of Ethics. A large language model was used only to refine the writing and assist in preparing visualizations; all core ideas, methods, and results were developed independently by the authors. We adhere to the highest standards of research integrity and transparency as set out in the ICLR guidelines.

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

SUPPLEMENTARY MATERIAL

The supplementary material is structured as follows.

- Appendix A provides the theoretical results and their proofs.
- Appendix B contains additional simulation results without neural networks.
- Appendix C reports extra experimental details and results with neural networks.

# A    THEORETICAL RESULTS

In this section, we provide proofs for the statements presented in Section 4.

## A.1    THEORETICAL RESULTS FOR SECTION 4.1

**Lemma A.1.** *If $f \in \mathcal{F}$, then the following holds.*

*(i)* $f(0) = 0$.

*(ii)* $0 = \arg\max |f'(x)|$.

*(iii)* $\lim_{x \to \infty} f'(x) = 0$.

*Proof.* (i) It is trivial that odd symmetry implies $f(0) = 0$. (ii) Since $f'$ is an even function, by Definition 4.1(v) $f'(x)$ has the maximum value at $x = 0$. (iii) Since $f'$ is strictly decreasing on $[0, \infty)$ and $f'(x) > 0$ for all $x \in \mathbb{R}$, the Monotone Convergence Theorem for real functions implies that $\ell := \lim_{x \to \infty} f'(x)$ exists for some $\ell \in [0, \infty)$. Suppose that $\ell > 0$. Then there exists $R > 0$ such that

$$f'(x) \geq \frac{\ell}{2} \quad \text{for all } x \geq R.$$

Integrating from $R$ to $x$, we obtain

$$f(x) - f(R) = \int_R^x f'(t)\, dt \ \geq \ \frac{\ell}{2}(x - R) \quad \text{for all } x > R.$$

Hence $f(x) \to \infty$ as $x \to \infty$, which contradicts the boundedness of $f$. Therefore $\ell = 0$. $\square$

Lemma A.1 summarizes the basic regularity and saturation properties of odd-sigmoid activations in $\mathcal{F}$. In particular, it shows that every $f \in \mathcal{F}$ has a unique global slope maximum at the origin and becomes arbitrarily flat in the tails. These simple but structural features will be crucial for understanding how the gain parameter $a$ reshapes the fixed-point structure of the scalar map $x \mapsto f(ax)$.

**Proposition 4.2**    Suppose $f \in \mathcal{F}$ with $\omega := 1/f'(0)$, and for a fixed $a > 0$ define $\phi_a(x) := f(ax)$. Then

*(i)* If $0 < a \leq \omega$, then $f(ax)$ has a unique fixed point $x^* = 0$.

*(ii)* If $a > \omega$, then $f(ax)$ has three distinct fixed points: $x^* = -\xi_a,\ 0,\ \xi_a$ such that $\xi_a > 0$.

*Proof.* For $a > 0$, consider $g(x, a) := f(ax) - x$. We have $g(0, a) = 0$ and $g'(x, a) = a\, f'(ax) - 1$.

Case (i): $0 < a \leq \omega$. For $x > 0$ we have $ax > 0$, and $f'(ax) < f'(0)$; hence

$$g'(x, a) = a\, f'(ax) - 1 < a\, f'(0) - 1 \ \leq \ 0,$$

for all $x > 0$. Thus $g(\cdot, a)$ is strictly decreasing on $(0, \infty)$, $g(0, a) = 0$, and $\lim_{x \to \infty} g(x, a) = L - x = -\infty$, so $g(x, a) < 0$ for all $x > 0$ and there is no positive root. Since $g(\cdot, a)$ is odd, there is no negative root either. Hence $x = 0$ is the unique solution.

Case (ii): $a > \omega$. Then $g'(0,a) = af'(0) - 1 > 0$. $g'(\cdot, a)$ is strictly decreasing on $[0,\infty)$, and by the Lemma A.1

$$\lim_{x\to\infty} g'(x,a) = a \lim_{x\to\infty} f'(ax) - 1 = -1 < 0.$$

By the intermediate value theorem, there exists a unique $\hat{x} > 0$ with $g'(\hat{x}, a) = 0$. Hence $g(\cdot, a)$ is strictly increasing on $[0, \hat{x}]$ and strictly decreasing on $[\hat{x}, \infty)$. Since $g(0,a) = 0$ and $g'(0,a) > 0$, we have $g(x,a) > 0$ for all $0 < x \le \hat{x}$. Using (3), $\lim_{x\to\infty} g(x,a) = -\infty$; because $g$ is strictly decreasing on $[\hat{x}, \infty)$ and continuous, there exists a unique $\xi_a > \hat{x}$ with $g(\xi_a, a) = 0$. Thus, there is exactly one positive, nonzero root. Oddness of $g$ yields the symmetric negative root $-\xi_a$, so the full set of real solutions is $\{-\xi_a, 0, \xi_a\}$. $\qquad\square$

Proposition 4.2 provides a precise characterization of how the fixed points of the scalar map $x \mapsto f(ax)$ bifurcate as the gain $a$ crosses the critical value $\omega = 1/f'(0)$. For $a \le \omega$ the origin is the only fixed point, while for $a > \omega$ a symmetric nonzero pair $\pm\xi_a$ emerges, exhibiting a classical pitchfork structure. We next lift this static picture to the dynamical setting, showing that the iterates of $x_{n+1} = f(ax_n)$ converge to the corresponding fixed point for every positive initial condition.

**Theorem 4.3**  Suppose $f \in \mathcal{F}$ with $\omega := 1/f'(0)$, and for a fixed $a > 0$ define

$$x_0 > 0, \qquad x_{n+1} = \phi_a(x_n), \qquad n = 0, 1, 2, \ldots.$$

Then the sequence $\{x_n\}$ converges for every $x_0 > 0$. Furthermore,

(1) if $0 < a \le \omega$, then $x_n \to 0$ as $n \to \infty$.

(2) if $a > \omega$, then $x_n \to \xi_a$ as $n \to \infty$.

*Proof.* (1) Since $f$ is odd and $f'$ is strictly decreasing on $[0,\infty)$, we have $0 < f'(x) < f'(0)$ for all $x \ne 0$; hence, for any $a \in (0, \omega)$ and any $x_n > 0$, it follows that $x_{n+1} = f(ax_n) < x_n$ for all $n \in \mathbb{N}$. By the monotone convergence theorem, it converges to the fixed point $x^* = 0$.
(2) Let $x_0 < \xi_a$. Since $\phi'(x)$ is decreasing for $x \ge 0$, with $\xi_a$ is the unique fixed point for $x > 0$, it holds that $x_n < x_{n+1} < \xi_a$ for all $n \in \mathbb{N}$. Thus, by the monotone convergence theorem, the sequence converges to the fixed point $\xi_a$. The proof is similar when $x_0 > \xi_a$. By the monotone convergence theorem, the sequence also converges to the fixed point $\xi_a$. $\qquad\square$

**Corollary 4.4**  Let $f_1, f_2 \in \mathcal{F}$ and let $c_1, c_2 \ge 0$ with $(c_1, c_2) \ne (0,0)$. If $g = c_1 f_1 + c_2 f_2$, then $g \in \mathcal{F}$. Furthermore, it holds that

$$\frac{1}{\omega_g} = \frac{c_1}{\omega_{f_1}} + \frac{c_2}{\omega_{f_2}}.$$

*Proof.* Let $g = c_1 f_1 + c_2 f_2$. Since $c_1, c_2 \ge 0$, the linear combination preserves oddness and bounded saturation by linearity. Strict monotonicity on $\mathbb{R}$ follows from $g'(x) = c_1 f_1'(x) + c_2 f_2'(x) > 0$, and slope decay on $[0, \infty)$ is preserved because a positive linear combination of strictly decreasing functions is strictly decreasing. Thus $g \in \mathcal{F}$. Evaluating at the origin gives

$$g'(0) = c_1 f_1'(0) + c_2 f_2'(0) = \frac{c_1}{\omega_{f_1}} + \frac{c_2}{\omega_{f_2}},$$

so by definition $1/\omega_g = g'(0)$, which yields the claimed identity. $\qquad\square$

In practice this means that we can build richer odd-sigmoid activations by mixing simpler ones without losing the pitchfork structure described above. We now move from the constant–gain setting to the more general case where the layerwise gains $(a_n)$ are allowed to vary with depth.

**Proposition A.2.** *Let $f \in \mathcal{F}$ and $\{a_n\}_{n=1}^{\infty}$ be a positive real sequence, i.e., $a_n > 0$ for all $n \in \mathbb{N}$, such that only finitely many elements are greater than $\omega = 1/f'(0)$. For a positive sequence $\{a_n\}_{n \geq 1}$, set*

$$\Phi^m := \phi_{a_m} \circ \phi_{a_{m-1}} \circ \cdots \circ \phi_{a_1}.$$

*Then for any $x \in \mathbb{R}$*

$$\lim_{m \to \infty} \Phi^m(x) = 0.$$

*Proof.* Let $N := \max\{n : a_n > \omega\}$ (take $N = 0$ if the set is empty) and define

$$b_n = \begin{cases} a_n, & n \leq N, \\ 0, & n > N, \end{cases} \qquad c_n = \begin{cases} a_n, & n \leq N, \\ \omega, & n > N. \end{cases}$$

Let $\hat{\Phi}^m := \phi_{b_m} \circ \cdots \circ \phi_{b_1}$ and $\tilde{\Phi}^m := \phi_{c_m} \circ \cdots \circ \phi_{c_1}$. For $x \geq 0$ and $n > N$, by definition of $\mathcal{F}$, we obtain

$$\hat{\Phi}^m(x) \ \leq \ \Phi^m(x) \ \leq \ \tilde{\Phi}^m(x).$$

Oddness yields the same in absolute value for all $x \in \mathbb{R}$:

$$|\hat{\Phi}^m(x)| \ \leq \ |\Phi^m(x)| \ \leq \ |\tilde{\Phi}^m(x)|.$$

By Proposition 4.3, $\hat{\Phi}^m(x) \to 0$ and $\tilde{\Phi}^m(x) \to 0$. The squeeze theorem gives $\Phi^m(x) \to 0$. $\qquad\square$

**Corollary A.3.** *Let $\epsilon > 0$ be given and set $\omega := 1/f'(0)$. Suppose that $\{a_n\}_{n=1}^{\infty}$ be a positive real sequence such that only finitely many elements are lower than $\omega + \epsilon$. Then for any $x \in \mathbb{R} \setminus \{0\}$*

$$\liminf_{m \to \infty} |\Phi^m(x)| \ \geq \ \xi_{\omega+\epsilon}.$$

*Proof.* Let $N := \max\{n : a_n < \omega + \epsilon\}$ (take $N = 0$ if the set is empty) and define

$$b_n = \begin{cases} a_n, & n \leq N, \\ \omega + \epsilon, & n > N. \end{cases} \qquad \hat{\Phi}_m := \phi_{b_m} \circ \cdots \circ \phi_{b_1}, \qquad \Phi^m := \phi_{a_m} \circ \cdots \circ \phi_{a_1},$$

where $\phi_a(x) := f(ax)$. By definition of $\mathcal{F}$,

$$|\hat{\Phi}^m(x)| \ \leq \ |\Phi^m(x)| \qquad (\forall x \in \mathbb{R}, \ \forall m).$$

Taking $\liminf$ in the inequality yields

$$\liminf_{m \to \infty} |\Phi^m(x)| \ \geq \ \xi_{\omega+\epsilon}.$$

$\qquad\square$

Proposition A.2 and Corollary A.3 together describe two opposite extremes of how layerwise gains affect the one-dimensional dynamics. Roughly speaking, Proposition A.2 says that if, after some finite depth, all gains $a_n$ stay below the critical value $\omega$, then the composed map $\Phi^m$ always drives the signal back to zero, no matter what happened in the earlier layers. In contrast, Corollary A.3 shows that if the gains are eventually bounded away from $\omega$ by a fixed margin $\varepsilon > 0$, then the compositions cannot collapse to zero: for any nonzero input, the iterates stay at least as large (in absolute value) as the positive fixed point $\xi_{\omega+\varepsilon}$.

## A.2 THEORETICAL RESULTS FOR SECTION 4.2

**Lemma 4.5** Using the elementwise formulation in equation 1 and employing the proposed weight initialization, fix an arbitrary layer $\ell$ and index $i$ such that $x_i^\ell \neq 0$. Then, conditionally on $x^\ell$,

$$a_i^{\ell+1} \sim \mathcal{N}\left(\omega, \frac{\sigma_z^2}{N_\ell}\left(1 + \sum_{j \neq i}\left(\frac{x_j^\ell}{x_i^\ell}\right)^2\right)\right).$$

Moreover, if $|x_j^\ell| \leq M$ for all $j$ and $|x_i^\ell| \geq \varepsilon > 0$, then

$$\frac{\sigma_z^2}{N_\ell} \leq \operatorname{Var}\left(a_i^{\ell+1} \mid x^\ell\right) \leq \sigma_z^2 \frac{M^2}{\varepsilon^2}.$$

*Proof.* From equation 1 and the proposed initialization, we write

$$W^{\ell+1} = D^{\ell+1} + Z^{\ell+1},$$

where the diagonal of $D^{\ell+1}$ equals $\omega$ and $Z^{\ell+1}$ has independent entries $(Z^{\ell+1})_{ij} \sim \mathcal{N}(0, \sigma_z^2/N_\ell)$. The pre-activation at coordinate $i$ reads

$$s_i^{\ell+1} = \sum_{j=1}^{N_\ell}\left((D^{\ell+1})_{ij} + (Z^{\ell+1})_{ij}\right)x_j^\ell = \omega\, x_i^\ell + (Z^{\ell+1})_{ii}\, x_i^\ell + \sum_{j \neq i}(Z^{\ell+1})_{ij}\, x_j^\ell.$$

On the event $x_i^\ell \neq 0$, we define the effective gain

$$a_i^{\ell+1} := \frac{s_i^{\ell+1}}{x_i^\ell} = \omega + (Z^{\ell+1})_{ii} + \sum_{j \neq i}(Z^{\ell+1})_{ij}\frac{x_j^\ell}{x_i^\ell}.$$

Conditionally on $x^\ell$, the coefficients $\{x_j^\ell/x_i^\ell\}$ are deterministic, whereas the random variables $\{(Z^{\ell+1})_{ij}\}_{j=1}^{N_\ell}$ are independent, centered Gaussians with variance $\sigma_z^2/N_\ell$. Therefore $a_i^{\ell+1} \mid x^\ell$ is a linear combination of independent Gaussians, hence Gaussian:

$$a_i^{\ell+1} \mid x^\ell \sim \mathcal{N}\left(\omega, \frac{\sigma_z^2}{N_\ell}\left(1 + \sum_{j \neq i}\left(\frac{x_j^\ell}{x_i^\ell}\right)^2\right)\right).$$

This yields the conditional variance

$$\operatorname{Var}\left(a_i^{\ell+1} \mid x^\ell\right) = \frac{\sigma_z^2}{N_\ell}\left(1 + \sum_{j \neq i}\left(\frac{x_j^\ell}{x_i^\ell}\right)^2\right).$$

Since the summation term is nonnegative, we immediately obtain the conditional lower bound $\operatorname{Var}(a_i^{\ell+1} \mid x^\ell) \geq \sigma_z^2/N_\ell$ and hence the unconditional lower bound $\operatorname{Var}(a_i^{\ell+1}) \geq \sigma_z^2/N_\ell$.

For the upper bound, note that

$$1 + \sum_{j \neq i}\left(\frac{x_j^\ell}{x_i^\ell}\right)^2 = \frac{(x_i^\ell)^2 + \sum_{j \neq i}(x_j^\ell)^2}{(x_i^\ell)^2} = \frac{\|x^\ell\|_2^2}{(x_i^\ell)^2},$$

whence

$$\operatorname{Var}\left(a_i^{\ell+1} \mid x^\ell\right) = \frac{\sigma_z^2}{N_\ell}\frac{\|x^\ell\|_2^2}{(x_i^\ell)^2}.$$

If $|x_j^\ell| \leq M$ for all $j$, then $\|x^\ell\|_2^2 = \sum_{j=1}^{N_\ell}(x_j^\ell)^2 \leq N_\ell M^2$, and if $|x_i^\ell| \geq \varepsilon > 0$, we obtain

$$\frac{\|x^\ell\|_2^2}{(x_i^\ell)^2} \leq \frac{N_\ell M^2}{\varepsilon^2},$$

which implies

$$\operatorname{Var}\left(a_i^{\ell+1} \mid x^\ell\right) \leq \frac{\sigma_z^2}{N_\ell}\frac{N_\ell M^2}{\varepsilon^2} = \sigma_z^2\frac{M^2}{\varepsilon^2}.$$

The unconditional upper bound follows from $\operatorname{Var}(a_i^{\ell+1}) = \mathbb{E}[\operatorname{Var}(a_i^{\ell+1} \mid x^\ell)] + \operatorname{Var}(\mathbb{E}[a_i^{\ell+1} \mid x^\ell])$, and the fact that $\mathbb{E}[a_i^{\ell+1} \mid x^\ell] = \omega$ is constant and the first term is bounded by the conditional upper bound. $\square$

Lemma 4.5 shows that, under our structured initialization, the effective gain $a_i^{\ell+1}$ is approximately Gaussian with mean $\omega$ and a variance that scales like $\sigma_z^2$ times a data dependent factor. In particular, the lower bound $\mathrm{Var}(a_i^{\ell+1} \mid x^\ell) \geq \sigma_z^2/N_\ell$ guarantees a nontrivial amount of gain noise at every layer, while the upper bound prevents the variance from blowing up when the activations remain bounded away from zero. These properties will be crucial when we study how the gain variance behaves as depth and noise scale vary.

**Theorem 4.6** Let $f \in \mathcal{F}$ be an odd–sigmoid activation with $\omega := 1/f'(0)$, and fix any $\varepsilon > 0$. Consider the feedforward network and proposed initialization, except that the diagonal element is set to $a_0 := \omega + \varepsilon$, and let $a_i^{\ell+1}$ be the effective gain defined in equation 1. Fix a tolerance $\gamma \in (0,1)$ and a finite depth $L \in \mathbb{N}$. Then there exist a threshold depth $L_0 \leq L$ and a noise threshold $\sigma_0 > 0$ such that, for all $0 < \sigma_z \leq \sigma_0$,

$$\mathbb{P}\Big((1-\gamma)\,\sigma_z^2 \ \leq \ \mathrm{Var}(a_i^{\ell+1} \mid x^\ell) \ \leq \ (1+\gamma)\,\sigma_z^2 \quad \text{for all } L_0 \leq \ell < L,\ 1 \leq i \leq N_\ell\Big) \ \geq \ 1-\gamma. \quad (7)$$

*Proof.* We first consider $\sigma_z = 0$. In this case we set $\mathbf{W}^\ell = \mathbf{D}^\ell \in \mathbb{R}^{N_\ell \times N_{\ell-1}}$ with $(\mathbf{D}^\ell)_{ij} = a_0$ if $i \equiv j \pmod{N_{\ell-1}}$ and $0$ otherwise, so that the layerwise update reduces to

$$x^{\ell+1} = f(a_0 x^\ell).$$

Hence each coordinate $x_i^\ell$ evolves independently according to the scalar recurrence $x_{n+1} = f(a_0 x_n)$. Since $a_0 f'(0) > 1$, Theorem 4.3 implies that this map has three fixed points $\{0, \pm\xi_{a_0}\}$ and, for any $x_0 \neq 0$,

$$x_n \to \pm\xi_{a_0} \quad \text{as } n \to \infty,$$

with the sign determined by $\mathrm{sign}(x_0)$. For every $\delta > 0$ and every initial value $x_0 \neq 0$ there exists an integer $N_i(\delta)$ such that

$$|x_n(x_0)| \in [\xi_{a_0} - \delta,\ \xi_{a_0} + \delta] \qquad \text{for all } n \geq N_i(\delta).$$

Since each layer contains only finitely many neurons, we can take

$$L_0(\delta) \ := \ \max_{1 \leq i \leq N_\ell} N_i(\delta),$$

so that

$$|x_i^\ell(0)| \ \in \ [\xi_{a_0} - \delta,\ \xi_{a_0} + \delta] \qquad \text{for all } 1 \leq i \leq N_\ell,\ \ell \geq L_0(\delta), \quad (8)$$

where $x^\ell(0)$ denotes the activations at depth $\ell$ in the noiseless case $\sigma_z = 0$.

From these bounds we obtain, for all $\ell \geq L_0(\delta)$,

$$N_\ell(\xi_{a_0} - \delta)^2 \ \leq \ \|x^\ell(0)\|_2^2 \ \leq \ N_\ell(\xi_{a_0} + \delta)^2, \qquad (x_i^\ell(0))^2 \in [(\xi_{a_0} - \delta)^2, (\xi_{a_0} + \delta)^2].$$

Therefore the deterministic ratio defined in Lemma 4.5

$$R_\ell(0; i) := \frac{\|x^\ell(0)\|_2^2}{N_\ell (x_i^\ell(0))^2}$$

satisfies

$$R_\ell(0; i) \in \left[\frac{(\xi_{a_0} - \delta)^2}{(\xi_{a_0} + \delta)^2},\ \frac{(\xi_{a_0} + \delta)^2}{(\xi_{a_0} - \delta)^2}\right] \qquad \text{for all } \ell \geq L_0(\delta), i.$$

As $\delta \to 0$, this interval shrinks to 1. Hence, given any $\gamma \in (0,1)$ we can choose $\delta > 0$ such that

$$\big|R_\ell(0; i) - 1\big| \leq \frac{\gamma}{2} \quad \text{for all } \ell \geq L_0(\delta),\ i.$$

Now we consider $\sigma_z > 0$. Recall that in our initialization we write

$$\mathbf{Z}^{(\ell)} = \frac{\sigma_z}{\sqrt{N_{\ell-1}}} \mathbf{G}^{(\ell)}, \qquad (\mathbf{G}^{(\ell)})_{ij} \sim \mathcal{N}(0,1) \text{ i.i.d.,}$$

so that the randomness is entirely carried by the Gaussian matrices $\mathbf{G}^{(1)}, \ldots, \mathbf{G}^{(L)}$. For any deterministic family of matrices $\widehat{\mathbf{G}}^{(1)}, \ldots, \widehat{\mathbf{G}}^{(L)}$ we define, for $\sigma_z \geq 0$, the corresponding activations $\widehat{\mathbf{x}}^\ell(\sigma_z)$ recursively by

$$\widehat{\mathbf{x}}^0(\sigma_z) := \mathbf{x}^0,$$

$$\widehat{\mathbf{x}}^\ell(\sigma_z) := f\Big(\big(a_0 \mathbf{D}^\ell + \tfrac{\sigma_z}{\sqrt{N_{\ell-1}}} \widehat{\mathbf{G}}^{(\ell)}\big) \widehat{\mathbf{x}}^{\ell-1}(\sigma_z)\Big), \qquad \ell = 1, 2, \ldots, L,$$

where $f$ is applied coordinatewise. For each fixed choice of $(\widehat{\mathbf{G}}^{(1)}, \ldots, \widehat{\mathbf{G}}^{(L)})$, each layer $\ell$ and each coordinate $i$, this defines a map

$$\sigma_z \longmapsto \widehat{x}_i^\ell(\sigma_z).$$

Since $f$ is continuous and, for fixed $\widehat{\mathbf{G}}^{(\ell)}$, the map $\sigma_z \mapsto \big(a_0 \mathbf{D}^{(\ell)} + \tfrac{\sigma_z}{\sqrt{N_{\ell-1}}} \widehat{\mathbf{G}}^{(\ell)}\big) \widehat{\mathbf{x}}^{\ell-1}(\sigma_z)$ is affine in $\sigma_z$, it follows by induction on $\ell$ that

$$\sigma_z \longmapsto \widehat{x}_i^\ell(\sigma_z) \quad \text{is continuous on } [0, \sigma_1] \text{ for any finite } \sigma_1 > 0. \tag{9}$$

For each $\ell$ and $i$ with $x_i^\ell(\sigma_z) \neq 0$, define

$$R_\ell(\sigma_z; i) := \frac{\|\mathbf{x}^\ell(\sigma_z)\|_2^2}{N_\ell \, (x_i^\ell(\sigma_z))^2}, \qquad R_\ell(0; i) := \frac{\|\mathbf{x}^\ell(0)\|_2^2}{N_\ell \, (x_i^\ell(0))^2},$$

so that $R_\ell(0; i)$ is exactly the deterministic ratio considered above. On the event that $x_i^\ell(\sigma_z) \neq 0$ for all $\ell \leq L$ and all $0 \leq \sigma_z \leq \sigma_1$, equation 9 implies that $\sigma_z \mapsto R_\ell(\sigma_z; i)$ is continuous. Hence, for each fixed pair $(\ell, i)$ and any $\gamma \in (0, 1)$ there exists $\sigma_0(\ell, i) > 0$ such that

$$\big|R_\ell(\sigma_z; i) - R_\ell(0; i)\big| \leq \frac{\gamma}{2} \qquad \text{for all } 0 \leq \sigma_z \leq \sigma_0(\ell, i). \tag{10}$$

Combining equation 10 with the deterministic bound $\big|R_\ell(0; i) - 1\big| \leq \gamma/2$ (valid for all $\ell \geq L_0(\delta)$ from equation 8), we obtain

$$\big|R_\ell(\sigma_z; i) - 1\big| \leq \big|R_\ell(\sigma_z; i) - R_\ell(0; i)\big| + \big|R_\ell(0; i) - 1\big| \leq \gamma$$

for all $0 \leq \sigma_z \leq \sigma_0(\ell, i)$ and all $\ell \geq L_0(\delta)$.

Since we only consider a finite set of layers $\ell < L$ and indices $1 \leq i \leq N_\ell$, we may define, for each fixed $(\widehat{\mathbf{G}}^{(1)}, \ldots, \widehat{\mathbf{G}}^{(L)})$,

$$\sigma_0(\widehat{\mathbf{G}}^{(1)}, \ldots, \widehat{\mathbf{G}}^{(L)}) := \min_{L_0(\delta) \leq \ell < L} \min_{1 \leq i \leq N_\ell} \sigma_0(\ell, i) > 0,$$

so that the bound

$$\big|R_\ell(\sigma_z; i) - 1\big| \leq \gamma \tag{11}$$

holds simultaneously for all $L_0(\delta) \leq \ell < L$ and all $1 \leq i \leq N_\ell$ whenever $0 \leq \sigma_z \leq \sigma_0(\widehat{\mathbf{G}}^{(1)}, \ldots, \widehat{\mathbf{G}}^{(L)})$.

Now regard $(\mathbf{G}^{(1)}, \ldots, \mathbf{G}^{(L)})$ as random Gaussian matrices and set

$$S := \sigma_0(\mathbf{G}^{(1)}, \ldots, \mathbf{G}^{(L)}).$$

By the construction above we have $S > 0$ almost surely. Hence, for a given $\gamma \in (0, 1)$ we can choose a deterministic constant $\sigma_0 > 0$ such that $\mathbb{P}(S \geq \sigma_0) \geq 1 - \gamma$ (for example, take $\sigma_0$ to be the $(1 - \gamma)$–quantile of $S$). On the event $\{S \geq \sigma_0\}$ the estimate equation 11 therefore holds for all $0 \leq \sigma_z \leq \sigma_0$, all $L_0(\delta) \leq \ell < L$ and all $1 \leq i \leq N_\ell$.

Finally, Lemma 4.5 (with $a_0$ in place of $\omega$) gives the conditional variance formula

$$\text{Var}(a_i^{\ell+1} \mid x^\ell) = \frac{\sigma_z^2}{N_\ell} \sum_{j=1}^{N_\ell} \Big(\frac{x_j^\ell(\sigma_z)}{x_i^\ell(\sigma_z)}\Big)^2 = \sigma_z^2 \, R_\ell(\sigma_z; i).$$

Combining this with equation 11 and the choice of $\sigma_0$ yields, for all $0 < \sigma_z \leq \sigma_0$ and all $L_0(\delta) \leq \ell < L$, $1 \leq i \leq N_\ell$,

$$(1 - \gamma)\, \sigma_z^2 \leq \text{Var}(a_i^{\ell+1} \mid x^\ell) \leq (1 + \gamma)\, \sigma_z^2,$$

on an event of probability at least $1 - \gamma$. $\qquad\qquad\qquad\qquad\qquad\qquad\qquad\qquad\qquad\qquad\quad\square$

The previous results describe how the magnitude of the effective gains behaves across layers (Figures 8 and 10). To control the sign dynamics, we now turn to a scalar surrogate model in which the gains are i.i.d. Gaussian with mean $\omega$ and variance $\sigma^2$. In this simplified setting, the only quantity that matters is the probability that the scalar iterate becomes negative at a given depth. The next lemma provides a closed-form recursion for this negative rate.

**Lemma A.4.** *Let $f \in \mathcal{F}$ and $x_0 > 0$, for every $j \geq 1$,*

$$\pi_j = \tfrac{1}{2}\Big(1 - (1 - 2p_-)^j\Big), \qquad p_- = \mathbb{P}(A_1 < 0) = \Phi\Big(-\frac{k}{\sigma}\Big). \tag{12}$$

*Proof.* Since $f$ is odd and strictly increasing, $\mathrm{sign}(f(u)) = \mathrm{sign}(u)$ for all $u$, hence

$$\{X_j < 0\} = \{A_j X_{j-1} < 0\} = \big(\{A_j < 0\} \cap \{X_{j-1} > 0\}\big) \cup \big(\{A_j > 0\} \cap \{X_{j-1} < 0\}\big),$$

up to null sets (because $\mathbb{P}(A_j = 0) = 0$ for Gaussian $A_j$). Independence of $A_j$ and $X_{j-1}$ yields

$$\pi_j = \mathbb{P}(A_j < 0)\,\mathbb{P}(X_{j-1} > 0) + \mathbb{P}(A_j > 0)\,\mathbb{P}(X_{j-1} < 0) = p_-(1 - \pi_{j-1}) + (1 - p_-)\pi_{j-1}.$$

Thus $\pi_j = (1 - 2p_-)\pi_{j-1} + p_-$. With $\pi_0 = \mathbb{P}(X_0 < 0) = 0$, the first-order linear recursion solves to

$$\pi_j - \tfrac{1}{2} = (1 - 2p_-)\big(\pi_{j-1} - \tfrac{1}{2}\big) \implies \pi_j - \tfrac{1}{2} = (1 - 2p_-)^j\big(\pi_0 - \tfrac{1}{2}\big) = -\tfrac{1}{2}(1 - 2p_-)^j.$$

The value of $p_-$ follows from $A_1 \sim \mathcal{N}(\omega, \sigma^2)$. $\square$

**Theorem 4.7** Fix a target $p \in [0, \tfrac{1}{2})$, a depth $\ell \in \mathbb{N}$, and $\omega > 0$. There exists a unique $\sigma^\star = \sigma^\star(p, \ell, \omega) > 0$ such that $\pi_\ell(\sigma^\star) = p$, and it is given by

$$\sigma^\star(p, \ell, \omega) = -\frac{\omega}{\Phi^{-1}\Big(\dfrac{1 - (1 - 2p)^{1/\ell}}{2}\Big)}. \tag{13}$$

*Proof.* $\pi_\ell(\cdot)$ is continuous and strictly increasing from 0 (at $\sigma \downarrow 0$) to $1/2$ (as $\sigma \uparrow \infty$). Hence for any $p \in [0, 1/2)$ there exists a unique $\sigma^\star > 0$ with $\pi_\ell(\sigma^\star) = p$.

To obtain the explicit form, set $q := p_-(\sigma^\star) = \Phi(-\omega/\sigma^\star) \in (0, 1/2)$. From Lemma A.4 we have

$$p = \pi_\ell(\sigma^\star) = \tfrac{1}{2}\Big(1 - (1 - 2q)^\ell\Big) \implies q = \frac{1 - (1 - 2p)^{1/\ell}}{2}.$$

Applying the inverse CDF $\Phi^{-1}$ to $q = \Phi(-\omega/\sigma^\star)$ yields

$$-\frac{k}{\sigma^\star} = \Phi^{-1}(q) \implies \sigma^\star = -\frac{k}{\Phi^{-1}(q)}.$$

Since $q \in (0, 1/2)$, the quantile $\Phi^{-1}(q) < 0$, so the right-hand side is positive. $\square$

## A.3 THEORETICAL RESULTS FOR SECTION 4.3

Before turning to the empirical comparisons in Section 4.3, we connect our gain calibration to gradient propagation. In the main text we argued that the trainability of very deep networks is governed by how the norm of the backpropagated gradient evolves with depth. In this appendix we make this precise by computing, under standard mean-field assumptions, the layerwise gradient amplification factor $\chi_\ell$ for our structured initialization and contrasting it with the Gaussian i.i.d. case. This will justify the claims in Section 4.3 about the robustness of the proposed scheme to the choice of variance.

Let $\mathbf{g}^\ell := \partial \mathcal{L}/\partial \mathbf{x}^\ell \in \mathbb{R}^{N_\ell}$ denote the gradient at layer $\ell$ for a loss $\mathcal{L}$. By the chain rule and the layerwise relation $\mathbf{h}^{\ell+1} = \mathbf{W}^{\ell+1}\mathbf{x}^\ell + \mathbf{b}^{\ell+1}$, we can write

$$\mathbf{g}^\ell = \left(\mathbf{W}^{\ell+1}\right)^\top \left(f'(\mathbf{h}^{\ell+1}) \odot \mathbf{g}^{\ell+1}\right) = \left(\mathbf{J}^{\ell+1}\right)^\top \mathbf{g}^{\ell+1},$$

where $\odot$ denotes the Hadamard product and

$$\mathbf{J}^{\ell+1} := \operatorname{diag}\left(f'(\mathbf{h}^{\ell+1})\right) \mathbf{W}^{\ell+1}$$

is the Jacobian matrix of layer $\ell + 1$.

In the wide layer regime ($N_\ell \to \infty$) and under standard mean field assumptions given $\mathbf{h}^{\ell+1}$), the squared gradient norms satisfy the approximate recursion

$$\frac{1}{N_\ell} \mathbb{E}\|\mathbf{g}^\ell\|_2^2 \approx \chi_{\ell+1} \frac{1}{N_{\ell+1}} \mathbb{E}\|\mathbf{g}^{\ell+1}\|_2^2, \tag{14}$$

where the layerwise amplification factor $\chi_{\ell+1}$ is defined by

$$\chi_{\ell+1} := \frac{1}{N_{\ell+1}} \mathbb{E}\|\mathbf{J}^{\ell+1}\mathbf{u}\|_2^2,$$

with $\mathbf{u} \sim \mathcal{N}(\mathbf{0}, \mathbf{I}_{N_{\ell+1}})$ independent of $\mathbf{J}^{\ell+1}$. Iterating equation 14 over $\ell = 0, \ldots, L-1$ gives

$$\frac{1}{N_0} \mathbb{E}\|\mathbf{g}^0\|_2^2 \approx \left(\prod_{\ell=0}^{L-1} \chi_{\ell+1}\right) \frac{1}{N_L} \mathbb{E}\|\mathbf{g}^L\|_2^2.$$

We now compute $\chi_{\ell+1}$ for the proposed structured initialization $\mathbf{W}^\ell = \omega \mathbf{D}^\ell + \mathbf{Z}^\ell$. By definition of $\chi_{\ell+1}$,

$$\chi_{\ell+1} = \frac{1}{N_{\ell+1}} \mathbb{E}\|\mathbf{J}^{\ell+1}\mathbf{u}\|_2^2 = \frac{1}{N_{\ell+1}} \mathbb{E} \sum_{i=1}^{N_{\ell+1}} \left(\sum_{j=1}^{N_\ell} w_{ij}^{\ell+1} f'(h_i^{\ell+1}) u_j\right)^2.$$

Conditioning on $\mathbf{W}^{\ell+1}$ and $\mathbf{h}^{\ell+1}$, the inner sum is a centred Gaussian in $\mathbf{u}$ with variance

$$\sum_{j=1}^{N_\ell} \left(w_{ij}^{\ell+1}\right)^2 f'\left(h_i^{\ell+1}\right)^2.$$

Taking the expectation over $\mathbf{u}$ and then over the weights and preactivations yields

$$\chi_{\ell+1} = \frac{1}{N_{\ell+1}} \mathbb{E} \sum_{i=1}^{N_{\ell+1}} f'\left(h_i^{\ell+1}\right)^2 \sum_{j=1}^{N_\ell} \left(w_{ij}^{\ell+1}\right)^2$$

$$\approx \mathbb{E}\left[f'\left(h_1^{\ell+1}\right)^2 \sum_{j=1}^{N_\ell} \left(w_{1j}^{\ell+1}\right)^2\right],$$

where we used exchangeability of the rows of $\mathbf{W}^{\ell+1}$ and of the coordinates of $\mathbf{h}^{\ell+1}$.

For the proposed initialization, $(\mathbf{Z}_{ij}^{\ell+1}) \sim \mathcal{N}(0, \sigma_z^2/N_\ell)$, so

$$\sum_{j=1}^{N_\ell} \left(w_{1j}^{\ell+1}\right)^2 = \omega^2 + \sum_{j=1}^{N_\ell} \left(Z_{1j}^{\ell+1}\right)^2.$$

Taking expectation over $\mathbf{Z}^{\ell+1}$ and using $\mathbb{E}\big[(Z_{1j}^{\ell+1})^2\big] = \sigma_z^2/N_\ell$ gives

$$\mathbb{E}\Big[\sum_{j=1}^{N_\ell}(w_{1j}^{\ell+1})^2\Big] = \omega^2 + \sum_{j=1}^{N_\ell}\mathbb{E}\big[(Z_{1j}^{\ell+1})^2\big] = \omega^2 + \sigma_z^2.$$

Moreover, by construction of the initialization, $\mathbf{W}^{\ell+1}$ and $\mathbf{h}^{\ell+1}$ are independent, and the coordinates $h_i^{\ell+1}$ are exchangeable. Hence

$$\mathbb{E}\Big[f'\big(h_1^{\ell+1}\big)^2 \sum_{j=1}^{N_\ell}(w_{1j}^{\ell+1})^2\Big] \approx \mathbb{E}\big[f'(h^{\ell+1})^2\big]\,\mathbb{E}\Big[\sum_{j=1}^{N_\ell}(w_{1j}^{\ell+1})^2\Big] = \big(\omega^2 + \sigma_z^2\big)\,\mathbb{E}\big[f'(h^{\ell+1})^2\big].$$

Thus,

$$\chi_{\ell+1} \approx \big(\omega^2 + \sigma_z^2\big)\,\mathbb{E}\big[f'(h^{\ell+1})^2\big]. \tag{15}$$

Equation equation 15 shows that, for the proposed initialization, the mean-field amplification factor $\chi_{\ell+1}$ depends on the combined scale $\omega^2 + \sigma_z^2$ rather than on a bare variance parameter alone. In particular, increasing the noise level $\sigma_z$ decreases $\mathbb{E}[f'(h^{\ell+1})^2]$ while increasing the prefactor $\omega^2 + \sigma_z^2$, so that $\chi_{\ell+1}$ remains close to one over a much wider range of $\sigma_z$ than in the Gaussian i.i.d. case, where $\chi_{\ell+1}^\star \approx \sigma_w^2\,\mathbb{E}[f'(h^{\ell+1})^2]$ has no analogous $\omega^2$ term. This analytic behavior underlies the empirical observations in Section 4.3 that our initialization preserves gradient norms more effectively across depth and is substantially more robust to variance misspecification.

# B ADDITIONAL EXPERIMENTAL RESULTS WITHOUT TRAINED NETWORKS

## B.1 DEFINITIONS OF ACTIVATION FUNCTIONS

This section introduces the activation functions considered in this paper. The following functions belong to the odd-sigmoid function class.

**Gudermannian function**    The Gudermannian function $\mathrm{gd} : \mathbb{R} \to \mathbb{R}$ is defined by

$$\mathrm{gd}(x) = \int_0^x \frac{dt}{\cosh t} = 2 \arctan\left(\tanh\left(\tfrac{x}{2}\right)\right).$$

**Error Function.**    The error function $\mathrm{erf} : \mathbb{R} \to \mathbb{R}$ is defined as

$$\mathrm{erf}(x) = \frac{2}{\sqrt{\pi}} \int_0^x e^{-t^2}\, dt.$$

**Softsign-Type Functions.**    For $k \geq 1$, we define the generalized softsign function

$$\mathrm{softsign}_k(x) \;=\; \frac{x}{\left(1 + |x|^k\right)^{1/k}}, \qquad x \in \mathbb{R}.$$

This family interpolates between several commonly used smooth odd activations:

$$\mathrm{softsign}_1(x) = \frac{x}{1 + |x|}, \quad \mathrm{softsign}_2(x) = \frac{x}{\sqrt{1 + x^2}}, \quad \mathrm{softsign}_3(x) = \frac{x}{\left(1 + |x|^3\right)^{1/3}}.$$

We also use the combined variant

$$\mathrm{softsign}_{1+3}(x) := \mathrm{softsign}_1(x) + \mathrm{softsign}_3(x).$$

**Hyperbolic Tangent.**    The hyperbolic tangent function $\tanh : \mathbb{R} \to \mathbb{R}$ is defined by

$$\tanh(x) = \frac{e^x - e^{-x}}{e^x + e^{-x}}.$$

**Arctangent.**    The (scaled) arctangent function $\arctan : \mathbb{R} \to \mathbb{R}$ is given by

$$\arctan(x) = \int_0^x \frac{dt}{1 + t^2}.$$

In practice, we often use the normalized form $\frac{2}{\pi} \arctan(x)$ so that its range matches that of $\tanh(x)$.

### B.2 Properties of the Odd-Sigmoid Functions

This section empirically investigates the properties of the iteration $x_{n+1} = f(ax_n)$. When the gain $a$ is fixed across all iterations, Figure 8 shows how the dynamics depend on the initial data: for any nonzero input, the iterates converge to the same nonzero fixed point $\xi_a$. Figure 9 further demonstrates that this limit depends only on the gain, not on the input, by comparing distinct gains $a \neq a' > 0$ and observing convergence to different fixed points $\xi_a$ and $\xi_{a'}$. These results indicate that, by appropriately choosing $a$, one can control how long the signal remains informative across layers and thus preserve information up to a desired depth.

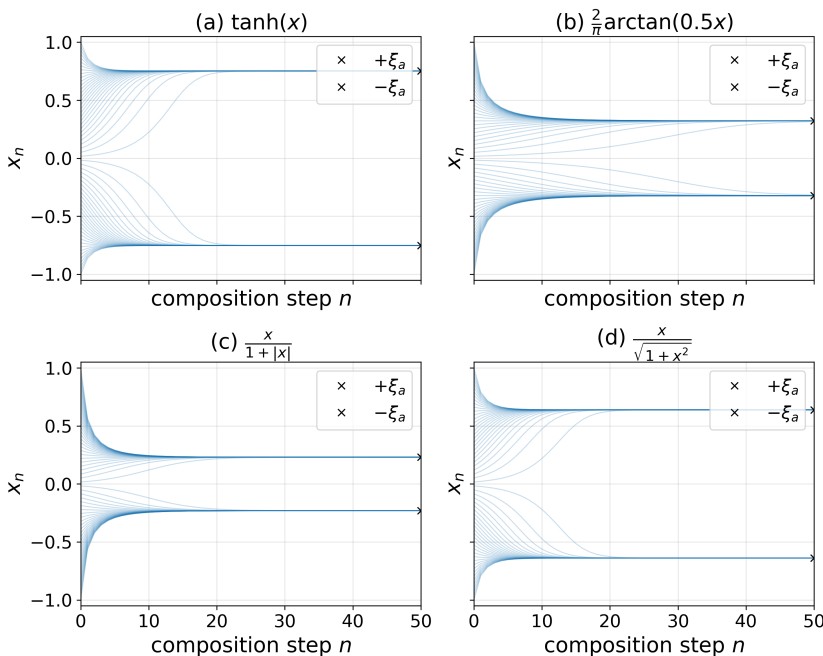

Figure 8: Iterated scalar dynamics for four odd-sigmoid activations under a fixed supercritical gain. For each activation $f$, we compute $\omega = 1/f'(0)$ and set $a = \omega + 0.3$. We then iterate the one–dimensional map $x_{n+1} = f(ax_n)$ for $n = 0, \ldots, 50$ starting from 60 initial values $x_0 \in [-1, 1]$. The limiting fixed points $\pm\xi_a$ satisfying $f(a\xi_a) = \xi_a$ are approximated by iterating from $\pm 1$ and are marked at $n = 50$ with "$\times$".

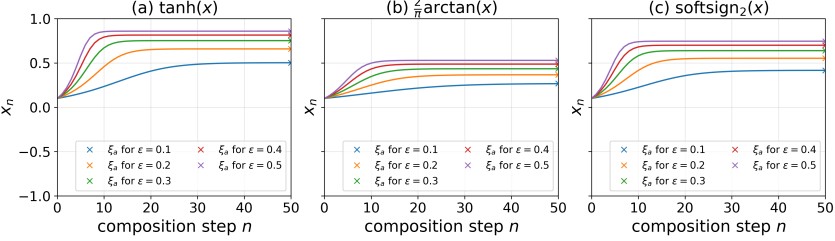

Figure 9: Convergence of the iteration $x_{n+1} = f(ax_n)$ for odd–sigmoid $f$ with $a = \omega + \epsilon$ ($\epsilon \in \{0.1, 0.2, 0.3, 0.4, 0.5\}$); curves show $x_n$ up to $n = 50$ from $x_0 = 0.1$, and '$\times$' marks the positive fixed point $\xi_a$ solving $f(a\xi_a) = \xi_a$.

### B.3 NEGATIVE RATE FUNCTION

We calibrate the noise scale $\sigma_z$ in the surrogate model by targeting a desired negative rate at a specified depth. For the scalar recursion with gains $A_j \sim \mathcal{N}(\omega, \sigma^2)$ and depth $L$, let $\pi_L(\sigma)$ denote the probability that the iterate is negative at layer $L$. Given a target $p \in [0, \frac{1}{2})$, Theorem 4.7 yields a unique $\sigma^*(p, L, \omega)$ such that $\pi_L(\sigma^*) = p$. Figure 10 illustrates this calibration: panel (a) shows the closed-form scale $\sigma^*(p, L, \omega)$ as a function of $p$ for $L = 100$ and $\omega = 1$, while panel (b) plots $\pi_L(\sigma)$ over $(L, \sigma)$ for $\omega = 1$. As expected, $\pi_L(\sigma) \to 0$ as $\sigma \to 0$ (no sign flips) and $\pi_L(\sigma) \to \frac{1}{2}$ as $\sigma \to \infty$ (full symmetry), so the negative rate traces a narrow band between these two extremes.

Figure 11 evaluates the resulting calibration from a mean-field perspective. For $f(x) = \tanh(x)$ and $\omega = 1$, we choose $\sigma_z = \sigma^*(p, L, \omega)$ for target negative rates $p = 0.01$ and $p = 0.49$, and compute the corresponding gradient amplification factor $\chi_\ell \approx (\omega^2 + \sigma_z^2)\,\mathbb{E}[f'(h^\ell)^2]$ across depths. The curves show that $\chi_\ell$ remains very close to 1 over a wide range of depths, up to $L = 2 \times 10^5$, for both target negative rates, indicating that the negative-rate calibration keeps gradients in a trainable regime even in extremely deep networks.

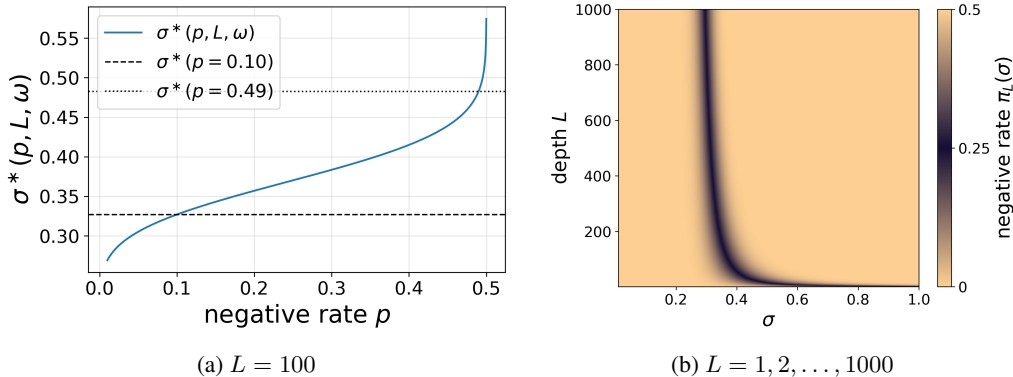

(a) $L = 100$           (b) $L = 1, 2, \ldots, 1000$

Figure 10: Closed-form characterization of the scalar surrogate. **(a)** Closed form scale $\sigma^*(p, L, \omega)$ as a function of the target negative rate $p$ for $L = 100$ and $\omega = 1$, with reference lines at $p = 0.10$ and $p = 0.49$. **(b)** Closed form scalar surrogate negative rate $\pi_L(\sigma)$ for $\omega = 1$, network depths $L = 1, 2, \ldots, 1000$, and $\sigma \in [0.01, 1.0]$.

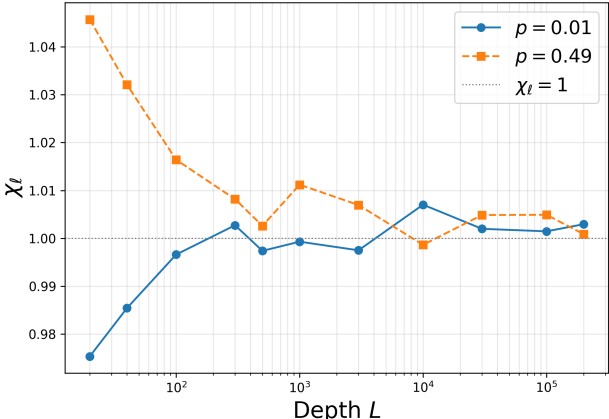

Figure 11: $\chi_\ell$ for the proposed initialization with $f(x) = \tanh(x)$, evaluated at the closed form noise scales $\sigma^*(p, L, \omega)$ corresponding to target negative rates $p = 0.01$ and $p = 0.49$. For each depth $\ell$, we estimate $\chi_\ell(\sigma_z) \approx (\omega^2 + \sigma_z^2)\,\mathbb{E}[f'(h^\ell)^2]$.

### B.4 NEGATIVE RATE SURROGATE ANALYSIS

We begin from the elementwise representation of a fully connected network with odd–sigmoid activation $f \in \mathcal{F}$. For each layer $\ell$ and neuron index $i$, the forward update can be written as

$$x_i^{\ell+1} = f\left(a_i^{\ell+1} x_i^\ell\right), \qquad a_i^{\ell+1} = w_{ii}^{\ell+1} + \sum_{j \neq i} \frac{w_{ij}^{\ell+1} x_j^\ell}{x_i^\ell}, \tag{16}$$

so that $a_i^{\ell+1}$ plays the role of an effective scalar gain for neuron $i$ in layer $\ell + 1$. Under our proposed initialization $W^{\ell+1} = \omega D^{\ell+1} + Z^{\ell+1}$ with $(Z_{ij}^{\ell+1}) \sim \mathcal{N}(0, \sigma_z^2/N_\ell)$, Lemma 4.5 shows that, conditional on $x^\ell$,

$$a_i^{\ell+1} \sim \mathcal{N}\left(\omega, \frac{\sigma_z^2}{N_\ell}\left(1 + \sum_{j \neq i}\left(\frac{x_j^\ell}{x_i^\ell}\right)^2\right)\right). \tag{17}$$

To handle this dependence on the current activations, we approximate the gain distribution by a scalar surrogate model $a_i^{\ell+1} \sim \mathcal{N}(\omega, \sigma_z^2)$ and estimate sign statistics from this Gaussian approximation.

We define the negative rate at depth $L$ as

$$\pi_L(\sigma) := \mathbb{P}\left(x_L < 0 \,\middle|\, x_0 > 0\right),$$

that is, the probability that a neuron whose initial activation is positive ends up with a negative sign at layer $L$. Because every $f \in \mathcal{F}$ is odd and strictly increasing, the sign of $x_i^\ell$ is entirely controlled by the product of the effective gains along the path, and the probability that a positive entry becomes negative at depth $L$ equals the probability that a negative entry becomes positive. It therefore suffices to track a single scalar chain starting from $x_0 > 0$.

Our initialization has a nonzero mean gain $\omega$, which makes sign changes along a given coordinate well defined across layers. We interpret frequent sign flips during the forward pass as a form of information loss, and we therefore use the surrogate calibration to control the negative rate at the final layer. In the scalar surrogate, for a given depth $L$ and gain variance $\sigma^2$ we can compute $\pi_L(\sigma)$ in closed form. We then define, for a target $p \in (0, \frac{1}{2})$, the calibrated scale $\sigma^*(p, L, \omega)$ as the unique solution to $\pi_L(\sigma^*) = p$ (see Theorem 4.7). In practice, we choose a desired "real" negative rate $p_{\text{real}} = 0.4$ and select $\sigma_z$ so that the empirical FFNN-driven negative rate at depth $L$ is close to $p_{\text{real}}$, which preserves most sign information while still leaving enough randomness for learning.

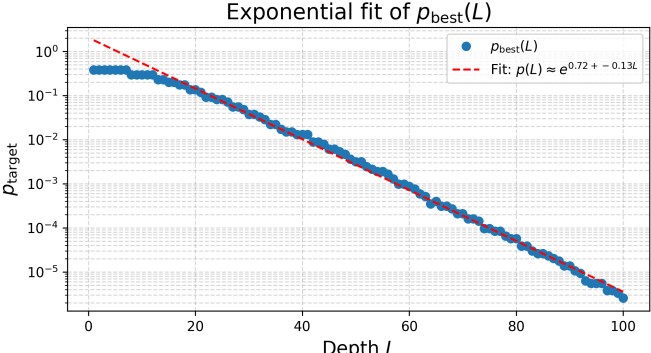

Figure 12: Best surrogate target $p_{\text{target}}(L)$ producing an FFNN-driven negative rate $\tilde{\pi}_L \approx 0.4$ as a function of depth $L$ for a $\tanh$ network with width $64$. The curve shows that $p_{\text{target}}(L)$ stays near $0.3$–$0.4$ for shallow depths and then decays approximately exponentially with $L$, providing an empirical calibration rule for choosing the surrogate negative rate in deep networks.

To understand how the scalar surrogate relates to the actual network, we compare $\pi_L(\sigma^*)$ with the empirical negative rate obtained from an initialized FFNN. For each $p_{\text{target}}$ and depth $L$, we first compute $\sigma^*(p_{\text{target}}, L, \omega)$, initialize an $L$-layer $\tanh$ network with weights $W^\ell = \omega I + (\sigma^*/\sqrt{N})G^\ell$, extract the effective gains $a_i^\ell$, and form FFNN-driven scalar chains $y_0 = 1$,

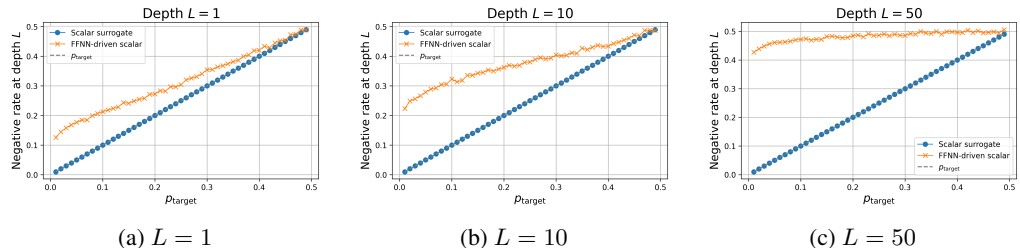

(a) $L = 1$        (b) $L = 10$        (c) $L = 50$

Figure 13: Negative rate at depth $L = 1, 10, 50$ as a function of the target negative rate $p_{\text{target}}$. The curve $\pi_1(\sigma^\star)$ denotes the scalar surrogate prediction at the calibrated scale $\sigma^\star(p_{\text{target}})$, $\tilde{\pi}_1$ is the empirical negative rate obtained from the FFNN-driven scalar chains, and the dashed line shows the identity $p_{\text{target}}$.

$y_{\ell+1} = \tanh(a^\ell y_\ell)$. Measuring $\tilde{\pi}_L = \mathbb{P}(y_L < 0)$ and comparing it to the analytic $\pi_L(\sigma^*)$ reveals that the discrepancy grows with depth (see Figure 13), as expected from the increasing influence of higher-order correlations. Nevertheless, the relationship between the surrogate target $p_{\text{target}}$ and the FFNN-driven negative rate $\tilde{\pi}_L$ remains systematic.

In particular, for each depth $L$ we can invert this relationship numerically: we search over $p_{\text{target}}$ and find the value $p_{\text{sur}}(L)$ such that the FFNN-driven negative rate $\tilde{\pi}_L$ is as close as possible to $p_{\text{real}} = 0.4$ (see Section B.7 ).

The resulting sequence $p_{\text{sur}}(L)$ is reported in Figure 12. For small depths $L$, the surrogate target remains near 0.3–0.4, but for larger $L$ it decays approximately exponentially. Fitting a simple model $\log p_{\text{sur}}(L) \approx c_0 + c_1 L$ on the tail (e.g., $L \geq 10$) yields

$$p_{\text{sur}}(L) \approx Ce^{-\alpha L},$$

for some $C > 0$ and $\alpha > 0$. This empirical law expresses how aggressively the surrogate target negative rate must shrink with depth in order for the actual network to maintain a fixed, moderate negative rate at its final layer.

These calibration insights are consistent with our training experiments. Figures 14 and 15 plot validation accuracy as a function of the surrogate target $p$ for deep ($L = 50$) and shallow ($L = 3$) networks. For the deep case, performance is maximized when $p_{\text{sur}}$ is on the order of 0.01, in line with the exponentially decayed target predicted by the surrogate analysis. In contrast, for shallow networks the best performance is attained near $p_{\text{sur}} \approx 0.4$, matching the regime where the fitted decay has not yet taken effect. Finally, Figures 16 and 17 show that, for depths $L = 50$ and $L = 100$,

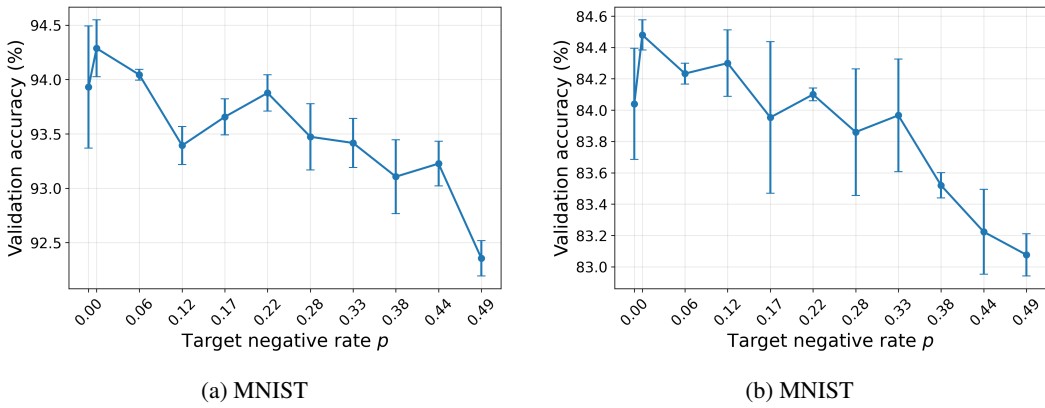

(a) MNIST        (b) MNIST

Figure 14: Validation accuracy as a function of the target negative rate $p$ for a 50-layer fully connected network (width 64) with activation $f(x) = \tanh(x)$ under the proposed initialization. Each point shows the mean $\pm$ standard deviation over 5 runs, where each run is trained for 600 iterations, with $\sigma^*(p, L, \omega)$ computed from the scalar surrogate calibration.

the learning curves are optimized near the $\sigma^*$ values obtained from the calibrated surrogate, with accuracy degrading as we move away from these scales. Together, these results validate that the negative-rate surrogate provides a useful, quantitatively accurate guideline for choosing the noise level $\sigma_z$ across depths, and that the resulting diagonal–plus–noise initialization indeed preserves signal sign statistics in deep networks.

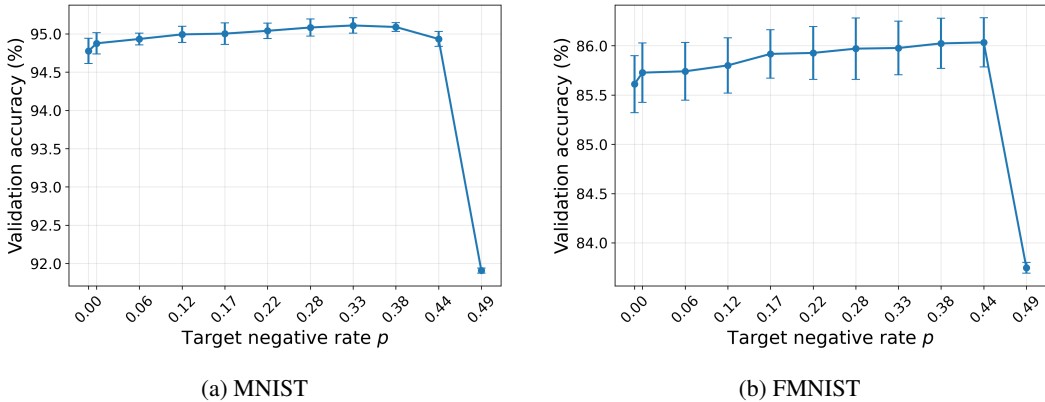

(a) MNIST

(b) FMNIST

Figure 15: Validation accuracy as a function of the target negative rate $p$ for a 3-layer fully connected network (width 512) with activation $f(x) = \tanh(x)$ under the proposed initialization. Each point shows the mean $\pm$ standard deviation over 5 runs, where each run is trained for 600 iterations, with $\sigma^*(p, L, \omega)$ computed from the scalar surrogate calibration.

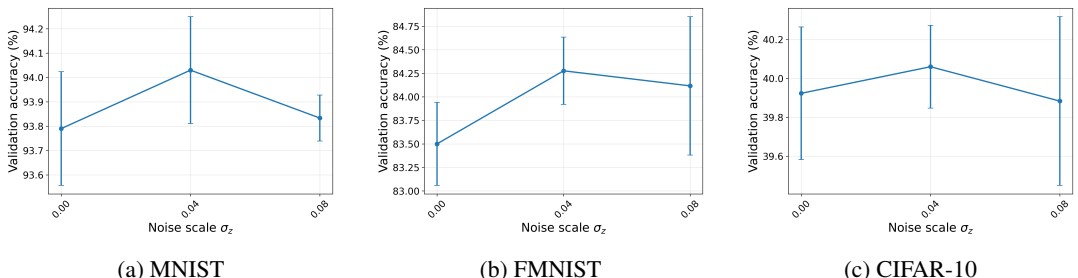

(a) MNIST

(b) FMNIST

(c) CIFAR-10

Figure 16: Validation accuracy as a function of the noise scale $\sigma_z$ for 50 layer fully connected networks (width 64) with activation $f(x) = \tanh(x)$ under the proposed initialization, on MNIST, Fashion MNIST, and CIFAR-10. For each $\sigma_z \in \{0, 0.03, 0.06\}$, we report the mean $\pm$ standard deviation over 5 training runs.

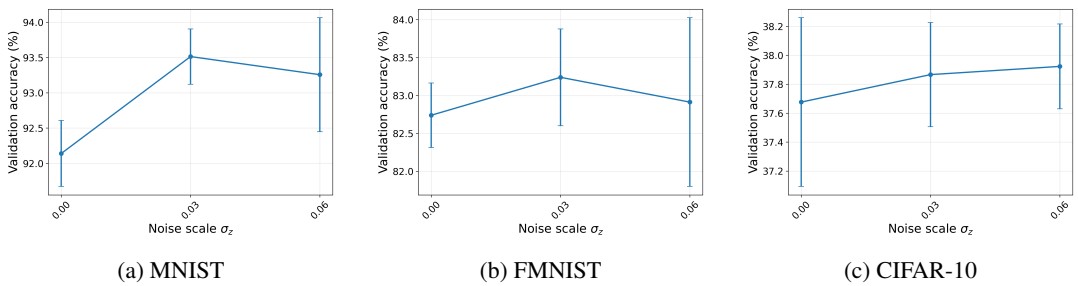

(a) MNIST

(b) FMNIST

(c) CIFAR-10

Figure 17: Validation accuracy as a function of the noise scale $\sigma_z$ for 100 layer fully connected networks (width 64) with activation $f(x) = \tanh(x)$ under the proposed initialization, on MNIST, Fashion MNIST, and CIFAR-10. For each $\sigma_z \in \{0, 0.03, 0.06\}$, we report the mean $\pm$ standard deviation over 5 training runs.

## B.5   FORWARD SIGNAL PROPAGATION

In Section 4.3 we study forward signal propagation under the proposed initialization. Figure 10 (a) reports how well the last-layer activation distribution is preserved as the depth $L$ increases (with fixed width $W = 64$), while panel (b) varies the width $N_\ell$ at fixed depth $L = 1000$ to test whether the distribution remains well spread even in relatively narrow networks. In both settings, the proposed initialization maintains a dispersed last-layer distribution up to depth $L = 10{,}000$ and down to width $N_\ell = 20$, whereas the EOC (Gaussian i.i.d.) initialization quickly collapses toward a more concentrated distribution.

To summarize how well the last-layer distribution is dispersed, we employ a normalized histogram-entropy score

$$\text{Spread}(x) = \frac{-\sum_{i=1}^{B} p_i \log p_i}{\log B}, \qquad p_i = \int_{\text{bin } i} \hat{f}_x(t)\, dt, \ \ \sum_{i=1}^{B} p_i = 1, \tag{18}$$

where $\hat{f}_x$ is the empirical density over $[-1, 1]$ using $B$ bins. Values close to $1$ indicate a highly dispersed (near-uniform) last-layer distribution, whereas values near $0$ correspond to a highly concentrated distribution.

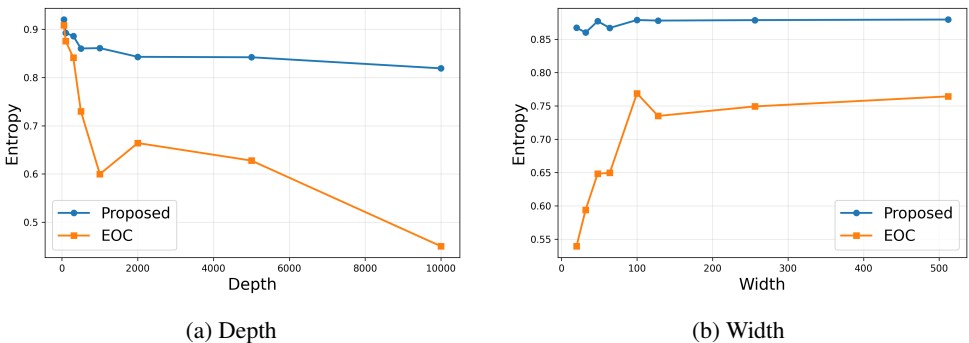

(a) Depth                           (b) Width

Figure 18: Entropy of the last-layer activation distribution for tanh networks under the proposed initialization and the EOC initialization. **(a)** Entropy as a function of depth $L$ with fixed width $W = 64$. **(b)** Entropy as a function of width $N_\ell$ with fixed depth $L = 1000$, using widths $W \in \{20, 32, 48, 64, 100, 128, 256, 512\}$ as shown in the panels. The entropy is defined in Equation 18.

Figures 19, 20, and 21 visualize the last-layer activation histograms obtained after the initial forward pass in 1000-layer FFNNs of width $64$ with activations $0.1 \tanh(x)$, $\tanh(x)$, and $10 \tanh(x)$, respectively. Under the proposed initialization, the activation distribution remains well dispersed in all three cases, whereas Gaussian i.i.d. initializations quickly collapse toward a narrow band around zero. Figures 22 and 23 show a similar comparison as the width is reduced: even for very narrow networks, the proposed scheme preserves a spread-out last layer distribution, while Gaussian initializations drive the last layer activations to saturate near zero.

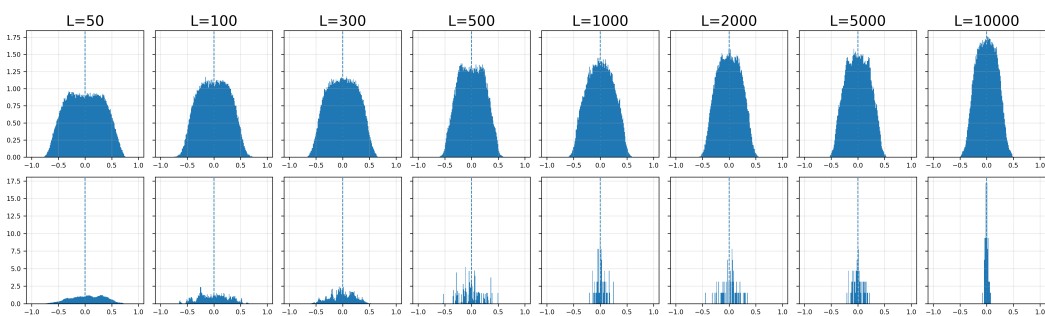

Figure 19: Last layer activation histograms for $\tanh$ networks with width $W = 64$ and varying depth under the proposed initialization (top row) and the EOC initialization (bottom row). Each column corresponds to a different depth $L$.

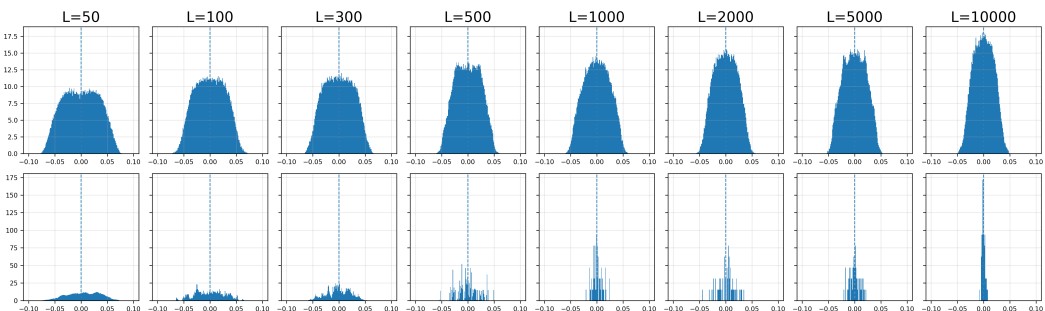

Figure 20: Last layer activation histograms for $0.1 \tanh$ networks with width $W = 64$ and varying depth under the proposed initialization (top row) and the EOC initialization (bottom row). Each column corresponds to a different depth $L$.

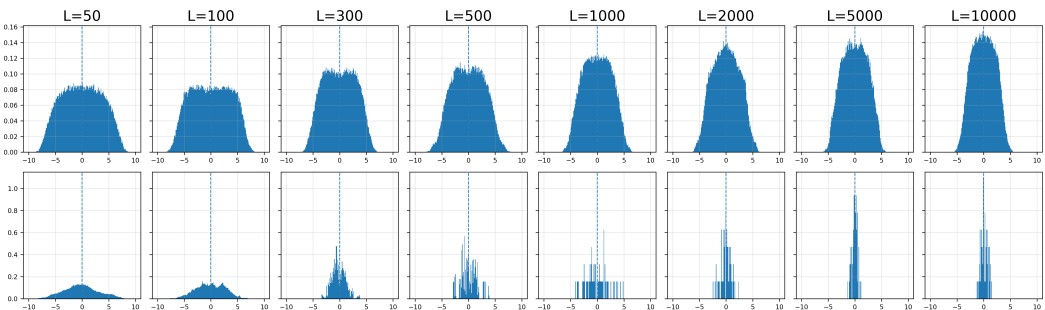

Figure 21: Last layer activation histograms for $10 \tanh$ networks with width $W = 64$ and varying depth under the proposed initialization (top row) and the EOC initialization (bottom row). Each column corresponds to a different depth $L$.

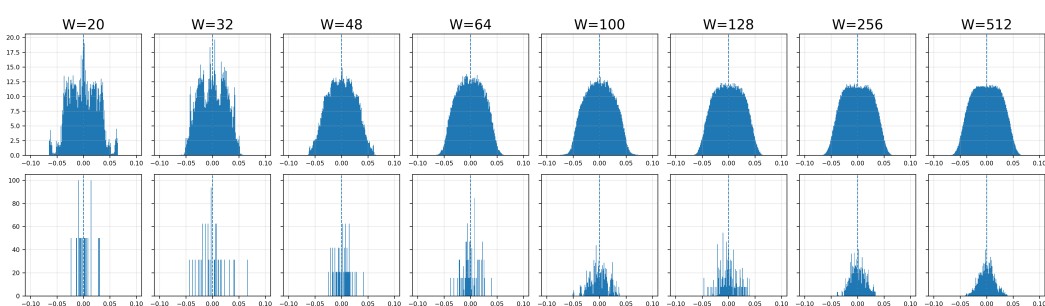

Figure 22: Last layer activation histograms for $0.1 \tanh$ networks with depth $L = 1000$ and varying width under the proposed initialization (top row) and the EOC initialization (bottom row).

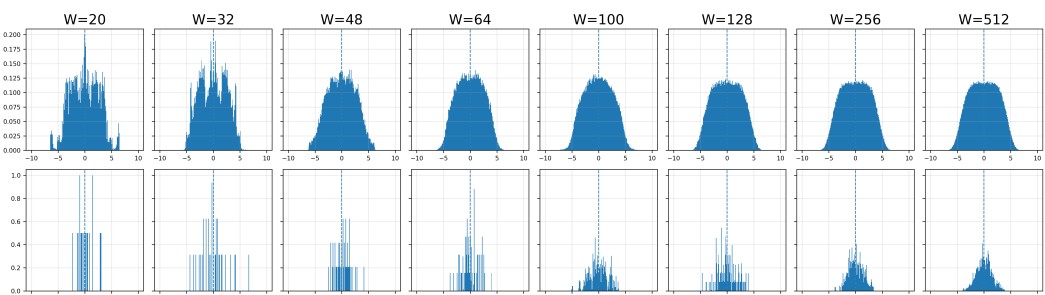

Figure 23: Last layer activation histograms for $10 \tanh$ networks with depth $L = 1000$ and varying width under the proposed initialization (top row) and the EOC initialization (bottom row).

### B.6 BACKWARD SIGNAL PROPAGATION

Figure 24 visualizes $|\chi_L(\sigma) - 1|$ for the proposed initialization and a Gaussian initialization with $\tanh$ activations, as a function of depth $L$ and scale $\sigma$. For the proposed scheme, the region where $\chi_L(\sigma) \approx 1$ occupies a broad band in the $(L, \sigma)$ plane, whereas for the Gaussian initialization it is confined to a narrow strip around a single variance. This indicates that our initialization keeps $\chi_L$ close to 1 over a much wider range of noise scales and depths, and is therefore more robust and better suited for stable training in deep, narrow networks.

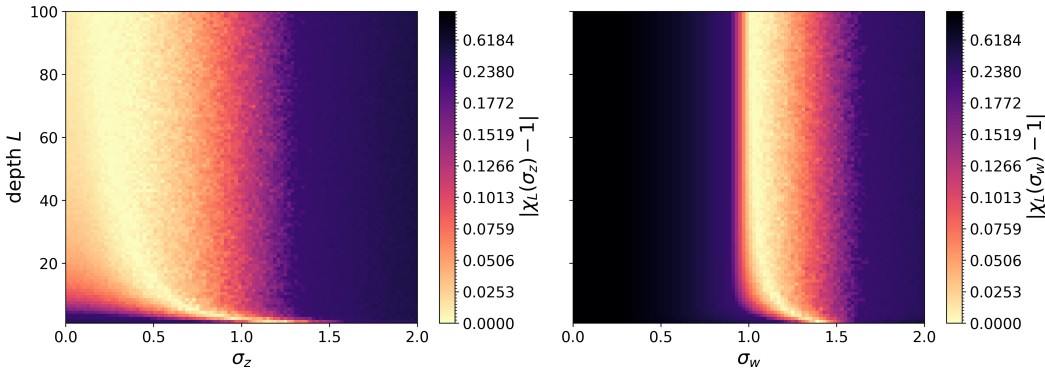

Figure 24: Heatmaps of the deviation $|\chi_L(\sigma) - 1|$ for the proposed initialization (**left**) and a Gaussian i.i.d. initialization (**right**) with $\tanh$ activations, as a function of depth $L$ and scale $\sigma$. Brighter bands indicate near critical regimes where forward and backward signals are approximately preserved.

## B.7 Choice of the target negative rate $p_{\text{real}} = 0.4$.

Because $f \in \mathcal{F}$ is odd and strictly increasing, the sign of each coordinate $x_i^\ell$ is entirely determined by the product of the effective gains along that coordinate. We therefore use the coordinate-wise sign flip probability

$$\tilde{\pi}_L := \mathbb{P}(x_L < 0 \mid x_0 > 0)$$

as a simple indicator for how much "sign information" about the input is preserved at depth $L$. Two extreme regimes are undesirable. If $\tilde{\pi}_L \approx 0$, almost all coordinates preserve their initial sign, so the network behaves nearly like an identity map at initialization; this preserves information but yields very limited expressiveness and weak exploration of the odd–sigmoid nonlinearity. At the other extreme, if $\tilde{\pi}_L \approx 0.5$, the final sign is essentially a fair coin flip regardless of the initial sign, meaning that the directional information carried by the input has been almost completely randomized and we interpret this as a form of information loss.

In practice, we therefore target an intermediate regime in which most coordinates keep their initial sign, but a non-negligible fraction flip so that the representation can change meaningfully. Concretely, we fix a desired "real" negative rate $p_{\text{real}} = 0.4$ and choose $\sigma_z$ so that the empirical FFNN-driven negative rate at depth $L$ satisfies $\tilde{\pi}_L \approx p_{\text{real}}$. This calibration preserves the sign of the majority of coordinates ($\approx 60\%$) while still allowing a substantial minority ($\approx 40\%$) to flip, which provides enough randomness for learning without completely destroying the initial sign structure.

We do not claim that $p_{\text{real}} = 0.4$ is an information-theoretically optimal value. Rather, it is an empirically grounded target: across a wide range of depths, widths, datasets, and activation scales, we consistently observe that the $\sigma_z$ obtained from the $p_{\text{real}} = 0.4$ calibration yields the best or near-best validation performance, with accuracy degrading as we move to significantly smaller or larger negative rates (see Figures 14, 15, 16, and 17). This suggests that the proposed negative rate criterion provides a practically useful operating point for odd–sigmoid networks.

# C ADDITIONAL EXPERIMENTAL RESULTS WITH NEURAL NETWORKS

**Experimental Setting.** We evaluate the proposed initialization using the Adam optimizer with a batch size of 128 and reserve 15% of the training data for validation. All experiments are implemented in PyTorch without skip connections and without learning rate decay. We set the learning rate to $10^{-4}, \omega$ and we use the same learning rate for all Gaussian i.i.d. baseline initializations (Xavier, He, and EOC) for a fair comparison.

## C.1 NETWORK SIZE INDEPENDENCE

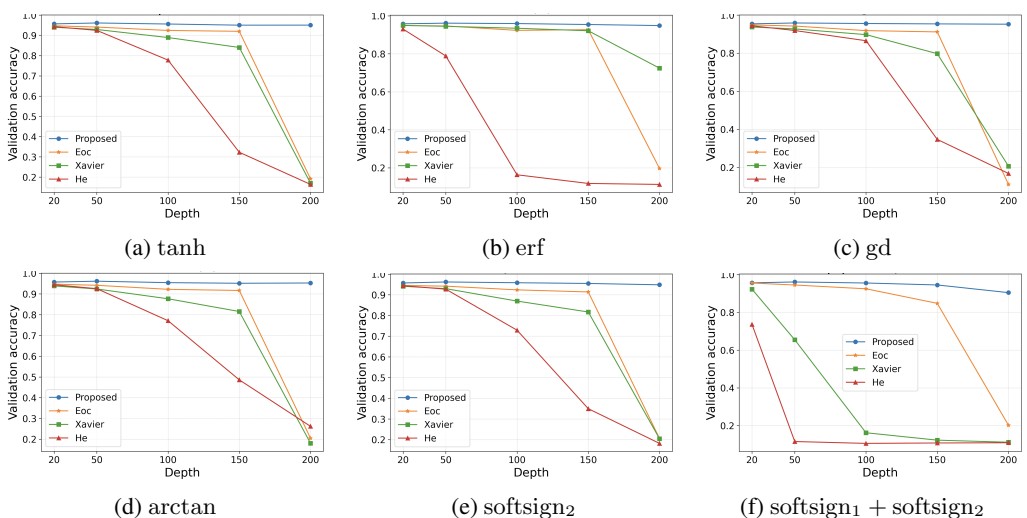

Figure 25: MNIST validation accuracy versus depth for FFNNs (width 64) with odd-sigmoid activations and four initializations (Proposed, EOC, Xavier, He). Each panel fixes one activation and shows the best validation accuracy over 10 epochs for depths $L \in \{20, 50, 100, 150, 200\}$.

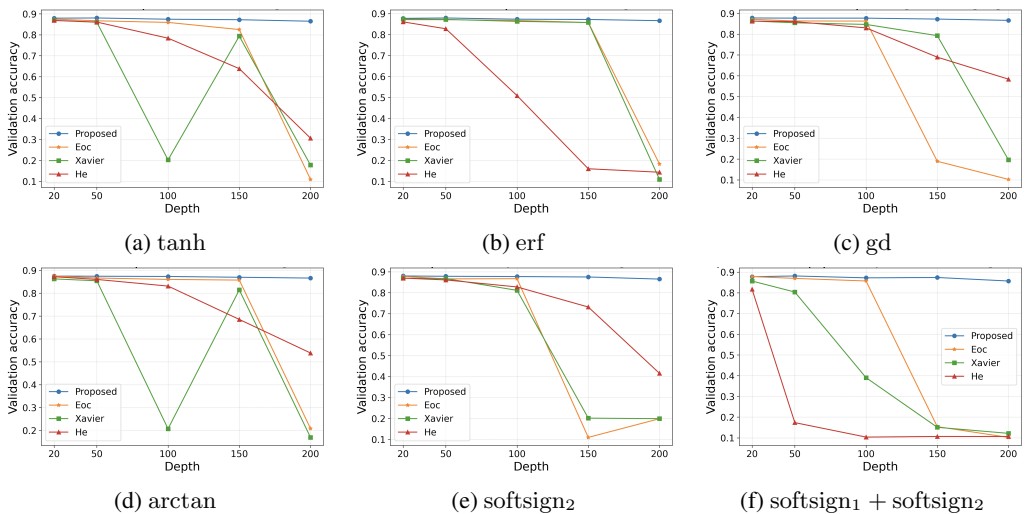

Figure 26: Fashion MNIST validation accuracy versus depth for FFNNs (width 64) with odd-sigmoid activations and four initializations (Proposed, EOC, Xavier, He). Each panel fixes one activation and shows the best validation accuracy over 10 epochs for depths $L \in \{20, 50, 100, 150, 200\}$.

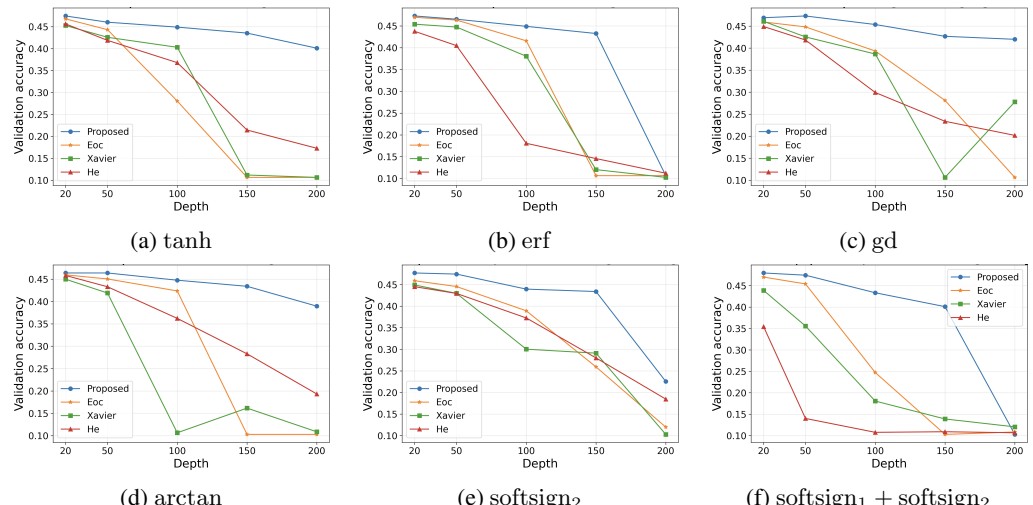

Figure 27: CIFAR 10 validation accuracy versus depth for FFNNs (width 64) with odd-sigmoid activations and four initializations (Proposed, EOC, Xavier, He). Each panel fixes one activation and shows the best validation accuracy over 10 epochs for depths $L \in \{20, 50, 100, 150, 200\}$.

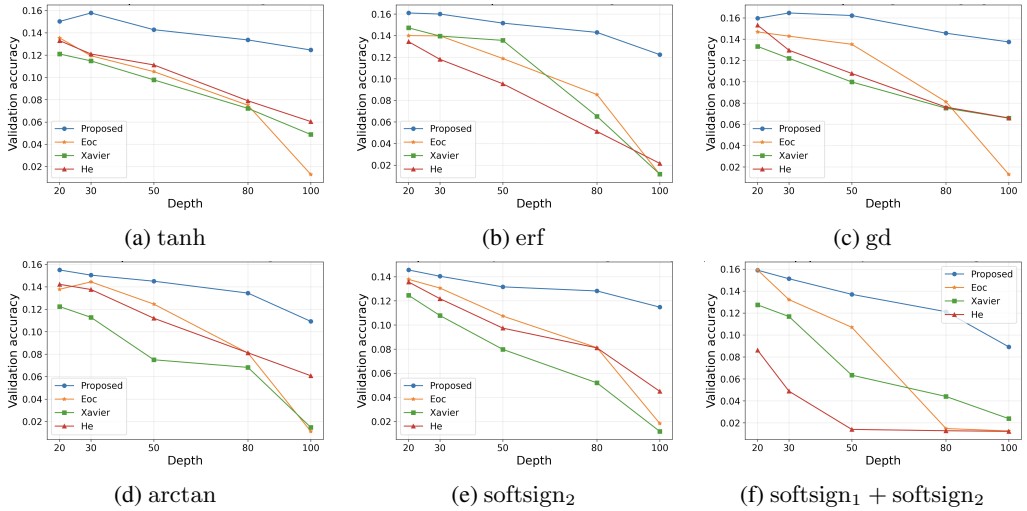

Figure 28: CIFAR 100 validation accuracy versus depth for FFNNs (width 64) with odd-sigmoid activations and four initializations (Proposed, EOC, Xavier, He). Each panel fixes one activation and shows the best validation accuracy over 10 epochs for depths $L \in \{20, 50, 100, 150, 200\}$.

Figures 25, 26, 27, and 28 report the validation accuracy as a function of depth for networks with six different activation functions. On the more complex CIFAR-10 and CIFAR-100 datasets, even the proposed initialization exhibits some change in performance once the depth becomes sufficiently large, whereas on MNIST and Fashion-MNIST the validation accuracy is essentially preserved across all depths considered. These results indicate that, across diverse datasets and for odd–sigmoid activations with effective gain parameter $\omega$ close to one, the proposed initialization behaves substantially more depth invariant than standard Gaussian initializations.

## C.2 BATCH NORMALIZATION FREE TRAINING

Batch normalization (BN) has made training deep networks substantially easier, but it also incurs a significant computational overhead. This raises the question of whether our initialization can match or surpass BN-based methods while avoiding this overhead. Figure 30 evaluates $\tanh$ networks and shows that, across all datasets, the proposed initialization without BN outperforms or matches competing schemes that rely on BN. Figures 29 and 31 extend this comparison to activations with $\omega = 1/f'(0)$ far from 1, including an 800 layer network with width 32. In this challenging deep and narrow regime, our method still trains reliably without BN, whereas Gaussian initializations (with or without BN) either fail to converge or achieve worse accuracy. These results suggest that the proposed initialization can substitute for BN in many settings, enabling stable training of very deep and narrow networks without the cost of normalization layers.

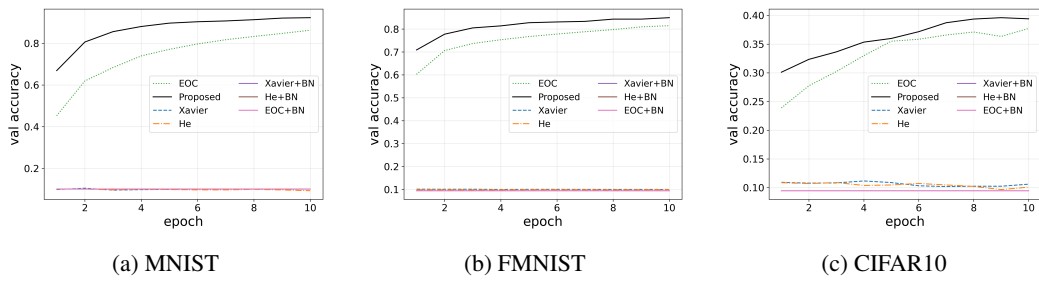

| (a) MNIST | (b) FMNIST | (c) CIFAR10 |

Figure 29: Validation accuracy for the $f(x) = \tanh(ax) + \mathrm{erf}(bx) + x/(1 + |cx|) + \mathrm{gd}(dx)$ in FFNN with 100 hidden layers of width 64. We compare seven strategies: Proposed, Xavier, He, EOC, and their BN variants. Each panel uses a different dataset and a different choice of coefficients $(a, b, c, d)$: **(a)** (10,1000,10,1), **(b)** (100,1000,10,1000)), **(c)**(1000,100,0.1,0.01).

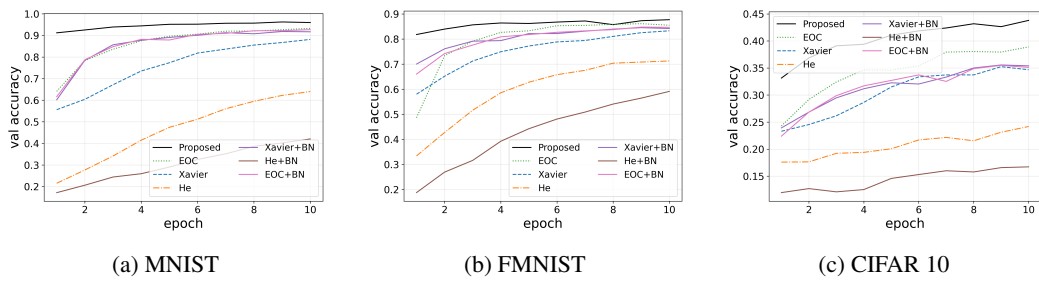

| (a) MNIST | (b) FMNIST | (c) CIFAR 10 |

Figure 30: Validation accuracy over 10 epochs for fully connected 100 layer networks with $tanh$ activations and width 64. The curves compare the Proposed, EOC, Xavier, and He initializations, with and without batch normalization.

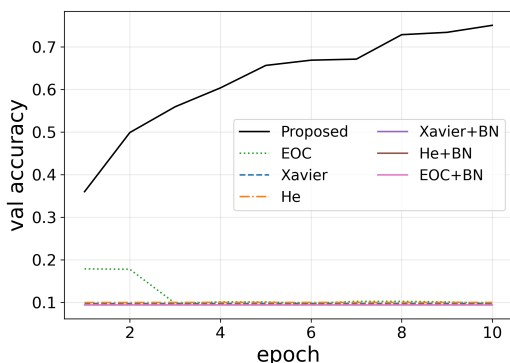

Figure 31: Validation accuracy on Fashion MNIST for a fully connected network with 800 hidden layers of width 32 and $f(x) = \tanh(ax) + \operatorname{erf}(bx) + x/(1 + |cx|) + \operatorname{gd}(dx)$, using coefficients $(a, b, c, d) = (100, 1000, 10, 1000)$.

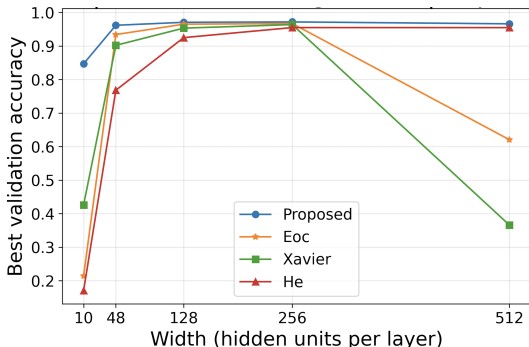

Figure 32: Best validation accuracy on MNIST versus width $\{10, 48, 128, 150, 200, 512\}$ for 100 layer $\tanh$ FFNN. Each curve compares the Proposed, EOC, Xavier, and He initialization schemes, and each point reports the best validation accuracy over 20 training epochs.

## C.3 LEARNABLE LEARNING RATE

Our proposed initialization depends explicitly on the $\omega = 1/f'(0)$, so rescaling the activation also rescales the initialized weights. In particular, when we replace $f$ by a scaled activation $\alpha f(x)$ or $f(\alpha x)$, the corresponding value of $\omega$ changes and the scale of the initial weight matrix is adjusted accordingly. As a consequence, the learning rate that yields comparable gradient updates should also depend on $\omega$. In our experiments we therefore choose $\eta$ from an $\omega$–scaled band (e.g., $\eta \in [10^{-5}\omega, 10^{-3}\omega]$) when using the proposed initialization. Across all settings, the proposed initializer remains trainable over a wider interval of $\eta$ than Gaussian initializations, indicating that it is more robust to the choice of learning rate even when the activation scale varies significantly.

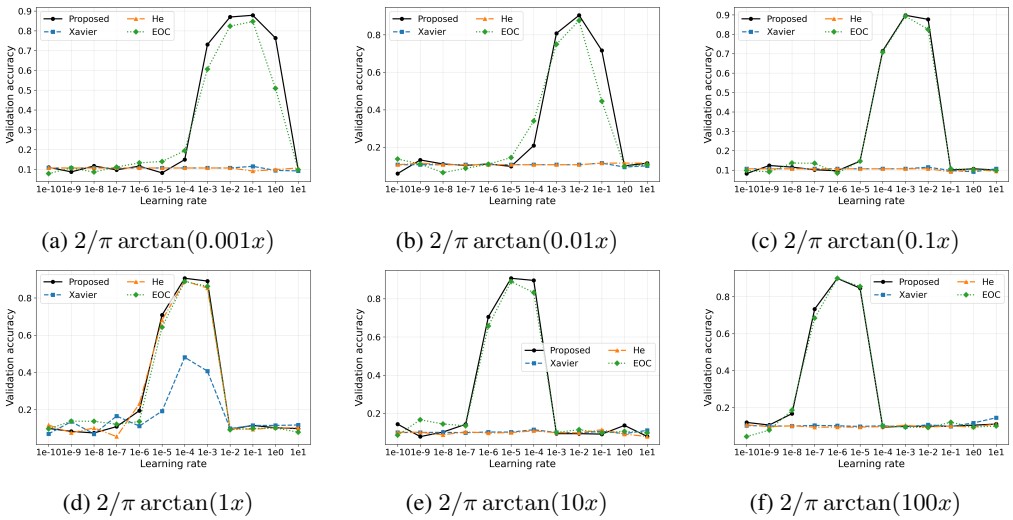

Figure 33: Learning rate accuracy curves on MNIST for a 20 layer, width 512 feedforward network with activation $f(x) = \frac{2}{\pi} \arctan(\alpha x)$. Each panel corresponds to a different activation scale $\alpha \in \{10^2, 10^1, 1, 10^{-1}, 10^{-2}, 10^{-3}\}$. For each learning rate, we train for 200 iterations on a 10k training subset and report the validation accuracy.

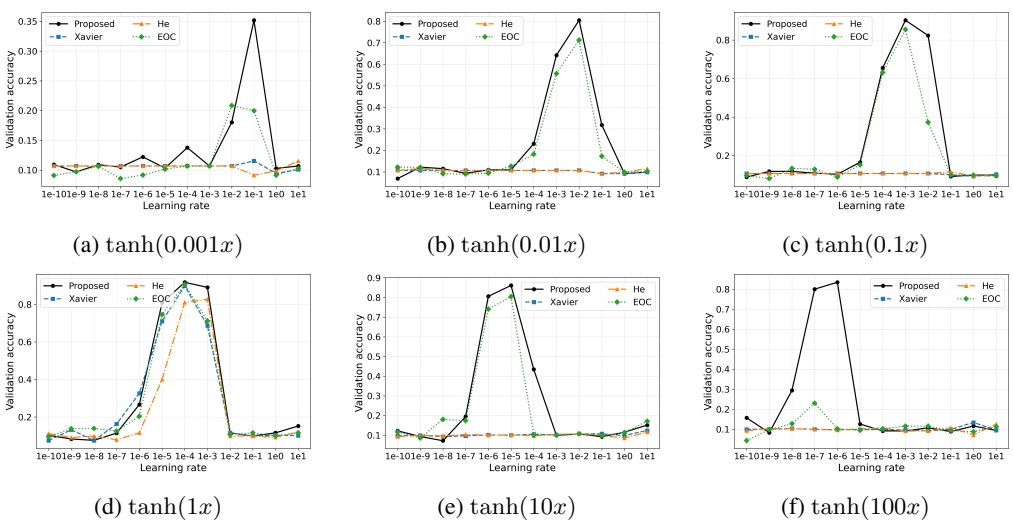

Figure 34: Learning rate accuracy curves on MNIST for a 20 layer, width 512 feedforward network with activation $f(x) = \tanh(\alpha x)$. Each panel corresponds to a different activation scale $\alpha \in \{10^2, 10^1, 1, 10^{-1}, 10^{-2}, 10^{-3}\}$. For each learning rate, we train for 200 iterations on a 1k training subset and report the validation accuracy.

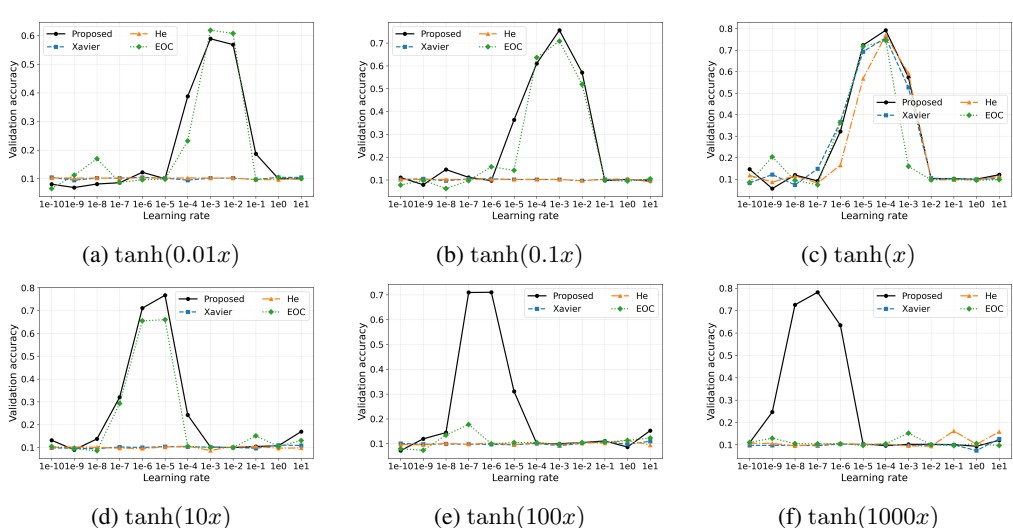

(a) $\tanh(0.01x)$      (b) $\tanh(0.1x)$      (c) $\tanh(x)$

(d) $\tanh(10x)$      (e) $\tanh(100x)$      (f) $\tanh(1000x)$

Figure 35: Learning rate accuracy curves on Fashion MNIST for a 20 layer, width 512 feedforward network with activation $f(x) = \tanh(\alpha x)$. Each panel corresponds to a different activation scale $\alpha \in \{10^3, 10^2, 10^1, 1, 10^{-1}, 10^{-2}\}$. For each learning rate, we train for 200 iterations on a 10k training subset and report the validation accuracy. Curves compare four initializations: Proposed, Xavier, He, and EOC.

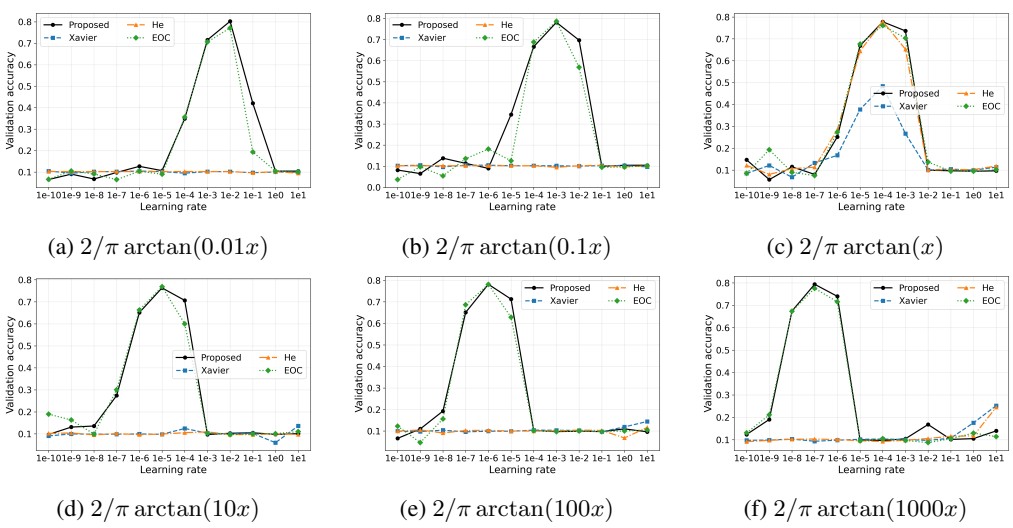

(a) $2/\pi \arctan(0.01x)$      (b) $2/\pi \arctan(0.1x)$      (c) $2/\pi \arctan(x)$

(d) $2/\pi \arctan(10x)$      (e) $2/\pi \arctan(100x)$      (f) $2/\pi \arctan(1000x)$

Figure 36: Learning rate accuracy curves on Fashion MNIST for a 20 layer, width 512 feedforward network with activation $f(x) = 2/\pi \arctan(\alpha x)$. Each panel corresponds to a different activation scale $\alpha \in \{10^3, 10^2, 10^1, 1, 10^{-1}, 10^{-2}\}$. For each learning rate, we train for 200 iterations on a 10k training subset and report the validation accuracy. Curves compare four initializations: Proposed, Xavier, He, and EOC.

## C.4 SCALE PRESERVING ODD–SIGMOID ACTIVATIONS.

Using a scaled activation function amounts to adjusting the effective range of both the input and output axes of the nonlinearity. For example, $\tanh(x)$ has output range $[-1, 1]$, whereas $\alpha \tanh(x)$ has range $[-\alpha, \alpha]$. With our initialization we therefore ask whether signals remain well propagated over the full interval $[-\alpha, \alpha]$ when the activation is scaled. As shown in Figure 37, for several values of $\alpha$ the proposed initialization keeps the last layer activations of $\alpha \tanh(x)$ spread over $[-\alpha, \alpha]$ even in very deep networks (up to $L = 10^5$), while Gaussian i.i.d. initialization rapidly drives the activations to saturate near zero. In all $\alpha$-scale experiments (Tables 2–7), each initializer was given its own LR grid search. Even under this per initializer tuning, Gaussian and EOC schemes fail to train for large $\alpha$, while our method remains stable

Since the proposed initialization lets us target the output scale via the choice of $\alpha$, it is natural to expect benefits on regression tasks where the range of the target $y$ matters. To test this, we evaluate our method in physics informed neural networks (PINNs), using scaled odd–sigmoid activations for Burgers' equation and the Black–Scholes equation. The PINN setup and the precise PDE formulations are described in the following subsections.

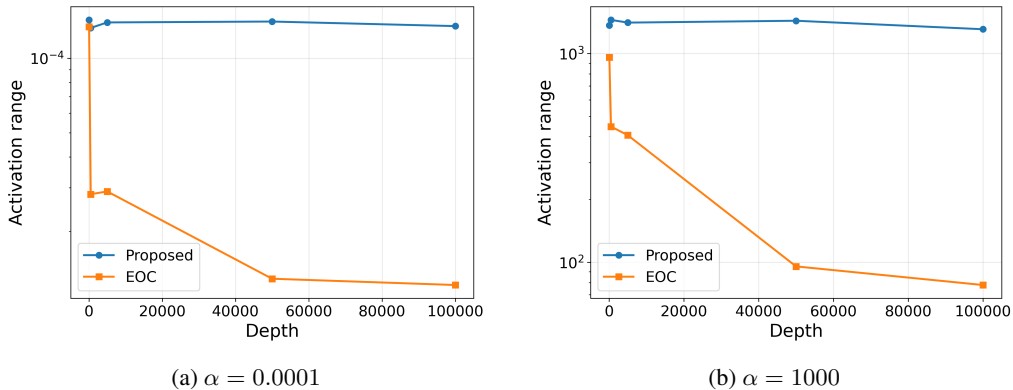

(a) $\alpha = 0.0001$          (b) $\alpha = 1000$

Figure 37: Activation range as a function of depth $L \in \{50, 500, 5{,}000, 50{,}000, 10^5\}$ for fully connected width 64 networks with activation $\alpha \tanh(x)$ under the Proposed and EOC initializations. Panel (a) uses $\alpha = 10^{-4}$ and panel (b) uses $\alpha = 10^3$.

Table 2: Validation accuracy on MNIST (left) and Fashion MNIST (right) for a 50 layer, width 128 fully connected neural network with activation $a \tanh(x)$. Each row corresponds to a different activation scale $a$, and for every $a$ the learning rate is set to $\eta = 10^{-4}/a$ for both initializations.

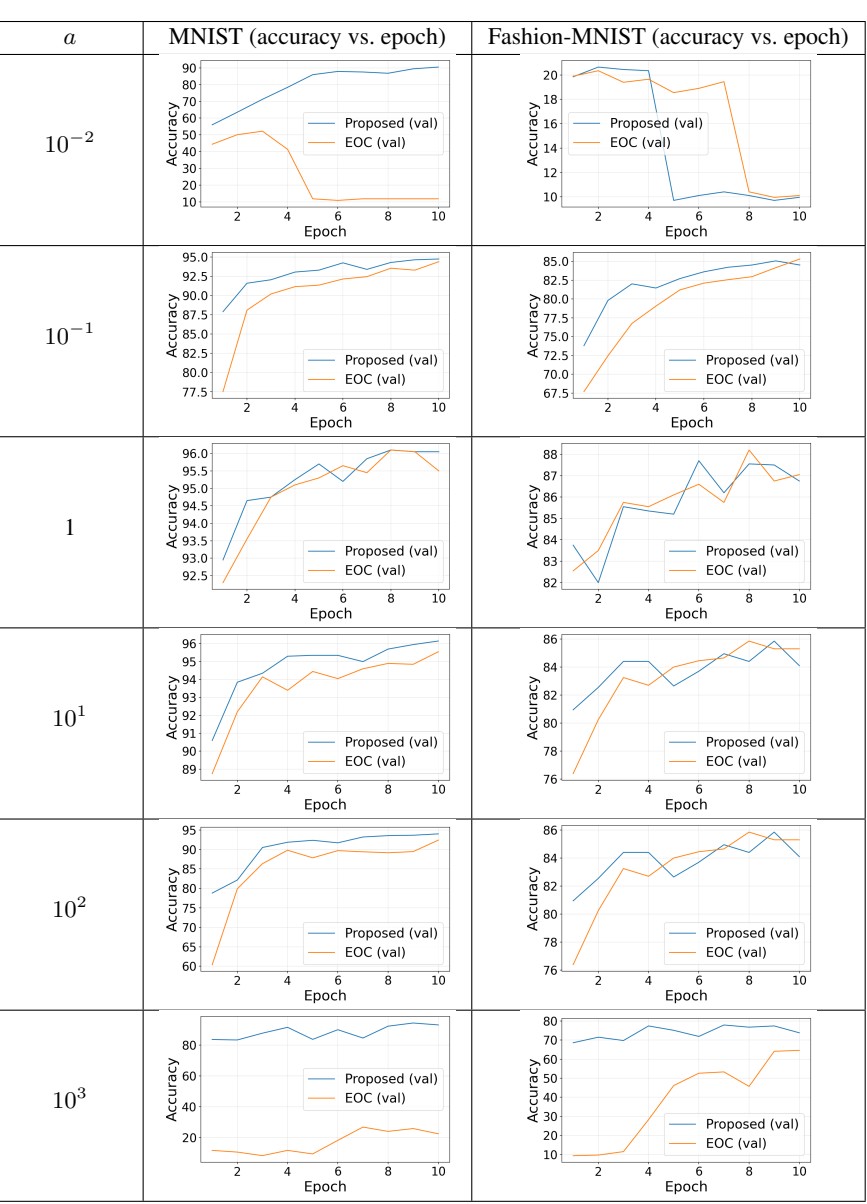

Table 3: Validation accuracy on MNIST (left) and Fashion MNIST (right) for a 50 layer, width 128 fully connected neural network with activation $a \tanh(x)$. Each row corresponds to a different activation scale $a$, and for every $a$ the learning rate is set to $\eta = 10^{-4}/a$ for both initializations.

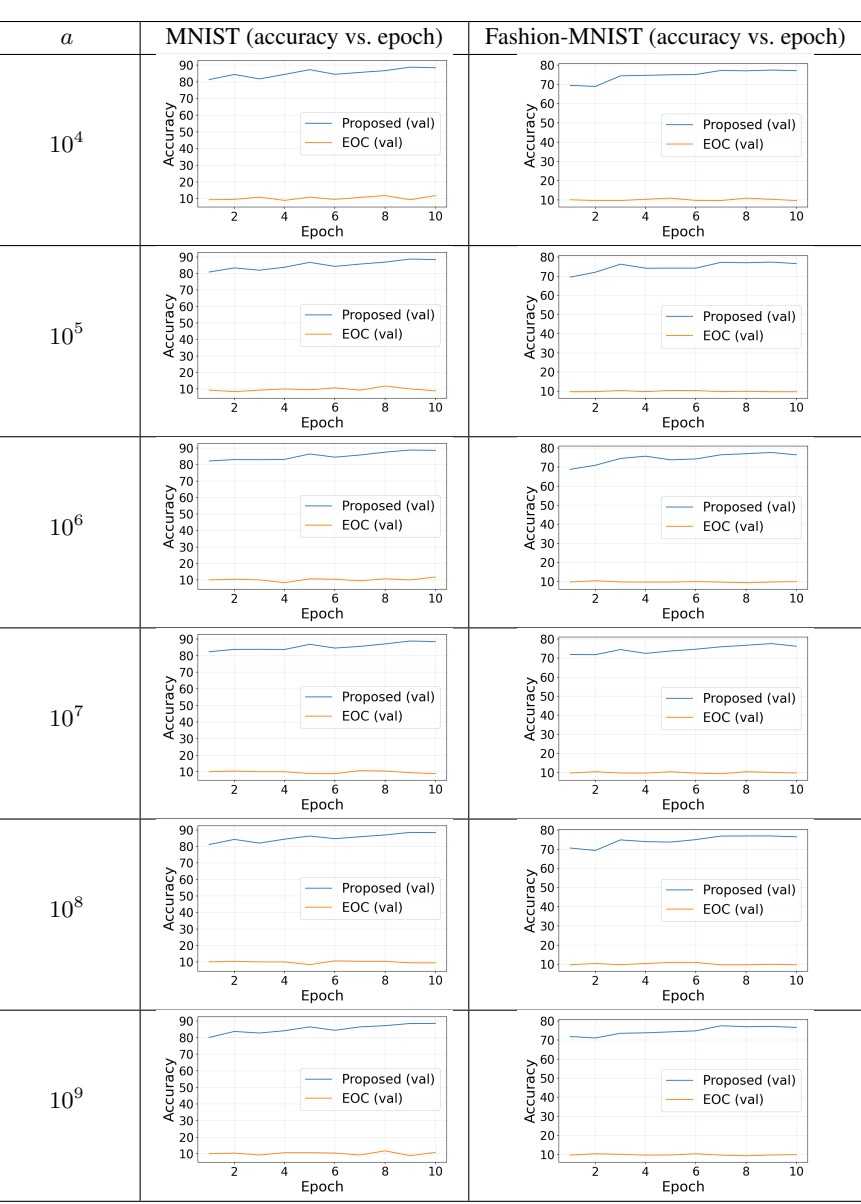

Table 4: Validation accuracy on MNIST (left) and Fashion MNIST (right) for a 50 layer, width 128 fully connected neural network with activation $a \arctan(x)$. Each row corresponds to a different activation scale $a$, and for every $a$ the learning rate is set to $\eta = 10^{-4}/a$ for both initializations.

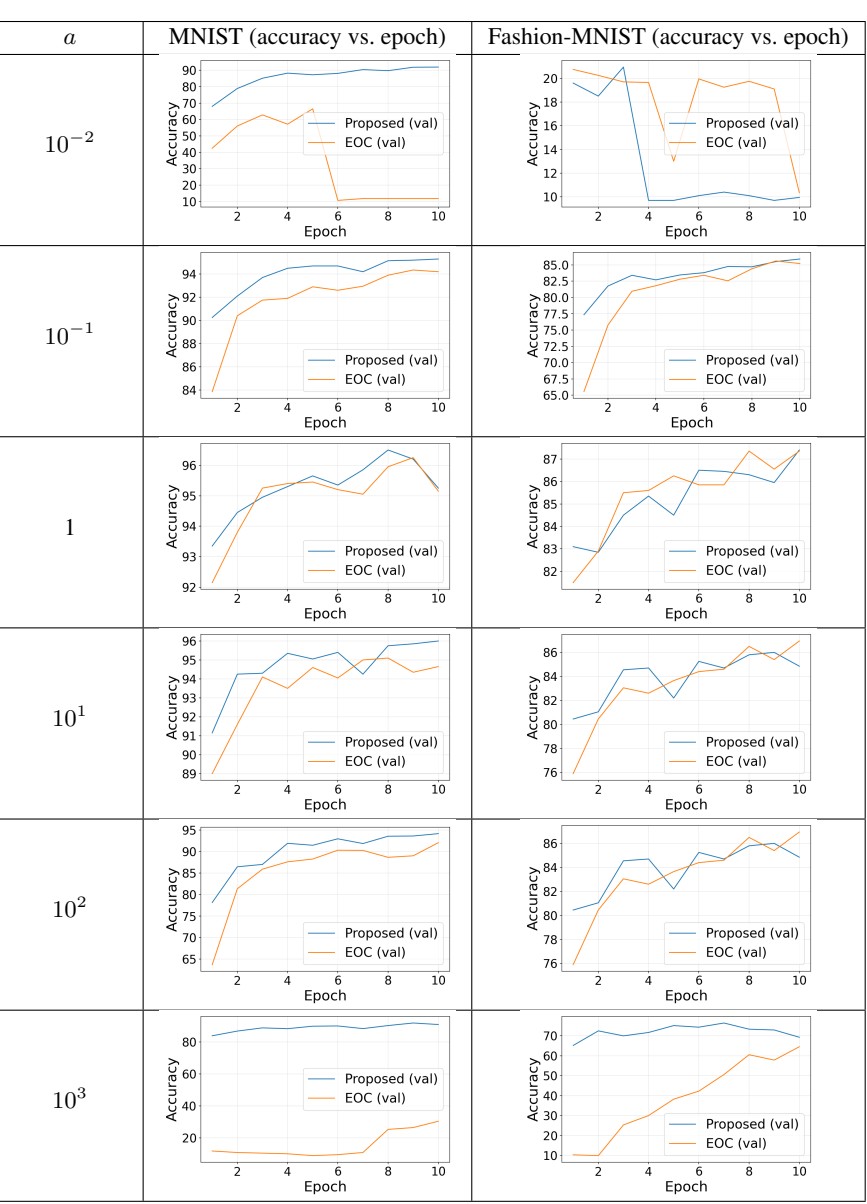

Table 5: Validation accuracy on MNIST (left) and Fashion MNIST (right) for a 50 layer, width 128 fully connected neural network with activation $a \arctan(x)$. Each row corresponds to a different activation scale $a$, and for every $a$ the learning rate is set to $\eta = 10^{-4}/a$ for both initializations.

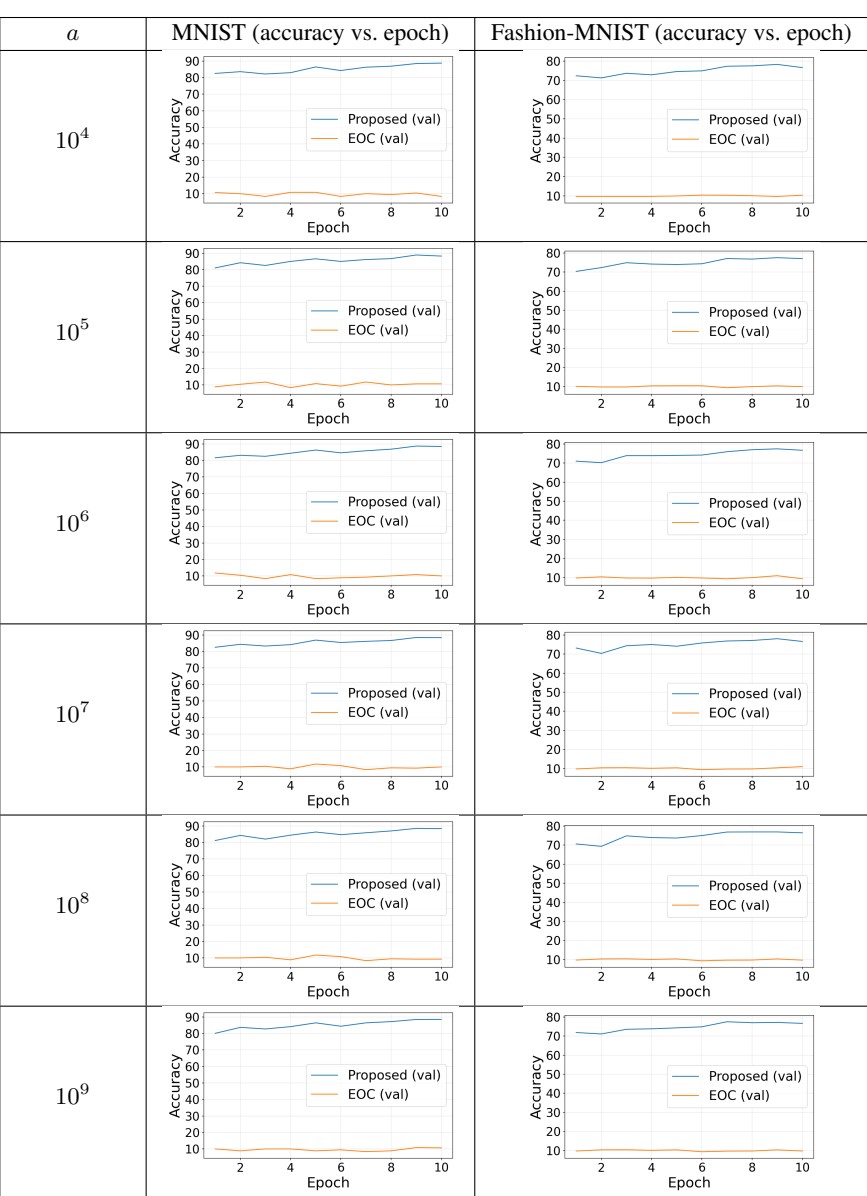

Table 6: Validation accuracy on MNIST (left) and Fashion MNIST (right) for a 50 layer, width 128 fully connected neural network with activation $a\,\mathrm{softsign}_2(x)$. Each row corresponds to a different activation scale $a$, and for every $a$ the learning rate is set to $\eta = 10^{-4}/a$ for both initializations.

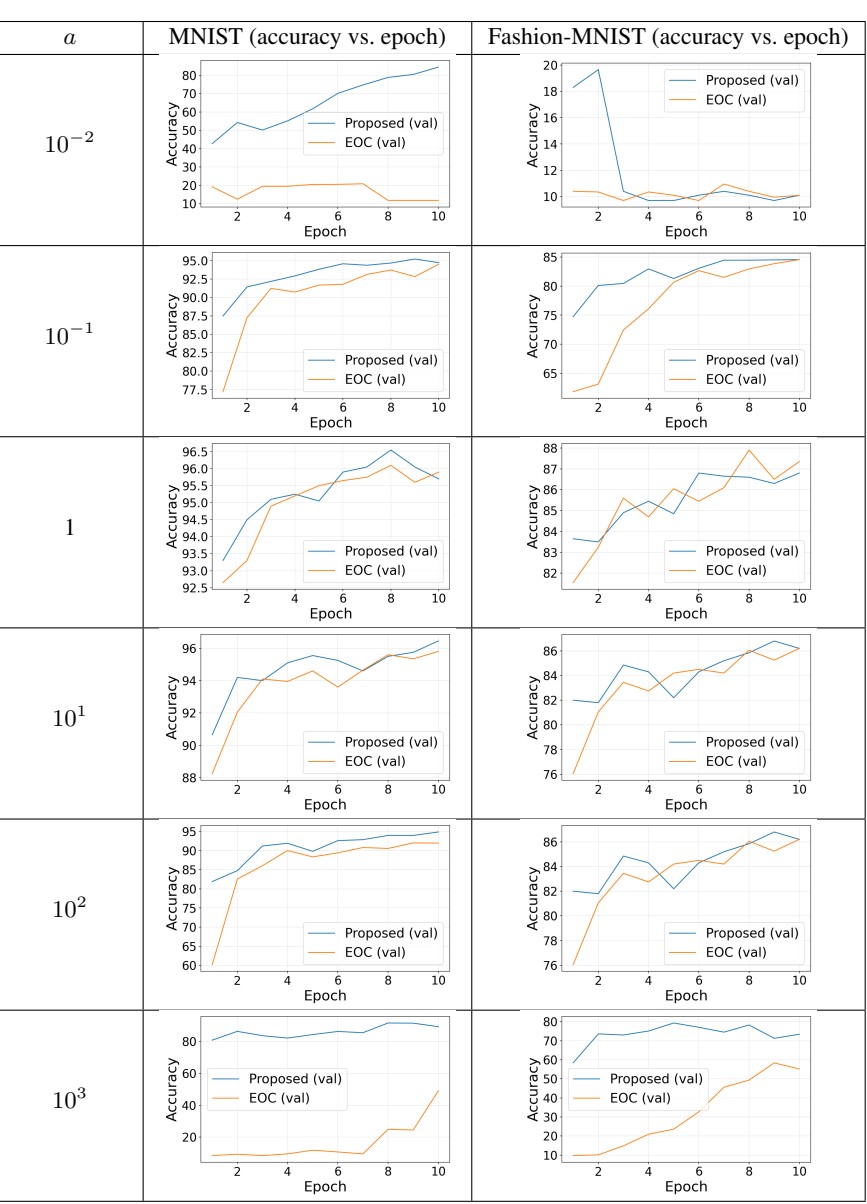

Table 7: Validation accuracy on MNIST (left) and Fashion MNIST (right) for a 50 layer, width 128 fully connected neural network with activation $a\,\mathrm{softsign}_2(x)$. Each row corresponds to a different activation scale $a$, and for every $a$ the learning rate is set to $\eta = 10^{-4}/a$ for both initializations.

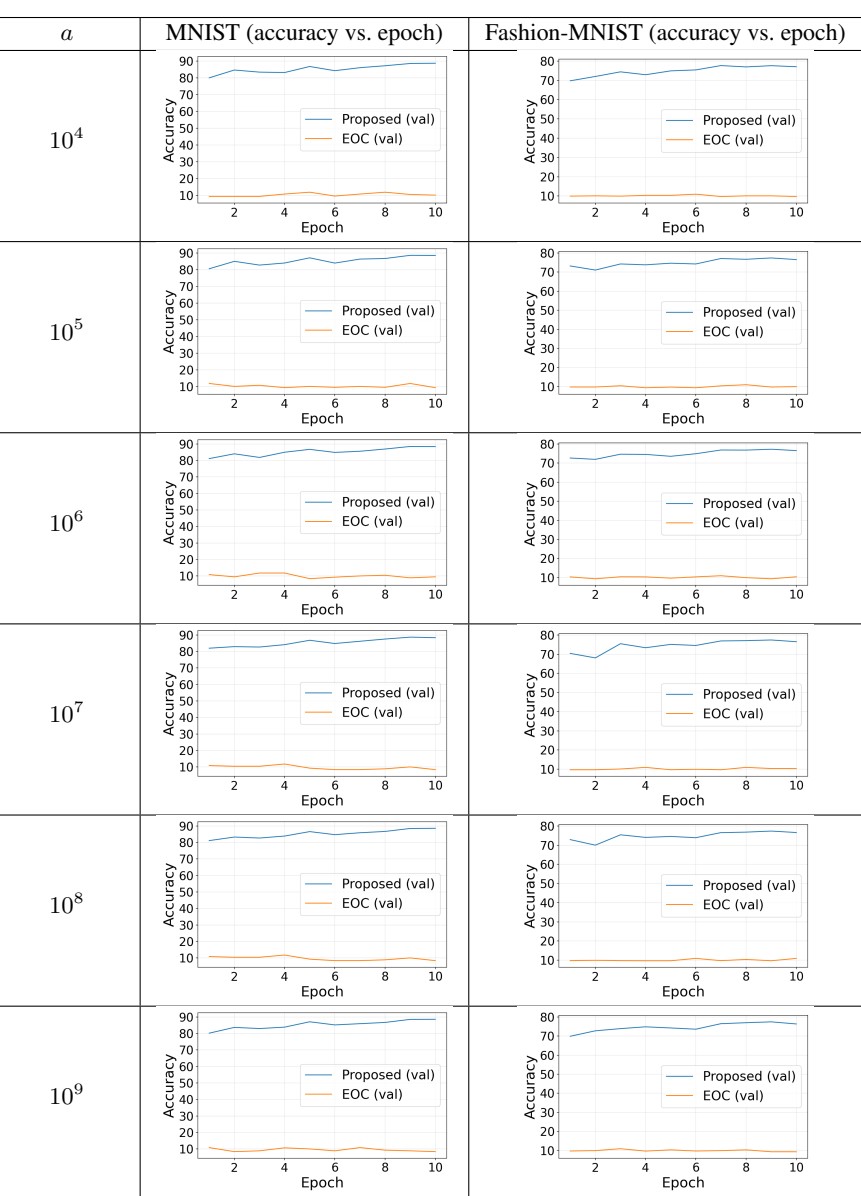

**Common PINN configuration.** For both PDE benchmarks we use fully connected PINNs with input $(\xi_1, \xi_2)$ corresponding to either $(S, \tau)$ or $(x, t)$, 50 hidden layers, and width 64 in every hidden layer. The hidden activation is either $a \tanh(x)$ or $b \, \text{softsign}_2(x) = b \, x/\sqrt{1 + x^2}$, and we compare the proposed initialization against the EOC initialization for scales $a, b \in \{0.1, 0.3, 0.5, 0.8, 1.0, 1.3, 1.5, 1.8, 2.0\}$. For the proposed scheme we set the structured weights using the negative-rate calibration with target $p = 0.49$ at depth $L = 50$, and for EOC we compute $(\sigma_b, \sigma_w)$ using the Gauss–Hermite fixed point equations for the corresponding activation. In all experiments we train with Adam for iterations (600 for Black–Scholes, 600 for Burgers) followed by L-BFGS for iterations (1000 and 1000, respectively). The learning rate is scaled by the local gain of the activation via $\eta = 10^{-4}/f'(0)$.

**Black–Scholes equation.** We consider the Black–Scholes PDE in time-to-maturity form

$$-V_\tau + \tfrac{1}{2}\sigma^2 S^2 V_{SS} + rSV_S - rV = 0,$$

with volatility $\sigma = 0.2$, interest rate $r = 0.05$, asset price $S \in [S_{\min}, S_{\max}] = [0, 1]$, and time-to-maturity $\tau \in [0, T_{\max}] = [0, 1]$. The initial condition at $\tau = 0$ (maturity) is

$$V(S, 0) = \max(S - K, 0), \qquad K = 0.5,$$

and the spatial boundary conditions are

$$V(0, \tau) = 0, \qquad V(S_{\max}, \tau) = S_{\max} - Ke^{-r\tau}.$$

We sample $30,000$ collocation points $(S, \tau)$ uniformly from $[0, 1] \times [0, 1]$, $n_{\text{IC}} = 4,000$ points along $\tau = 0$, and $n_{\text{SB}} = 4,000$ boundary points along $S = 0$ and $S = S_{\max}$.

**Burgers' equation.** For Burgers' equation we solve

$$u_t + u \, u_x - \nu u_{xx} = 0,$$

with viscosity $\nu = 0.01/\pi$, spatial domain $x \in [X_{\min}, X_{\max}] = [-1, 1]$, and time interval $t \in [0, T_{\max}] = [0, 1]$. The initial condition is

$$u(x, 0) = -\sin(\pi x),$$

and we impose homogeneous Dirichlet boundary conditions

$$u(X_{\min}, t) = 0, \qquad u(X_{\max}, t) = 0.$$

We draw $20,000$ collocation points $(x, t)$ uniformly from $[-1, 1] \times [0, 1]$, use $n_{\text{IC}} = 1,000$ points along $t = 0$, and $n_{\text{SB}} = 1,000$ boundary points along $x = X_{\min}$ and $x = X_{\max}$.

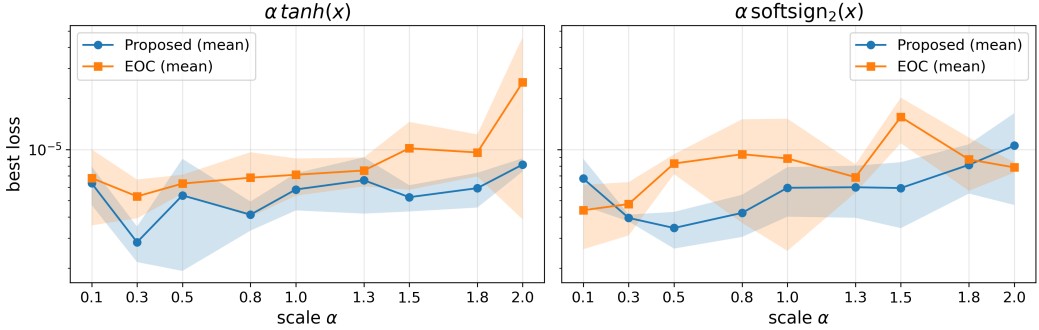

Figure 38: Best PINN loss versus activation scale for the Burgers PINN (depth 50, width 64), comparing the proposed and EOC initializations. Left: $a \tanh(x)$ with $a \in \{0.1, 0.3, 0.5, 0.8, 1.0, 1.3, 1.5, 1.8, 2.0\}$. Right: $b \, \text{softsign}_2(x)$ with the same set of scales.

