# OpenReview forum: "Signal Preserving Weight Initialization for Odd-Sigmoid Activations"
_ICLR.cc/2026/Conference — Submitted to ICLR 2026_

### Official Review · Reviewer_Wx7z · 2025-10-28

**Soundness:** 2
**Presentation:** 2
**Contribution:** 2
**Rating:** 2
**Confidence:** 4

**Summary:**

This paper studies the initialization of neural networks and proposes a new strategy of initialization when using sigmoid-like activation functions, denoted by $f$. The entire study is based on the fixed points of $f$: when $f$ belongs to a given set of sigmoid-like functions, there exists a critical input scaling $\omega > 0$ such that $x \mapsto f(a x)$ has only one fixed point ($0$) when $a < \omega$, and three fixed points ($0, \xi_a, -\xi_a$) when $a > \omega$.

This paper proposes an initialization strategy based on this critical number $\omega$, which depends on the shape of $f$, which is meant to achieve several goals:
1. with well-chosen input scalings ($a > \omega$), the pre-activations are guaranteed not to converge to 0 in probability (Theorem 4.6);
2. the outputs of the network meet a specific requirement (Theorem 4.7).

Finally, some experiments in simple cases (MNIST, Fashion-MNIST) show the superiority of this initialization strategy compared to Xavier, He and Orthogonal.

**Strengths:**

The main idea is easy to follow.

It is still interesting to design initialization distributions for specific activation functions. Moreover, a complete study of the fixed-point distribution obtained after a large number of layers is still missing in the initialization of neural networks literature.

The proposed initialization distribution can be easily used.

**Weaknesses:**

## Motivation
**Discussion about former works.** The paper states that it proposes a method that "keeps activation distributions well dispersed across layers, mitigating collapse to zero or saturation." The paper should provide evidence that former initialization strategies fail to address these issues. At least, the paper must explain the limitations of former initialization strategies. Please note that the "Edge of Chaos" works [2, 3] (cited in the paper) already address the issues of "collapse to zero" or "saturation".

## Clarity
This paper contains several notations and terms that are not properly defined. This is a major issue, since it prevents the reader to fully understand some parts of the paper, for instance Theorem 4.7.

**Negative rate $p$.** I did not understand what this "negative rate" is and why it is so important. I only understand that it is related to the output of the model.

**Scalar surrogate $\pi_L$.** Same issue: what is it? What is the link with calibration?

**[Non-critical] Variance $\sigma_z$.** What is it? It seems that this notation comes from [1], but it is not defined in this paper (it should be removed...).

**[Non-critical] Functions $\Phi_m, \Phi_m^{\alpha}$.** These notations should be replaced with $\Phi^m$ and $\Phi^m_{\alpha}$, since they consist of functions $\phi_a$ combined $m$ times.

**Non-readable figures.** Figure 2 is very difficult to read: the text is too small.

## Significance of Theorem 4.6
I do not understand why Theorem 4.6 is significant whatsoever. It seems that the goal is to show (ii), stating that the probability that the model outputs sth. away from $0$ is bounded by below. However:
1. the "gains" $(a_1, \cdots, a_m)$ are related to the outputs of the neurons of the layers $(1, \cdots, m)$, so the assumption that $(a_1, \cdots, a_m)$ are Gaussian and independent does not hold (even if $a_i$ is Gaussian conditionally to the inputs of layer $i$);
2. the neural network is assumed to have $N_l = 1$ neuron per layer, which is obviously not true.

Overall, the restricted hypotheses of Theorem 4.6 are not discussed: we do not know if they are actually restrictive, and we do not know if the claims hold in practice (with dependent activations and several neurons per layer).

## Preceding work
This papers is entirely based on [1]: same notation, same basic ideas (e.g., rewriting of the propagation used in Eqn. (1)), and same focus on the fixed points. While [1] focuses on $\tanh$, this paper extends the study on [1] to a set of activation functions. This inspiration from [1] is entirely endorsed in the current paper, and there is nothing wrong about that.

However, the significance of the current paper compared to [1] can be discussed. As such, the initialization strategy proposed here is based on only two new contributions:
1. the study of $\tanh$ is extended to a set of $\tanh$-like functions;
2. the tuning of the input scaling is now founded on a heuristic (and not a numerical estimation), which allows us to compute it analytically.

The significance of Theorem 4.6 the clarity of Theorem 4.7 are discussed above.

## References

[1] Robust weight initialization for tanh neural networks with fixed point analysis, Lee et al., ICLR 2024.

[2] Exponential expressivity in deep neural networks through transient chaos, Poole et al., NeurIPS 2016.

[3] Deep information propagation, Schoenholz et al., ICLR 2016.

**Questions:**

What are the limitations of preceding works (e.g., Edge of Chaos)? Please be precise.

Could you clarify the "negative rate $p$", "$\pi_L$" and Theorem 4.7?

Discuss the hypotheses and the conclusion of Theorem 4.6. Limitation? Possibility to loosen the hypotheses? Does the conclusion hold in practice?

---

> ### Author Response · Authors · 2025-12-03
> **Official Comment by Authors**
>
> Thank you for your valuable comments! In the following section, we will address the weaknesses (W) and questions (Q) mentioned above. **The changes are marked in blue**.
>
> **W1** *"The paper states that it proposes a method that "keeps activation distributions well dispersed across layers, mitigating collapse to zero or saturation." The paper should provide evidence that former initialization strategies fail to address these issues. At least, the paper must explain the limitations of former initialization strategies. Please note that the "Edge of Chaos" works [2, 3] (cited in the paper) already address the issues of "collapse to zero" or "saturation""*
>
> A1. Thank you for this valuable suggestion. The existing results on signal and variance preservation (e.g., Poole et al. 2016; Schoenholz et al. 2016; Hayou et al. 2019) are largely derived under Gaussian initialization and typically assume wide or even infinite-width networks. Consequently, these theories do not fully explain or support training in the deep-and-narrow regimes that we target. In contrast, our approach does not rely on classical mean-field signal-propagation analyses: we reinterpret the network as a collection of parallel scalar dynamical systems, which leads to an initialization that propagates signals robustly even in deep and narrow architectures. Motivated by the reviewer’s valuable comments, we have revised the manuscript to more clearly emphasize, both theoretically and empirically, how our proposed method differs from standard Gaussian initializations, as follows:
> - We **revised Section 4.2 and added a new Section 4.3** to make it explicit that our proposed method offers a perspective distinct from classical signal-propagation theory based on Gaussian initializations.
> - We also **added Figures 3 and 19–23** to empirically demonstrate that our initialization is more robust than Gaussian initializations for forward signal propagation across different network sizes, and **Figure 4** to show that it better preserves backward signal propagation (i.e., gradient scales) at initialization.
>
> **W2** *"Clarity""*
>
> A1. Thank you for this valuable suggestion. In response to the reviewers' comments, we have revised the manuscript as follows:
> - We now define the negative rate explicitly in **Section 4.2** and provide additional motivation and derivations for this quantity in **Section B.4**.
>
> - We clarify the **exponential relationship between the surrogate model and the real model**, and we show that the scalar surrogate chain can be used to approximate the negative rate of the full FFNN in **Section B.4**.
>
> - We explicitly **introduce the notation $\sigma_z$** in the ``Proposed weight initialization'' part of Section 4.2.
>
> - We have **revised the notation** $\Phi_m$ to $\Phi^m$ (and $\Phi_m^{\alpha}$ to $\Phi^m_{\alpha}$).
>
> - We **recreated all figures with overly small notation** (including Figure 2) to improve readability.
>
> **W2** *"Significance of Theorem 4.6""*
>
> **Q3** *"Discuss the hypotheses and the conclusion of Theorem 4.6. Limitation? Possibility to loosen the hypotheses? Does the conclusion hold in practice?"*
>
>
> A3. Thank you for this valuable suggestion. We view the FFNN as a collection of parallel scalar dynamical systems, so that each coordinate evolves as $x^{\ell+1}_i = f\bigl(a_i^{\ell} x_i^{\ell}\bigr)$
>
> In our surrogate analysis, we idealize the gains $a_i^{\ell}$ as i.i.d. and work with a scalar chain ($N_\ell = 1$) to target a desired negative rate at depth $L$ and obtain a tractable closed-form calibration. We then show that the negative rate targeted in this scalar surrogate is exponentially related to the negative rate of the full FFNN-based scalar chain, which justifies using the surrogate to design the initialization. In response to the reviewers' comments, we have revised the manuscript as follows:
>
> - We **added Section B.4 to detail this motivation and derivation**, including why it is natural to assume $N_{\ell} = 1$ and data-independent gains in the surrogate model. We also **updated Theorem 4.6** to a more general formulation that keeps the same implication.

---

> > ### Author Response · Authors · 2025-12-03
> > **Official Comment by Authors**
> >
> > .**W3** *"Preceding work""*
> >
> > A4. Thank you for this valuable suggestion.  In response to the reviewers' comments, we have revised the manuscript as follows:
> >
> > - We **added Section 4.3** to clearly distinguish our proposed diagonal initialization from standard Gaussian based initializations.
> >
> > - **In Section 4.3**, we show from a mean-field perspective that Gaussian initializations are much more sensitive to the noise scale $\sigma_z$ and network size than our proposed scheme, whose behavior is controlled by the diagonal parameter $\omega$. Experimentally, **Figure 5** confirms this difference in robustness.
> >
> > - We further **demonstrate that the negative-rate based calibration of our initialization has a strong correlation between the scalar surrogate model and the actual FFNN**, justifying the use of the surrogate for designing the initialization (Section B.4).
> >
> > - To highlight the importance of preserving nontrivial output scales in regression settings, we **apply the proposed initialization to PINN tasks**, showing stable performance across a wide range of activation scales **(Section C.4)**.
> >
> > - We also show that activations that are typically very hard to train, such as $a \tanh(x)$ with $a = 10^9$, become trainable under our initialization, whereas Gaussian-based initializations fail in this extreme regime **(Section C.4)**.
> >
> > .**Q1** *"What are the limitations of preceding works (e.g., Edge of Chaos)? Please be precise.""*
> >
> > A5. Thank you for this valuable suggestion. The existing results on signal and variance preservation (e.g., Poole et al 2016; Schoenholz et al. 2016; Hayou et al. 2019) are largely derived under Gaussian initialization and typically assume wide or even infinite-width networks. As a result, these theories do not adequately explain or support training when the network is narrow and very deep. In contrast, our approach does not rely on these classical signal-propagation results: we reinterpret the network as a collection of parallel scalar dynamical systems, which allows us to design initializations that propagate signals robustly even in deep and narrow architectures. Motivated by the reviewer’s valuable comments, we have revised the manuscript to more clearly highlight, both theoretically and empirically, the differences between our proposed method and standard Gaussian initializations, as follows:
> >
> > - We **revised Section 4.2 and added a new Section 4.3** to make it explicit that our proposed method provides a perspective distinct from classical signal-propagation theory based on Gaussian initializations.
> > - We additionally **added Figures 3, 19–23** to empirically demonstrate that our initialization is more robust than Gaussian initializations for forward signal propagation across different network sizes, and **Figure 4** to show that it better preserves backward signal propagation (i.e., gradient scales) at initialization.
> >
> > .**Q2** *"Clarity Theorem 4.7""*
> >
> > A5. Thank you for this valuable suggestion. In response to the reviewers' comments, we have revised the manuscript as follows:
> > - We have **added Section B.4 to clarify the motivation** for the negative rate, to detail its derivation, and to explain the role of the surrogate model in our negative-rate based calibration.

---

### Official Review · Reviewer_4CNB · 2025-10-31

**Soundness:** 2
**Presentation:** 2
**Contribution:** 2
**Rating:** 4
**Confidence:** 2

**Summary:**

This paper proposes a weight initialization method for feed-forward neural networks (FFNNs) whose activation functions belong to a class referred to as odd-sigmoid functions.
An odd-sigmoid activation function denotes a monotonic, origin-symmetric nonlinearity such as tanh.
Building upon the analysis of fixed points of activation functions (similar to the discussion in Lee et al., 2025), the authors propose a weight initialization strategy that ensures the variance of activations in a layer remains above a certain threshold.
Experiments on MNIST and Fashion-MNIST demonstrate that the proposed initialization leads to higher accuracy during the early training epochs compared to conventional initialization schemes.

**Strengths:**

* The paper introduces a novel weight initialization method grounded in the fixed-point analysis of activation functions.
* It provides a theoretical analysis of the variance of activations in FFNNs initialized with the proposed scheme.

**Weaknesses:**

* Some theoretical assumptions discussed in the paper seem to deviate from realistic conditions (see the Questions section below).
* The proposed method is not applicable to activation functions that are not origin-symmetric, such as ReLU.
* The evaluation metrics used in the experiments differ from common benchmarks, making it unclear how practically useful the proposed approach is.

**Questions:**

* l.057: The paper states that “the proposed initialization reduces reliance on normalization layers such as Batch Normalization, thereby lowering the burden of hyperparameter tuning (e.g., depth/width selection).”
  However, it is unclear how the absence of normalization layers is related to reduced dependency on depth or width selection.

* In Proposition A.2 and Corollary A.3, the coefficient $a_n$ is assumed to be positive.
  However, according to Eq. (1), this assumption does not seem to hold in general. Could the authors clarify this point?

* In Theorem 4.6, the assumption $N_\ell = 1$ is introduced.
  Does this imply that the width of every layer in the FFNN is set to one?
  If so, what general insights can be drawn from this theorem for more realistic FFNNs with $N_\ell > 1$?

* What does the term “width proxy” in Figure 4 represent?

* What learning rate was used for the Adam optimizer?

* Regarding Table 1:

  * Since the initialization involves random sampling, the results should be reported as averages and standard deviations over multiple random seeds.
  * Why are only the first five epochs evaluated? It would be more informative to include results over a standard number of training epochs.

* Do “MLP” and “FFNN” refer to the same architecture throughout the paper?

* For Table 2: Training for only ten epochs with extremely limited data (e.g., 100 or 500 samples) seems unrealistic.
  What kind of practical implication can be drawn from such results?

* In Figure 5, the authors consider cases where even models with Batch Normalization fail to train properly.
  Therefore, the claim that the proposed method enables BN-free training does not appear to be justified based on these results.

---

> ### Author Response · Authors · 2025-12-03
> **Official Comment by Authors**
>
> Thank you for your valuable comments! In the following section, we will address the weaknesses (W) and questions (Q) mentioned above. **The changes are marked in blue**.
>
> **W1.** *"Some theoretical assumptions discussed in the paper seem to deviate from realistic conditions (see the Questions section below)."*
>
> **Q2.** *"In Proposition A.2 and Corollary A.3, the coefficient
>  is assumed to be positive. However, according to Eq. (1), this assumption does not seem to hold in general. Could the authors clarify this point?"*
>
> **Q3.** *"In Theorem 4.6, the assumption $N_\ell = 1$ is introduced. Does this imply that the width of every layer in the FFNN is set to one? If so, what general insights can be drawn from this theorem for more realistic FFNNs with $N_\ell > 1$?"*
>
> A1. Thank you for this valuable suggestion. We address W1, Q2 and Q3 together. In response to the reviewers' comments, we have revised the manuscript as follows:
> - We view the FFNN as a collection of parallel scalar dynamical systems and calibrate the noise scale $\sigma_z$ of our proposed initialization via the sign-flip probability of a single coordinate at layer $L$. For this reason, the analysis under the assumption $N_\ell = 1$ is sufficient: it captures the behavior of an arbitrary coordinate and serves as a scalar surrogate for the full network. We have **clarified this motivation, the detailed derivation, and the exponential relationship between the surrogate recursion and the actual multi-dimensional model in Section B.4.**
> ---
>
> **W2.** *"The proposed method is not applicable to activation functions that are not origin-symmetric, such as ReLU."*
>
> A2. Thank you for this valuable suggestion. Initialization schemes and activation functions are interdependent [1]. In this work, we therefore restrict our scope to weight initialization for sigmoidal (in particular, odd–sigmoid) activations, and do not address initialization for ReLU type activations.
>
> ---
>
> **W3.** *"The evaluation metrics used in the experiments differ from common benchmarks, making it unclear how practically useful the proposed approach is"*
>
> A3. Most classification experiments in our work follow the standard practice of evaluating performance on benchmark datasets using validation accuracy [1, 2, 3].
>
> ---
>
> **Q1.** *"The paper states that “the proposed initialization reduces reliance on normalization layers such as Batch Normalization, thereby lowering the burden of hyperparameter tuning (e.g., depth/width selection).” However, it is unclear how the absence of normalization layers is related to reduced dependency on depth or width selection"*
>
> A4. Thank you for this valuable suggestion. Batch Normalization is known to help train deep networks but adds extra computation and hyperparameter tuning. Our point is that, if a deep network can be trained stably without Batch Normalization under the proposed initialization, then the initialization is less sensitive to depth/width and reduces reliance on normalization layers.
>
> - We **added experiments with and without Batch Normalization (Section C.2)**, including very deep networks **(up to 800 layers)** with scaled odd–sigmoid activations, where our initialization still trains reliably.
>
> ---
>
> **Q4.** *"What does the term “width proxy” in Figure 4 represent?"*
>
> A5. Thank you for this valuable suggestion. In response to the reviewers' comments, we have revised the manuscript as follows:
> -  We originally used the label “width proxy” to denote network width, but we realized this could be confusing. We therefore removed this term and replaced the figure with a revised version that uses explicit width notation.
>
> ---
>
> **Q5.** *"What learning rate was used for the Adam optimizer?"*
>
> A6. We describe in Section 4.2 (Proposed weight initialization) how to choose the initial learning rate as a function of the diagonal scale $\omega$, and in Section C we fix the learning rate to $10^{-4}\omega$ in all experiments.
>
> ---
>
> **Q6.** *"Regarding Table 1"*
>
> **Q7,** *"Do “MLP” and “FFNN” refer to the same architecture throughout the paper?"*
>
> A7. Thank you for this valuable suggestion. We address Q6 and Q7 together. We confirm that “MLP” and “FFNN” refer to the same fully connected architecture throughout the paper. In response to the reviewers' comments, we have revised the manuscript as follows:
>
> - We performed **10 independent runs** for the experiments reported in **Table 1** and **5 independent runs** for those in **Figures 7, 14, 15, 16, 17, and 38**.
> - We have additionally included results trained for a wider range of epochs (beyond 5 epochs, up to 50 epochs) **(Section C.1, C.2, C.4 and Table 1)**.

---

> ### Author Response · Authors · 2025-12-03
> **Official Comment by Authors**
>
> **Q8** *"For Table 2: Training for only ten epochs with extremely limited data (e.g., 100 or 500 samples) seems unrealistic. What kind of practical implication can be drawn from such results?"*
>
> A8. Thank you for this valuable suggestion. We consider data efficiency an important practical criterion: if an initialization can successfully learn from very limited data, this indicates that it preserves informative features of the input distribution. For this reason, we included experiments **(Table 1)** in low-data regimes, and we have found that, across various activations and Gaussian-based baselines, our proposed initialization consistently achieves the best performance under scarce data, demonstrating superior data efficiency.
>
> ---
>
> **Q9** *"In Figure 5, the authors consider cases where even models with Batch Normalization fail to train properly. Therefore, the claim that the proposed method enables BN-free training does not appear to be justified based on these results."*
>
> A9. Thank you for this valuable suggestion. We have **added experiments with Batch Normalization (Section C.2)**. Across all these settings, the proposed initialization achieves strong performance both without Batch Normalization and when compared against Gaussian/EOC baselines with or without Batch Normalization. This shows that our method can train deep and narrow networks, even under very small or very large activation scales, without relying on BN.
>
>
>
>
>
> [1] He, Kaiming, et al. "Delving deep into rectifiers: Surpassing human-level performance on imagenet classification." Proceedings of the IEEE international conference on computer vision. 2015.
>
> [2] Trockman, Asher, and J. Zico Kolter. "Mimetic initialization of self-attention layers." International Conference on Machine Learning. PMLR, 2023.
>
> [3] Zhao, Jiawei, Florian Schäfer, and Anima Anandkumar. "Zero initialization: Initializing neural networks with only zeros and ones." arXiv preprint arXiv:2110.12661 (2021).

---

### Official Review · Reviewer_iEW8 · 2025-10-31

**Soundness:** 2
**Presentation:** 2
**Contribution:** 1
**Rating:** 2
**Confidence:** 4

**Summary:**

The paper proposes a weight initialization method designed for a special class of activation functions called odd–sigmoids. These are functions that are smooth, bounded, strictly increasing and odd. The goal of this initialization is to ensure signal preservation: as activations pass through deep layers, they neither collapse to zero nor saturate, without needing normalization layers like BatchNorm.

**Strengths:**

The exposition seems quite clear and accessible, the derivations are straightforward, the notation consistent and the ideas and goals quite clear.

**Weaknesses:**

- The theoretical contribution seem to heavily overlap with results on signal and variance preservation (e.g., Poole et al 2016; Schoenholz et al. 2016; Hayou et al.  2019) within a subclass of activations.

- The paper seems to only studY forward signal propagation and ignores gradients, Jacobian conditioning, or correlation dynamics, see for example the work by Pennington et al.

- The experiments are limited to MNIST and Fashion-MNIST with vanilla MLPs and quite small depths.  Reported improvements over Xavier, He, or Orthogonal initialization seem modest and could lie within statistical variation?

- From a theory perspective comparison with a number of prior works appears to be missing, particularly Hayou et al. (2019) and Murray et al. (2021) where there appears to be a clear conceptual overlap with regard to analyzing the role of the activation.


Links to some of the papers mentioned.
- https://openreview.net/pdf?id=H1W1UN9gg
- https://proceedings.neurips.cc/paper_files/paper/2017/file/d9fc0cdb67638d50f411432d0d41d0ba-Paper.pdf
- https://proceedings.mlr.press/v97/hayou19a.html
- https://www.sciencedirect.com/science/article/pii/S1063520321001111

**Questions:**

- It is not clear to me what the analysis of the recursion $x_{n+1} = f(a_n x_n)$ and the ``pitchfork bifurcation'' add beyond established initialization principles such as initialization on the edge of chaos? How does this proposed framework differentiate itself from prior `edge-of-chaos or dynamical isometry analyses?

- Can you contrast and compare your theoretical results with Hayou et al. (2019) and Murray et al. (2021)?

**Details Of Ethics Concerns:**

Not applicable.

---

> ### Author Response · Authors · 2025-12-03
> **Official Comment by Authors**
>
> Thank you for your valuable comments! In the following section, we will address the weaknesses (W) and questions (Q) mentioned above. **The changes are marked in blue**.
>
> **W1.** *"The theoretical contribution seem to heavily overlap with results on signal and variance preservation (e.g., Poole et al 2016; Schoenholz et al. 2016; Hayou et al. 2019) within a subclass of activations."*
>
> A1. Thank you for this valuable suggestion. The existing results on signal and variance preservation (e.g., Poole et al 2016; Schoenholz et al. 2016; Hayou et al. 2019)   are largely derived under Gaussian initialization and typically assume wide or even infinite-width networks. As a result, these theories do not adequately explain or support training when the network is narrow and very deep. In contrast, our approach does not rely on these classical signal-propagation results: we reinterpret the network as a collection of parallel scalar dynamical systems, which allows us to design initializations that propagate signals robustly even in deep and narrow architectures. Motivated by the reviewer’s valuable comments, we have revised the manuscript to more clearly highlight, both theoretically and empirically, the differences between our proposed method and standard Gaussian initializations, as follows:
>
> - We **revised Section 4.2** and added **a new Section 4.3** to make it explicit that our proposed method provides a perspective distinct from classical signal-propagation theory based on Gaussian initializations.
> - We additionally **added Figures 3, 19–23** to empirically demonstrate that our initialization is more robust than Gaussian initializations for forward signal propagation across different network sizes, and **Figure 4** to show that it better preserves backward signal propagation (i.e., gradient scales) at initialization.
> ---
> **W2** *"The paper seems to only studY forward signal propagation and ignores gradients, Jacobian conditioning, or correlation dynamics, see for example the work by Pennington et al."*
>
> A2. Thank you for this valuable suggestion. In response to the reviewers' comments, we have revised the manuscript as follows:
>
> - We added **Figure 4** to show that our initialization better preserves backward signal propagation, i.e., gradient scales, at initialization.
> - We have newly **added Section A.3**, where we compute the layerwise backpropagation factor $\chi_\ell$ under a mean-field approximation [1], and **Figure 24**, which shows that our initialization keeps $\chi_\ell$ closer to $1$ over a much wider region of depths $L$ and noise scales $\sigma_z$ than Gaussian initializations. Consistently, we also **added Figure 5**, which demonstrates that, across a range of activation scales, the proposed initialization enables successful training over a wider range of depths and noise levels than Gaussian initialization.
> ---
> **W3** *"The experiments are limited to MNIST and Fashion-MNIST with vanilla MLPs and quite small depths. Reported improvements over Xavier, He, or Orthogonal initialization seem modest and could lie within statistical variation?"*
>
> A3. Thank you for this valuable suggestion. In the revised manuscript, we therefore substantially expanded the experimental section to cover deeper networks, additional datasets, more challenging regimes, and a wider range of activation functions and activation scales. In response to the reviewers' comments, additional experiments were conducted, and the manuscript has been revised as follows:
>
> - We **added experiments on CIFAR-10 and CIFAR-100**, reporting depth–accuracy curves in **Figures 27-30**, and we now also include PINN experiments for the Black–Scholes and Burgers equations in **Section 5.2 and Appendix C.4 (Figures 7 and 38)**.
> - We added batch-normalization-free experiments, **including an 800-layer network of width 32 and other deep-and-narrow configurations in Figures 29-32**, which show that our initialization trains reliably without batch normalization whereas Gaussian initializations with or without BN often fail to converge or yield much lower accuracy.
> - We further demonstrate that, for odd–sigmoid activations $f \in \mathcal{F}$, scaled activations $\alpha f(x)$ remain trainable under our initialization even for $\alpha$ as large as $10^9$, while Gaussian initializations break down in this extreme scale regime, as summarized in **Section 5.2, Figure 37, and Tables 2–7**.
> - We performed **10 independent runs for the experiments reported in Table 1** and **5 independent runs for those in Figures 7, 14, 15, 16, 17, and 38**. Across these repetitions, we consistently observed performance gaps that exceed the run-to-run variation.

---

> ### Author Response · Authors · 2025-12-03
> **Official Comment by Authors**
>
> **W4** *"From a theory perspective comparison with a number of prior works appears to be missing, particularly Hayou et al. (2019) and Murray et al. (2021) where there appears to be a clear conceptual overlap with regard to analyzing the role of the activation."*
>
> A4. Thank you for this valuable suggestion. Murray et al. [3] fix the initialization family (Gaussian/orthogonal) and design the activation $\phi$ (in particular, the size of its linear region $a$) to satisfy EOC-like conditions. Since Murray et al. focus on activation design under standard Gaussian/orthogonal initializations, while our contribution is an initialization scheme for a fixed activation class, we do not position their method as a competing baseline in our experiments and instead include it as a conceptual reference.
> In response to the reviewers comments, additional experiments were conducted, and the manuscript has been revised as follows:
>
> - We **incorporated the EOC initialization [2]** as an additional baseline and **reran all relevant experiments**.
> EOC also defines a tanh-like function class and proposes an initialization scheme that can, in principle, be applied to a broad family of activations. In our revision, we systematically compare EOC initialization with the proposed initialization across a wide range of activation scales, network depths and widths, and with/without Batch Normalization. In all these settings, the proposed initialization **consistently outperforms EOC initialization**, with especially clear advantages for deep and narrow networks, extreme activation scales, and BN-free training.
>
> - We **have added an edge-of-chaos analysis in Section A.3**, where we compute the backpropagation factor $\chi_\ell$ for our initialization, and we **show in Section 4.3** that the diagonal term $\omega$ in our diagonal–plus–noise form yields a larger well-behaved region in $(L,\sigma_z)$ than Gaussian initializations.
>
> - We have evaluated $\chi_\ell$ over a grid of depths $L$ and noise scales $\sigma_z$ and **added Figure 5**, which shows that the region where $\chi_\ell \approx 1$ is significantly larger for our initialization than for Gaussian baselines.
>
> - We **have added Figure 4, which confirms on actual neural networks** that the proposed initialization is more robust to both depth $L$ and noise scale $\sigma_z$ than Gaussian initializations.
>
> ---
>
> **Q1** *"It is not clear to me what the analysis of the recursion $x_{n+1} = f(a_n x_n)$
>  and the ``pitchfork bifurcation'' add beyond established initialization principles such as initialization on the edge of chaos? How does this proposed framework differentiate itself from prior `edge-of-chaos or dynamical isometry analyses"*
>
> **Q2** *"Can you contrast and compare your theoretical results with Hayou et al. (2019) and Murray et al. (2021)?"*
>
> A5. Thank you for this valuable suggestion. We address Q1 and Q2 together.
> In the revised manuscript, we have added Murray et al. [3] as a reference only, rather than a competing baseline, for the reasons explained above. We have revised the manuscript as follows:
>
> - We **revised Sections 4.2 and B.4 to provide a more detailed** and transparent derivation of the proposed initialization, clarifying how the scalar recursion $x_{n+1} = f(a_n x_n)$ and pitchfork bifurcation analysis motivate our choice of $\omega$ and $\sigma_z$, and how this perspective differs from classical edge-of-chaos analyses.
>
> - We **added an edge-of-chaos analysis in Section A.3,** where we explicitly compute the backpropagation factor $\chi_\ell$ for our initialization, and we **show in Section 4.3** that the diagonal term $\omega$ in our diagonal–plus–noise form yields a larger well-behaved region in $(L,\sigma_z)$ than Gaussian-based initializations.
>
> - We complemented this theory with additional experiments, showing that the proposed initialization is more robust to network size **(Section C)** and to **activation scaling (Figure 29, 31, Tables 2–7)** than Gaussian (EOC) initializations.
>
>
>
>
> [1] Schoenholz, Samuel S., et al. "Deep information propagation." arXiv preprint arXiv:1611.01232 (2016).
>
> [2] Hayou, Soufiane, Arnaud Doucet, and Judith Rousseau. "On the impact of the activation function on deep neural networks training." International conference on machine learning. PMLR, 2019.
>
> [3] Murray, Michael, Vinayak Abrol, and Jared Tanner. "Activation function design for deep networks: linearity and effective initialisation." Applied and Computational Harmonic Analysis 59 (2022): 117-154.

---

### Author Response · Authors · 2025-12-03
**Summary of Revisions**

Dear Reviewers and AC,

We would like to thank all the reviewers for taking the time to review our work and for providing valuable feedback.
The latest revision of our paper has been uploaded, addressing all comments and queries raised by the reviewers. Edits in the PDF have been highlighted in red. Below, we provide a summary of the changes made to our work.

---

## Summary of Contributions
- We propose a **non-Gaussian initialization** for an odd–sigmoid activations, motivated by their pitchfork-bifurcation structure.

- We **introduce a new initialization design principle** by viewing the FFNN as a collection of parallel scalar dynamical systems and using the coordinate-wise negative rate as an explicit design criterion, rather than relying on classical mean-field assumptions.

- We show that the negative rate in the scalar surrogate and in the actual FFNN are **systematically  related**, which justifies using the surrogate to choose $\sigma_z$.

- We demonstrate that our initialization enables stable training **across a wide range of activation scales (e.g., $10^9 \tanh(x)$)** and very deep networks **(up to $800$ layers, without BatchNorm)**, and that it consistently outperforms Gaussian-based initializations such as EOC, Xavier, and He.

- We further show on **PINNs** that, because our initialization remains trainable across a wide range of activation scales, it effectively targets suitable output ranges and yields better regression performance than Gaussian-based initializations.



## Writing and Clarity
- We have revised the manuscript to **clarify notation, tighten the scope**, and better position our contributions relative to prior work.

- We revised the title to more clearly emphasize the comparison between standard Gaussian initializations and proposed initialization.

- We have made **minor edits** throughout the paper for clarity and space efficiency.

- We clarified that existing signal–propagation mean-field results [1,2] are derived under Gaussian initializations and wide/infinite-width assumptions, and that our goal is to design an initialization that works **robustly also for deep and narrow networks (Section 4.3)**.

- We **revised Section 4.2 and added a new Section 4.3** to make it explicit how our perspective **differs from classical Gaussian signal propagation theory**.

- We **clarified the negative rate in Section 4.2** and provide its **motivation and derivation in the new Section B.4, including its role in calibrating the noise scale.**

- We have **added Section B.4** to clarify the role of the scalar surrogate and its connection to the negative-rate based calibration.

- We **reworked all figures** with overly small text (including Figure 2) to improve readability.

- We improved the statements and **explanations of Theorem 4.6 and Theorem 4.7.**

## Relation to Prior Work and EOC Analyses
- We **incorporated the EOC initialization from Hayou et al. [1]** as an additional baseline and **reran all relevant experiments.**

- We have **added an edge-of-chaos analysis in Section A.3**, where we explicitly compute the backpropagation factor $\chi$
for our initialization, and **we show in Section 4.3** that the diagonal term $\omega$ in our diagonal–plus–noise parameterization yields a larger well-behaved region in $(L, \sigma)$ than Gaussian/EOC initializations.

- We also **added Section B.4** to explain why it is natural to work with **a scalar surrogate $N_\ell= 1$**
and **data-independent gains** in that model, and how this connects back to realistic multi-neuron layers.

## Experiments
- We **have added experiments** with and without Batch Normalization in **Section C.2**. The proposed initialization achieves strong performance both without BN and against Gaussian/EOC baselines with or without BN.
- We **have added experiments** across **a wide range of network sizes, including 800-layer** deep-and-narrow networks, to demonstrate that our initialization remains robust even without Batch Normalization **(Sections C.1 and C.2)**.
- We have performed each experiment with **multiple random seeds** to assess robustness to initialization noise.
- We **have added experiments** that systematically study robustness to activation scaling for odd–sigmoid activations of the form **$a f(x)$, including the extreme case $a = 10^9$**  **(Section C.2 and C.4)**.
- We **have added PINN experiments** to demonstrate that our initialization is also beneficial for regression-style tasks, where the output scale and avoidance of saturation are crucial **(Section C.4)**.

We have included all experimental results in our revised paper.

Best regards,

Authors of submission 23117


[1] Hayou, Soufiane, Arnaud Doucet, and Judith Rousseau. "On the impact of the activation function on deep neural networks training." International conference on machine learning. PMLR, 2019.

[2] Schoenholz, Samuel S., et al. "Deep information propagation." arXiv preprint arXiv:1611.01232 (2016).

---

> ### Author Response · Authors · 2025-12-03
> **Major concern 1**
>
> ### The **main major concerns** raised by the reviewers can be summarized as follows:
>
>  **1**. *How our approach differs from prior edge-of-chaos–based initialization methods, and*
>
>  **2**. *How the hyperparameter $\sigma_z$ of the proposed initialization should be chosen in practice*.
>
> We **have addressed both points in the revised manuscript**, and we briefly elaborate on these two issues in more detail below.
>
> ---
>
> ### 1. How our approach differs from prior edge-of-chaos–based initialization methods.
>
> Prior edge-of-chaos (EOC) [1,2,3] are formulated in a wide or infinite-width mean-field setting with i.i.d. Gaussian weights and biases. In these works, the main design principle is to choose the weight variance (and sometimes bias variance) so that the forward variance and the average gradient norm are neither exploding nor vanishing as depth grows.
>
> In contrast, our framework departs from this setting in two key ways:
>
> - **Diagonal-plus-noise structure instead of purely Gaussian weights.**
>   We explicitly parameterize each layer as
>     $W^\ell = \omega I +Z^\ell, \qquad Z^\ell_{ij} \sim \mathcal{N}\\bigl(0, \sigma_z^2 / N_{\ell}\bigr),$
>
>   where the diagonal term $\omega$ is chosen based on the fixed-point structure of odd–sigmoid activations. This “identity-like” structure is not covered by standard EOC analyses, which assume fully i.i.d. Gaussian weights with zero mean. As we show in Figure 5 and Figure 24, this diagonal term dramatically enlarges the region in $(L,\sigma_z)$ where both forward dispersion and backward gradients remain well behaved.
>
> - **Focus on robustness across depths, widths, and activation scaling (beyond wide Gaussian limits).**
>  EOC-based schemes are typically justified in the limit of wide networks and are empirically most stable when combined with Batch Normalization. By contrast, our goal is to obtain an initialization that works reliably across a broad range of depths and widths, and in particular remains stable for deep-and-narrow odd–sigmoid networks without normalization and under a wide range of activation scales $\alpha f(x)$. To this end, we reinterpret the FFNN as a collection of parallel scalar dynamical systems and design the initialization via a scalar surrogate that directly controls the sign statistics (negative rate) at a target depth. This leads to a different calibration objective than the variance-only criteria used in EOC analyses.
>
> Empirically, these differences manifest as follows.
> Figures 3–5 and 19–23 show that the proposed initialization preserves both forward dispersion and backward gradients over a much wider range of depths and noise scales than Gaussian/EOC baselines, especially in deep-and-narrow settings and for extreme activation scales (e.g., $a f(x)$ with $a = 10^9$. Moreover, our method trains very deep networks (up to 800 layers) without Batch Normalization, whereas Gaussian/EOC initializations either fail to converge or require BN to remain stable (Sections 4.3, 5.2, C.1–C.2).
>
> ---
>
> [1] Schoenholz, Samuel S., et al. "Deep information propagation." arXiv preprint arXiv:1611.01232 (2016).
>
> [2] Hayou, Soufiane, Arnaud Doucet, and Judith Rousseau. "On the impact of the activation function on deep neural networks training." International conference on machine learning. PMLR, 2019.
>
> [3] Murray, Michael, Vinayak Abrol, and Jared Tanner. "Activation function design for deep networks: linearity and effective initialisation." Applied and Computational Harmonic Analysis 59 (2022): 117-154.

---

> ### Author Response · Authors · 2025-12-03
> **Major concern 2**
>
> ### The **main major concerns** raised by the reviewers can be summarized as follows:
>
>  **1**. *How our approach differs from prior edge-of-chaos–based initialization methods, and*
>
>  **2**. *How the hyperparameter $\sigma_z$ of the proposed initialization should be chosen in practice*.
>
> We **have addressed both points in the revised manuscript**, and we briefly elaborate on these two issues in more detail below.
>
> ---
>
> ### 2. How the hyperparameter $\sigma_z$ of the proposed initialization should be chosen in practice.
>
> Regarding the use of a scalar surrogate with $N_\ell = 1$, our intention is to work at the level of a single coordinate, as made explicit in Section B.4. Starting from the elementwise representation $x_i^{\ell+1} = f\bigl(a_i^{\ell+1} x_i^{\ell}\bigr)$
> each $a_i^{\ell+1}$ acts as an effective scalar gain for neuron $i$. Under our proposed diagonal–plus–noise initialization
>
>   $W^\ell = \omega I + Z^\ell,\quad Z^\ell_{ij} \sim \mathcal{N}\bigl(0,\sigma_z^2 / N_{\ell-1}\bigr),$
>
> Lemma 4.5 shows that, conditional on $x^\ell$, the effective gain satisfies
> $a_i^{\ell+1} \sim \mathcal{N}\!\left(\omega,\ \frac{\sigma_z^2}{N_\ell}\left(1 + \sum_{j \ne i} \left(\frac{x_j^\ell}{x_i^\ell}\right)^2\right)\right)$,
> so $\omega$ is the mean of the gain distribution while the variance is still data dependent. To obtain a tractable calibration rule, we therefore introduce a scalar surrogate model in which the gains are approximated by i.i.d.variables $a_\ell \sim \mathcal{N}(\omega,\sigma_z^2)$ and define the negative rate as
> $\pi_L(\sigma_z) := \mathbb{P}(x_L < 0 \mid x_0 > 0)$.
> Because $f \in \mathcal{F}$ is odd and strictly increasing, the sign of $x_i^\ell$ is completely determined by the product of these scalar gains along the path, and the distribution of sign flips is the same for any coordinate. In this sense, tracking a single scalar chain with $N_\ell = 1$ amounts to tracking an arbitrary coordinate and provides a representative surrogate for the network-wide negative-rate statistics.
>
>
> In Section B.4, we study the relationship between the surrogate negative rate $\pi_L(\sigma_z)$ and the FFNN-driven negative rate $\tilde{\pi}_L$ and show that they are exponentially related in depth. We then use this relation to calibrate $\sigma_z$ so that the FFNN-driven negative rate at the final layer is close to a desired value.
>
> Empirically, Figures 14 - 17 confirm that the $\sigma_z$ obtained from this calibration yields the best validation accuracy for deep networks, while performance degrades as we move to nearby, non-calibrated values of $\sigma_z$. This supports the use of the $N_\ell = 1$ scalar surrogate as a practical and quantitatively accurate guideline for choosing $\sigma_z$ in the full FFNN.

---

### Meta-Review · Area_Chair_QXkV · 2026-01-06

**Summary:**

This paper proposes a weight initialization scheme tailored to odd–sigmoid activation functions (e.g., tanh-like nonlinearities), aiming to preserve forward signal magnitude and avoid saturation or collapse in deep networks without relying on normalization layers such as Batch Normalization. The approach is motivated by a fixed-point and scalar-dynamical-system analysis of activation propagation, leading to a calibration rule based on a “negative rate” criterion. Reviewers generally found the exposition readable and the mathematical derivations internally consistent, and acknowledged that designing initialization strategies specialized to certain activation classes is a reasonable direction.

**Reviewer Concerns:**

The rebuttal addressed several clarity and completeness issues. In particular, the authors added comparisons to edge-of-chaos (EOC) initializations, expanded experiments to deeper and narrower networks, included additional datasets (CIFAR-10/100 and PINN tasks), reported multiple random seeds, and augmented the theory with analyses of backward signal propagation and gradient-scale behavior. Definitions of previously unclear concepts (e.g., negative rate, scalar surrogate) were clarified, figures were improved, and the relationship to prior works such as Hayou et al. (2019) and Murray et al. (2021) was discussed more explicitly.

Despite these improvements, core concerns remain unresolved. Multiple reviewers emphasized that the theoretical contribution still heavily overlaps with established signal-propagation and fixed-point analyses, particularly edge-of-chaos and mean-field frameworks, with the main distinction being restriction to a narrow class of odd–sigmoid activations. The significance of key theoretical results (e.g., Theorem 4.6 and 4.7) remains questionable due to strong and unrealistic assumptions (e.g., scalar or width-one surrogates, independence and Gaussianity of gains), and it is unclear to what extent these results meaningfully extend to realistic multi-neuron networks. While the authors argue that the scalar surrogate captures essential behavior, reviewers remained unconvinced that this abstraction provides fundamentally new insight beyond existing analyses.

**Reviewer Scores:**

Reviewer iEW8 (score: 2): This reviewer found the contribution poor relative to prior work, citing substantial overlap with established theory, limited experimental scope, and unclear added value beyond edge-of-chaos analyses. While the rebuttal expanded experiments and comparisons, these steps do not fundamentally change the novelty assessment, so the score would remain at 2.

Reviewer Wx7z (score: 2): This reviewer raised deep concerns about clarity, theoretical assumptions, and the significance of the main theorems, as well as the paper’s close dependence on prior fixed-point analyses. Although presentation issues were improved, the core limitations persist, so the score would remain at 2.

Reviewer 4CNB (score: 4): This reviewer viewed the work as fair but limited, noting restrictive assumptions, unclear practical implications, and evaluation choices. The rebuttal addressed many technical questions, but the overall contribution level remains modest, so the score would likely remain at 4.

---

### Decision · Program_Chairs · 2026-01-26

Reject